# Walrus: A Cross-domain Foundation Model for Continuum Dynamics

**Michael McCabe** [1 2 3]  **Payel Mukhopadhyay** [1 4]  **Tanya Marwah** [1 2]  **Bruno Régaldo-Saint Blancard** [1 2]
**François Rozet** [1 2 5]  **Cristiana Diaconu** [4]  **Lucas Meyer** [1 2]  **Kaze W. K. Wong** [1]  **Hadi Sotoudeh** [1 4]  **Alberto Bietti** [1 2]
**Irina Espejo** [1 2 3]  **Rio Fear** [4]  **Siavash Golkar** [1 2 3]  **Tom Hehir** [1 4]  **Keiya Hirashima** [2 6]  **Geraud Krawezik** [1 2]
**François Lanusse** [1 2 7]  **Rudy Morel** [1 2]  **Ruben Ohana** [1 2]  **Liam Parker** [1 2 8]  **Mariel Pettee** [1 9]  **Jeff Shen** [1 10]
**Kyunghyun Cho** [1 11 12]  **Miles Cranmer** [1 4]  **Shirley Ho** [1 2 3 10]

## Abstract

Foundation models have transformed machine learning for language and vision, but achieving comparable impact in physical simulation remains a challenge. Data heterogeneity and unstable long-term dynamics inhibit learning from sufficiently diverse dynamics, while varying resolutions and dimensionalities challenge efficient training on modern hardware. Through empirical and theoretical analysis, we incorporate new approaches to mitigate these obstacles, including a harmonic-analysis–based stabilization method, load-balanced distributed 2D-3D training strategies, and compute-adaptive tokenization. Using these tools, we develop WALRUS, a transformer-based foundation model for fluid-like continuum dynamics. WALRUS is pretrained on nineteen diverse scenarios spanning astrophysics, geoscience, rheology, plasma physics, acoustics, and classical fluids. Experiments show that WALRUS outperforms prior foundation models on both short- and long-term prediction horizons on downstream tasks and across the breadth of pretraining data, while ablation studies confirm the value of our contributions to forecast stability, training throughput, and transfer performance over conventional approaches. Code and weights are released for community use[1].

---

[1]Polymathic AI [2]Flatiron Institute [3]New York University [4]University of Cambridge [5]University of Liège [6]RIKEN Center for iTHEMS [7]Université Paris-Saclay, Université Paris Cité, CEA, CNRS, AIM [8]University of California Berkeley [9]University of Wisconsin-Madison [10]Princeton University [11]Prescient Design, Genentech [12]CIFAR Fellow. Correspondence to: Michael McCabe <mmccabe@flatironinstitute.org>.

*Proceedings of the 43rd International Conference on Machine Learning*, Seoul, South Korea. PMLR 306, 2026. Copyright 2026 by the author(s).

[1]Github: PolymathicAI/walrus    Visualizations

## 1. Introduction

In recent years, researchers have explored data-driven emulation of physical systems as an alternative to numerical simulation. Numerical simulation is a cornerstone of modern engineering and scientific workflows, enabling practitioners to forecast the evolution of complex systems (Eyring et al., 2016; Berger & LeVeque, 2024), optimize engineering design (Biegler et al., 2003; Mohammadi & Pironneau, 2004), and infer parameters of unknown systems (Cranmer et al., 2020; Lemos et al., 2023). However, the incredible accuracy of these methods can exceed application tolerances while the cost can become a bottleneck. Furthermore, conventional simulation requires a strict, accurate definition of the system in terms of *partial differential equations* (PDEs) which can be infeasible for multi-physics scenarios or partial-information problems. Data-driven emulation has shown particular promise for such poorly-observed systems (Pfau et al., 2020; Jumper et al., 2021; Lam et al., 2022).

To address the data demands of modern deep learning, researchers have pursued the foundation model paradigm of pretraining large models on massive, diverse datasets (Subramanian et al., 2023b; McCabe et al., 2023a; Herde et al., 2024). However, physical emulation poses unique challenges: systems evolve over multiple temporal and spatial scales, learned models often become unstable over long horizons, and heterogeneous data (varying resolutions, dimensionalities, physical fields) challenges modern training architectures that favor consistent inputs. As a result, existing foundation models have focused on relatively homogeneous data, often only 2D problems or fixed resolutions.

Our work contributes to overcoming these barriers through: (1) *patch jittering*: a harmonic-analysis–derived stabilization method reducing long-horizon error in 84% of pretraining scenarios; (2) *2D-to-3D augmentation*: jointly handling 2D and 3D data by treating 2D data as a plane randomly embedded in 3D space; (3) *adaptive-compute tokenization*: dynamically allocating compute based on resolution or problem complexity; and (4) *topology-aware sampling*: tying sampling to distribution topology in order to increasing

training throughput by 262%.

Using these tools, we develop WALRUS, a 1.3B parameter transformer pretrained on 19 scenarios spanning astrophysics, geoscience, rheology, plasma physics, acoustics, and classical fluids with both 2D and 3D data. Experiments show that WALRUS outperforms prior foundation models on both short and long forecast horizons and that diversity-first pretraining leads to stronger transfer performance.

## 2. Background

**Problem Setting.** Our interest is in data-driven emulation of physical systems, specifically at the level of continuum operators. That is, for an arbitrary physics-driven spatiotemporal system $S$, we model the evolution of continuous-valued state variable $\boldsymbol{u}^S(\boldsymbol{x}, t) : \prod_{i \in [d]} [0, L_i^S] \times [0, \infty) \to \mathbb{R}^q$ where $d$ denotes the number of spatial dimensions and $q$ the number of observed fields. For modeling purposes, the system is discretized in both space and time. A snapshot $\boldsymbol{u}_t^S \in \mathbb{R}^{N^S}$ represents the value of state variable $\boldsymbol{u}^S$ at $N^S$ spatial discretization points at time $t$. We use the notation $\Delta(\cdot)$ to denote the difference between two consecutive quantities in sequence, for instance $\Delta\boldsymbol{u}_{t+\Delta t}^S = \boldsymbol{u}_{t+\Delta t}^S - \boldsymbol{u}_t^S$. We will drop the superscript $S$ and refer to space and time indices by integer values when the meaning is apparent.

Often these systems' evolutions will be described by known partial differential equations; however, for a foundation model, we do not want to limit ourselves to operating only in such settings. Instead, we adopt a broader pretraining objective: to identify a model $\mathcal{M}$ such that for any system $S$ sampled from a distribution over physical systems, given a sequence of $\tau$ snapshots $\boldsymbol{U}_t^S = [\boldsymbol{u}_{t-\tau\Delta t}^S, \ldots, \boldsymbol{u}_t^S]$ we have:

$$\boldsymbol{u}_{t+\Delta t}^S \approx \boldsymbol{u}_t^S + \mathcal{M}(\boldsymbol{U}_t^S). \tag{1}$$

We discuss the trade-offs inherent to the use of history vs. known parameters (PDE coefficients, explicit constitutive models, etc) in Appendix E. In this work, we use uniformly spaced samples in time such that the model must infer the relative timescales of physical processes from the behavior of the provided historical snapshots.

### 2.1. Related Work

Our work addresses data-driven emulation of physical systems, learning to forecast directly from observations without explicit governing equations (Kovachki et al., 2023; Lu et al., 2019; Li et al., 2020; 2021). This contrasts with physics-informed methods that encode known PDEs (Lagaris et al., 1998; Raissi et al., 2019; Bruna et al., 2022) or hybrid models augmenting numerical solvers (Kochkov et al., 2021; Duraisamy et al., 2019). The computational cost of training such models has motivated transfer learning approaches (Goswami et al., 2022; Li et al., 2021; Subel et al., 2023).

Early work explored in-context learning for 1D systems (Yang et al., 2023) and pretraining for linear PDEs (Subramanian et al., 2023a) while MPP (McCabe et al., 2023a) introduced autoregressive pretraining for nonlinear multi-dimensional systems. More recent efforts have explored denoising objectives (Hao et al., 2024), meta-learning (Morel et al., 2025), and text-conditioned models (Shen et al., 2024). Other work has explored flexible architectures suitable to the multi-physics setting (Herde et al., 2024; Takamoto et al., 2023; Alkin et al., 2024; Rahman et al., 2024; Holzschuh et al., 2025), using video pretrained models (Nguyen et al., 2025), or improving context-based approaches (Serrano et al., 2025; Cao et al., 2025).

## 3. Walrus Model

### 3.1. Architecture

WALRUS employs a space-time factorized transformer architecture (Ho et al., 2019; McCabe et al., 2023a) where alternating operations within a block attend along the space and time axes of space-time tensor-structured data. The procedure can be seen in Figure 1. Spatial processing uses the parallelized attention developed in (Wang, 2021) using axial RoPE (Su et al., 2023; Lu et al., 2023; 2024) for position encoding. Along the time axis, WALRUS uses causal attention with T5-style relative position encoding (Raffel et al., 2020). QK normalization (Dehghani et al., 2023) is used in both space and time blocks to improve training stability. Full architectural details can be found in Appendix B, but we highlight here the notable design decisions before diving into our novel contributions.

**Compute-Adaptive Compression.** We implement adaptive-compute tokenization using Convolutional Stride Modulation (Mukhopadhyay et al., 2026, CSM) in our encoder and decoder modules to natively handle data at varying resolutions by adapting the level of downsampling/upsampling in each encoder/decoder block. Prior emulation foundation models have used fixed compression encoders rendering them inflexible to varying resolution in downstream tasks. CSM allows us to alter the downsampling rate by adjusting the stride. During pretraining, to maximize device utilization, we choose a fixed number of tokens per axis and adjust the downsampling factor accordingly such that data of the same dimensionality produces similar numbers of tokens per frame across datasets. This process is described in more detail within the context of WALRUS and the hMLP described below in Appendix B.2.

**Shared Encoder-Decoder.** All physical systems $S$ of a given dimensionality $d$ share a single encoder and decoder block differing only in the first projection forcing the model to learn generalizable features. Since this study only features 2D and 3D data, this is a total of two encoders and decoders.

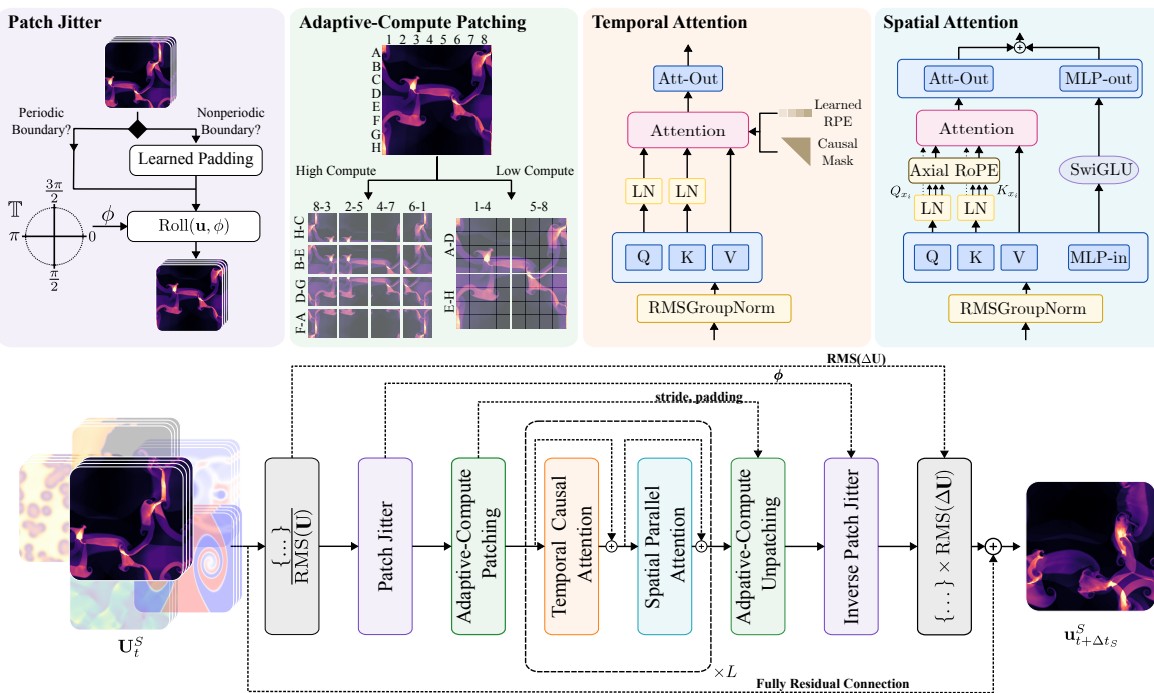

*Figure 1.* WALRUS is a modern transformer incorporating novel stabilizing techniques and recent adaptive-compute methods to learn from a highly diverse set of physical dynamics. WALRUS takes as input a short sequence of snapshots and predicts the next step in the sequence.

We use an Hierarchical MLP (hMLP) and transposed hMLP for lightweight encoding and decoding (Touvron et al., 2022) and the sub-selected projection approach of MPP (McCabe et al., 2023a) to handle varying input sizes $q^S$. These are lightweight alternatives to standard patchification in which the patching process is broken into multiple hierarchical stages separated by nonlinearities and normalization layers.

**RMS GroupNorm.** WALRUS employs RMSGroupNorm as the standard normalization approach within each transformer block. This is GroupNorm (Wu & He, 2018) implemented as RMSNorm (Zhang & Sennrich, 2019) over each group. Empirically, we found the larger normalization scope to improve performance in early development stages.

**Asymmetric Input/Output Normalization.** WALRUS learns to predict $\Delta u_{t+1}^S$ which is typically distributed differently from the input values $U_t^S$. We therefore use asymmetric normalization for model inputs and outputs. In particular, during pretraining, we normalize each unique input field by $\text{RMS}_{(\text{Time}\times\text{Space})}(U_t^S)$ or the RMS computed over space and time of the given field over the provided history. The output field is then de-normalized by $\text{RMS}_{(\text{Time}\times\text{Space})}(\Delta U_t^S)$.

### 3.2. Patch Jittering for Rollout Stability

Resampling used in ViT-style patch-based tokenization introduce spectral artifacts that accumulate during autoregressive rollouts, often manifesting as grid-like patterns in long-horizon predictions. Previous work on aliasing in physical emulation has focused on the challenge posed by nonlinear

operations (Raonić et al., 2023; McCabe et al., 2023b), but here we show that even in an idealized setting where we ignore nonlinear effects, the resampling operations (strided convolution and transposed convolutions) in ViT-style architectures alone produce a distinct spectral structure which leads to this grid-imprinting:

**Lemma 3.1 (Frequency-Domain Artifacts)** *For a bandlimited signal $u \in L^2(\mathbb{T})$ sampled uniformly at $N$ discrete points downsampled through strided convolution with stride equal to patch size (stride $P$, filter $g$) to $M$ points followed by upsampling using transposed convolution (filter $h$) of the same shape, the frequency response $\hat{v}$ satisfies:*

$$\hat{v}[k] = \hat{h}[k]\hat{g}[k]\hat{u}[k] + \sum_{j=1}^{P-1} \hat{h}[k]\hat{g}[k+jM]\hat{u}[k+jM] \quad (2)$$

*where the summation terms represent aliased frequencies introduced by resampling.*

We propose to address these artifacts through **patch jittering**: during each autoregressive step, randomly translating input data by offset $s$ before tokenization, then inverting this translation after reconstruction. In our simplified setting, it can be shown that this eliminates aliasing in expectation.

**Proposition 3.1 (Aliasing Cancellation via Jittering)** *Under the setting of Lemma 3.1, if we apply a random phase shift (translation) $s$ before strided convolution and invert the translation after transposed convolution, the*

*expected frequency response eliminates aliasing:*

$$\mathbb{E}_s[\hat{v}[k]] \propto \hat{h}[k]\hat{g}[k]\hat{u}[k]. \tag{3}$$

As a simple Monte Carlo averaging procedure, the convergence of this estimator in MSE follows the well-known $O(1/T)$ rate (Caflisch, 1998). The proofs (Appendix A.1) follow from standard harmonic analysis: Lemma 3.1 directly applies the frequency-domain representation of strided operations, while Proposition 3.1 exploits the Fourier shift property and the orthogonality of the trigonometric polynomials to show the aliased terms vanish in expectation. While in practice, these results are complicated by nonlinearity and boundary conditions, they provide clear intuition of how stochastic translation can reduce compounding errors.

In Appendix A.1.3, we show clear empirical value of the approach, even at the single-sample estimate used in practice. Evaluating on 20 randomly chosen validation trajectories per pretraining dataset, patch jittering reduces median long-horizon VRMSE by 54% across scenarios, with the number of unstable rollouts (VRMSE $\geq 1$) decreasing from 10 to 3 out of 19 datasets (full results in Appendix Table 2). This improvement appears in 84% of pretraining scenarios with negligible computational overhead.

# 4. Walrus Training

A key challenge for multi-physics foundation models is that, unlike in domains such as NLP, the available data tend to be larger on disk while only capturing a smaller fraction of all relevant physical dynamics. It is therefore vital to avoid overfitting to any given data source. We design WALRUS's training strategy to maximize the level of heterogeneity seen during training through aggressive diversity-focused augmentation. This choice requires sophisticated distribution strategies to maintain high levels of hardware utilization during training. During finetuning, we include the ability to relax some of these restrictions to improve fitting to downstream tasks which we discuss in Appendix C.2.1.

## 4.1. Augmenting into a Combined 2D-3D Space

**Dimension padding.** To jointly represent 2D and 3D Euclidean data within a single space, we begin by treating all 2D data as thin planes within a 3D space. The data is first projected into 3D by appending a singleton dimension and zero-padding the tensor-valued fields. For example in the left panel of Figure 2, a 2D velocity field of size $(H \times W)$ with values $\{v_x, v_y\}$ would be padded to size $(H \times W \times 1)$ and $\{v_x, v_y, 0\}$.

**Tensor-law Aware Augmentations.** Euclidean data is then augmented by the application of tensor-law aware transformations. Here this is limited to the full octahedral group of $90 \deg$ rotations and reflections to avoid misaligned bound-

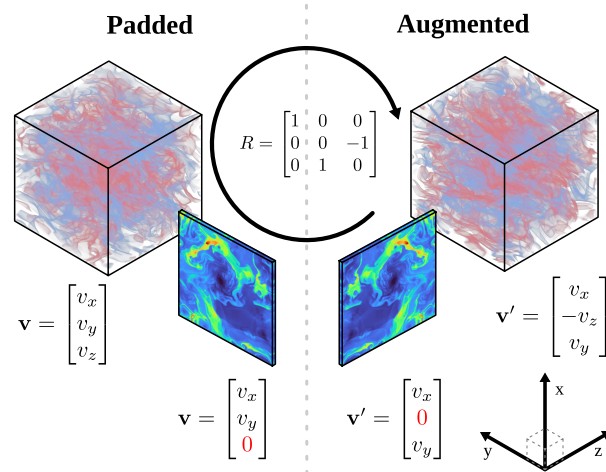

*Figure 2.* All raw data is projected into 3D by appending singleton dimensions and zero-padding tensor-valued fields prior to applying symmetry-preserving augmentations resulting in the input data being embedded in arbitrary axis-aligned directions in 3D.

aries. After augmentation, the 2D data is now embedded in a random orientation within the larger 3D space. When we transform the reference frame, tensor-valued fields must be similarly transformed following tensor transformation laws to preserve physical consistency. For order one tensor-value fields, if the reference frame transforms under transformation $\boldsymbol{R}$, then field $\boldsymbol{u}$ should also transform as $\boldsymbol{Ru}$. For order 2 tensors, this transformation should be $\boldsymbol{RuR}^T$.

**Variable time striding.** Rather than pre-training on a single time-stride, we train the model on randomly sampled time strides so that the model must learn to infer the relative time and velocity scales from context rather than fitting to a particular scale in the training data. This approach was previously used for the VICON foundation model (Cao et al., 2025). We sample the time stride from 1-5 during pretraining.

## 4.2. Efficient Multi-Task Training

**Sampling and loss.** We balance the contributions of various datasets and regimes within those datasets through a combination of locally normalized training objectives and balanced sampling. Walrus employs a hierarchical sampling scheme where we uniformly sample a system $S$ then uniformly select a starting frame from within the dataset. We then train the model with the normalized loss:

$$\mathcal{L} = \frac{1}{q} \sum_{i=1}^{q} \frac{\|\mathcal{M}(\boldsymbol{U}_t(i)) - \Delta\boldsymbol{u}_{t+1}(i)\|_1}{\mathrm{RMS}_{\mathrm{Space}\times\mathrm{Time}}(\Delta\boldsymbol{U}_t(i))} \tag{4}$$

In this way, errors predicting changes in rapidly changing fields are down-weighted compared to predictions on slower changing fields which we expect to be more predictable.

**Efficient Distribution**. While uniform sampling of datasets produces maximally diverse batches, employing this strat-

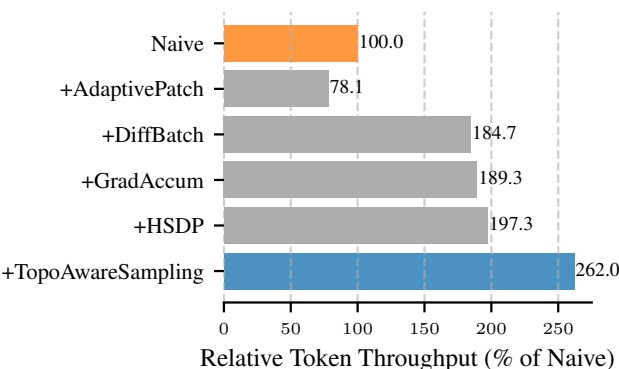

*Figure 3.* Tokenization and distribution strategies are carefully balanced to ensure each copy of WALRUS receives similarly sized buckets of tokens within each synchronous block, minimizing deadweight and maximizing throughput.

egy with hetereogeneous data creates severe load imbalance during distributed training. Patch-based tokenization ties token counts, and therefore compute, to resolution and dimensionality, so mixed 2D/3D workloads with varying resolution lead to dead cycles under distribution strategies with multiple synchronization points within each forward pass.

We address this hetereogeneity issue through three complementary strategies, balancing accuracy, efficiency, and diversity. First, adaptive-compute tokenization (Section 3.1) normalizes token counts across datasets within each dimensional regime (32/dim for 2D, 16/dim for 3D). This strictly increases the cost for lower resolution data, but balances compute between datasets. Next, we use differential batch sizes, scaling batch size and temporal history inversely with spatial token count so the dominant expense of operations scaling linearly with token count is approximately equalized. Finally, as the quadratic-in-token scaling of spatial attention still results in significant differences in cost between 2D and 3D batches, we introduce a *topology-aware sampling* strategy adapted from HSDP (Zhao et al., 2023): ranks within a sharding group sample the same dataset, while groups sample independently. When combined with gradient accumulation, this minimizes `AllGather` synchronization between hetereogeneous loads while averaging variance over micro-batches between `AllReduce` operations. Figure 3 shows these changes yield 262% throughput improvement over naive FSDP. More detail is provided in Appendix A.2.

## 5. Experiments

We design our experiments to answer three key questions about the efficacy of WALRUS as a foundation model:

1. How does WALRUS perform as a foundation model when compared with prior foundation models across a range of downstream tasks?

2. Is WALRUS truly a cross-domain foundation model? Are there particular areas of strength or weakness?

3. Does the focus on representational diversity during WALRUS's pretraining matter? Are the gains we see in other experiments purely due to scale and architectural choices or does the focus on diversity lead to improvement on downstream tasks?

**Data.** We use a mixture of datasets from the Well (Ohana et al., 2025) and FlowBench (Tali et al., 2024) for pretraining. the Well contains high resolution data derived from realistic scientific problems while FlowBench introduces geometrically complex obstacles in standard flow scenarios. This totals 19 datasets containing 63 state variables simulated over a wide variety of equations, boundary conditions, and physical parameterizations. Importantly, we use both 2D and 3D data during pretraining. To validate transfer performance, we finetune several held out datasets from the Well, Flowbench, PDEBench (Takamoto et al., 2022), PDEArena (Gupta & Brandstetter, 2023), and PDEGym (Herde et al., 2024). When provided, we use standard splits. Otherwise, we split the dataset by trajectory into 80/10/10 splits. Full data details are described in Appendix D.

**Training settings.** WALRUS was pretrained for approximately 400,000 steps following the distribution strategy described in Section 4.2 with a 2D micro-batch size of 192 and 3D size of 96. In total, WALRUS was pretrained on approximately 4 million examples from each 2D dataset and 2 million for each 3D. When finetuned on datasets used to pretrain WALRUS, all comparisons are trained on the combined volume WALRUS was shown during pretraining and finetuning (4.5 million samples for 2D, 2.5 million for 3D). Otherwise all models are given the same finetuning budget of 500k samples regardless of dataset size. All models are trained to predict the next system state using the AdamW (Loshchilov & Hutter, 2017) optimizer and a reciprocal square-root learning rate schedule (Zhai et al., 2022) with linear warmup and inverse quadratic cooldown (Hägele et al., 2024). Note that all models apart from MPP-AViT-L are finetuned with their original context lengths: WALRUS (6 2D/3 3D), DPOT (10), Poseidon (1), MPP-AViT-L (6). For AViT, we empirically found little loss of accuracy reducing the context length to match WALRUS but did not attempt to reduce further. Full details can be found in Appendix C. **Evaluation.** We primarily use VRMSE as in the Well benchmark (Ohana et al., 2025) either per-step or averaged over rollout windows for comparison as the normalization allows for easier cross-dataset comparison in limited space. As this analysis is inherently limited for longer term comparisons, we supplement this analysis in Appendix F with more exhaustive time series results along with distributional comparisons per trajectory using the Spatially Averaged W1 used in (Molinaro et al., 2025). These represent two extremes and provide different information. Pointwise metrics like VRMSE penalize deviations from the exact spatial arrangement of features, but lose informativeness over longer

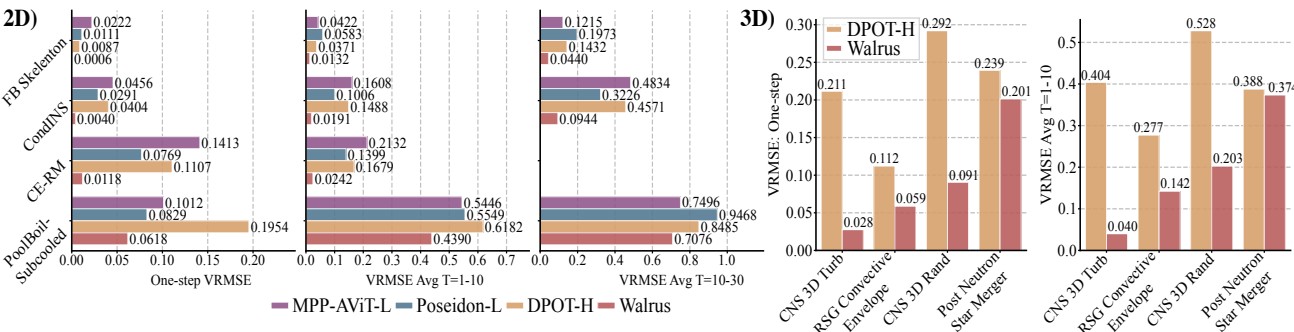

*Figure 4.* Loss (median VRMSE) on downstream tasks after finetuning each foundation model. Autoregressive prediction starts at T=17 (11) for 2D (3D) to account for model context-length differences. In cases where ground truth is shorter than window, loss is averaged over available frames. If no frames available, the window is left blank.

rollouts as trajectories decouple. On the other hand, the form of W1 used here is permutation invariant, but ensures the distribution of states matches over longer horizons. Particularly for chaotic systems, this is often the only type of comparison possible in the general setting. All metrics are defined in Appendix F.1. As many of these datasets are undergoing regime transitions and therefore prediction difficulty varies over time, all predicted trajectories begin from $T = 17$ to allow for equal comparisons with models requiring longer contexts.

It may be noted that physics-specific metrics are largely absent from this analysis. The reason for this is that the diversity of systems under analysis make many physics-specific metrics unwieldy. Metrics which assume a stationary long run distribution do not apply to regime transition problems while losses based on the governing equations are unreliable as the PDE describes infinitesimal changes while most systems under steady are sampled at a much coarser rate than the integrator step frequency used to generate the data.

### 5.1. Downstream Performance

**Experiment Settings.** The primary goal of any foundation model is to improve performance on diverse downstream tasks. To evaluate this capability in WALRUS, we finetune both WALRUS and several state-of-the-art foundation models for physical dynamics on a collection of downstream tasks in both 2D and 3D settings drawn from the Well (Ohana et al., 2025), PDEGym (Herde et al., 2024), Flowbench, (Tali et al., 2024), and PDEArena [2] (Gupta & Brandstetter, 2023). For this evaluation, we compare the performance of WALRUS (1.3B parameters) against pretrained MPP-AViT-L (McCabe et al., 2023a, 407M), Poseidon-L (Herde et al., 2024, 628M), and DPOT-H (Hao et al., 2024, 1.2B) using the highest parameter count public checkpoints available for each. Each pretrained model is finetuned with

---

[2]Note that while the CondINS task is included in DPOT's pretraining, in the finetuning setting, the field-specific embedding weights are replaced to better approximate a true downstream task.

500K samples regardless of the true size of the dataset, so for some datasets this will involve multiple full passes and others will be less than a full pass. This restriction mirrors the common downstream setting in which compute and data availability is significantly lower than in pretraining. The allocation of these samples is described in Appendix C.3.

We evaluate these models across a range of time steps. One-step VRMSE tracks how well the model does across the various regimes tracked in the dataset and provides a direct evaluation of generalization to new data within the framework of the training task. However, medium (1-10 step) and medium (11-30 step) rollouts are used for evaluating effectiveness at the emulation task itself which typically requires multi-step rollouts. For chaotic systems, longer-term pointwise losses like VRMSE can become meaningless as decoupling from the true trajectory is inevitable at which point any effective model will saturate to similar loss. Full loss over time plots can be seen in Appendix Figure 13 while direct numerical versions of these plots can be found starting in Appendix Table 13.

**Analysis.** WALRUS outperforms the baseline models across all downstream tasks providing an average 63.6% reduction in one-step loss, 56.2% for shorter trajectories, and 48.3% for medium-range trajectories (Figure 4). For non-chaotic tasks, the lower artifact generation of WALRUS from patch jittering leads to remarkably consistent performance over time despite the bulk of the network operations occurring in the fully compressed/tokenized space. In more stochastic spaces like the PoolBoilSubcool from BubbleML, while WALRUS initially has a significant lead, this lead is reduced over longer rollouts as the difficulty of inferring material and burner properties from a short history makes a larger impact. In Figure 34, we see that the model continues to generate bubbles over time rather than converging to a smooth mean state.

3D performance is of particular interest as most real-world physics of simulation interest is truly three-dimensional. Numerical solvers for 3D systems tend to be significantly

| Dataset | VRMSE: One-step | | | | VRMSE: Avg $T \in [1:20]$ | | | | VRMSE: Avg $T \in [21:60]$ | | | |
|---|---|---|---|---|---|---|---|---|---|---|---|---|
| | MPP-AViT-L | Poseidon-L | DPOT-H | **Walrus** | MPP-AViT-L | Poseidon-L | DPOT-H | **Walrus** | MPP-AViT-L | Poseidon-L | DPOT-H | **Walrus** |
| *Biological and chemical behavior* | | | | | | | | | | | | |
| Active Matter | 0.0157 | 0.0214 | 0.0476 | **0.0057** | 0.3195 | 0.3355 | 0.5272 | **0.1262** | 1.2481 | 1.3015 | 1.2867 | **1.2451** |
| Gray-Scott | nan | 0.0048 | 0.0061 | **0.0001** | nan | 0.0411 | 0.1354 | **0.0278** | nan | 0.1295 | 0.6567 | **0.1268** |
| *Acoustics and wave propagation* | | | | | | | | | | | | |
| Maze* | 0.0337 | 0.0116 | 0.0126 | **0.0099** | 0.0797 | 0.0785 | 0.0350 | **0.0345** | 0.1390 | 0.2429 | **0.0543** | 0.0560 |
| Inclusions | 0.0432 | 0.0127 | 0.0199 | **0.0089** | 0.1362 | 0.0510 | 0.0500 | **0.0430** | 0.4940 | 0.1407 | **0.1328** | 0.1427 |
| Discontinuous | 0.0097 | 0.0035 | 0.0055 | **0.0021** | 0.0460 | 0.0255 | 0.0162 | **0.0093** | 0.2310 | 0.0890 | 0.0398 | **0.0385** |
| Staircase | 0.0026 | 0.0019 | 0.0017 | **0.0005** | 0.0053 | 0.0201 | **0.0022** | 0.0040 | 0.0096 | 0.0507 | **0.0031** | 0.0074 |
| *Astrophysical and geoscience applications* | | | | | | | | | | | | |
| Supernova (3D) | — | — | 0.6417 | **0.2462** | — | — | 0.7366 | **0.6673** | — | — | **0.9669** | 1.0963 |
| TGC (3D) | — | — | 0.1002 | **0.0466** | — | — | 0.3482 | **0.2889** | — | — | 0.6809 | **0.6197** |
| PlanetSWE | 0.0035 | 0.0035 | 0.0013 | **0.0004** | 0.0307 | 0.0730 | 0.0081 | **0.0046** | 0.1213 | 0.3452 | 0.0317 | **0.0207** |
| *Plasmas* | | | | | | | | | | | | |
| MHD (3D) | — | — | 0.4734 | **0.0580** | — | — | 1.0250 | **0.6487** | — | — | 1.3055 | **1.2256** |
| *Viscous fluids* | | | | | | | | | | | | |
| Rayleigh-Benard | 0.0264 | 0.0215 | 0.0288 | **0.0059** | 0.4109 | 1.3819 | 3.4868 | **0.0992** | 0.7468 | 0.9586 | 1.2664 | **0.6441** |
| Shear Flow | 0.0071 | 0.0090 | 0.0162 | **0.0012** | 0.0377 | 0.0657 | 0.0772 | **0.0146** | 0.1814 | 0.2408 | 0.2880 | **0.0810** |
| FBHarmonics | 0.0104 | 0.0068 | 0.0031 | **0.0005** | 0.0420 | 0.0686 | 0.0525 | **0.0189** | 0.1785 | 0.2526 | 0.1546 | **0.0603** |
| RT Instability (3D) | — | — | 0.2165 | **0.0565** | — | — | 0.3191 | **0.0496** | — | — | 0.9321 | **0.4776** |
| *Inviscid fluids* | | | | | | | | | | | | |
| MultiQuadrantsP | 0.0418 | **0.0142** | 0.0397 | 0.0194 | 0.2010 | **0.0682** | 0.0749 | 0.0720 | 0.6860 | **0.2807** | 0.3839 | 0.3408 |
| MultiQuadrantsO | 0.0432 | 0.0158 | 0.0468 | **0.0112** | 0.3050 | 0.0846 | 0.4801 | **0.0587** | 1.1011 | 0.6478 | 2.7573 | **0.2823** |
| TRL (2D) | 0.1707 | 0.1323 | 0.1601 | **0.0831** | 0.4796 | 0.4117 | 0.5134 | **0.3393** | 0.8879 | **0.8441** | 0.9316 | 0.8648 |
| TRL (3D) | — | — | 0.2238 | **0.1588** | — | — | 0.5753 | **0.5216** | — | — | 0.8106 | **0.7839** |
| *Non-newtonian fluids* | | | | | | | | | | | | |
| Viscoelastics | 0.1030 | 0.0878 | 0.1398 | **0.0295** | 0.1578 | 0.1532 | 0.1989 | **0.0373** | — | — | — | — |

*Table 1.* Losses (Median VRMSE) averaged over various time horizons. One-step computed over all possible steps. Ranges computed by forecasting from initial conditions. Lower is better. Dark font signifies closer to best performance. Bold indicates top performing model. For trajectories shorter than window, averages computed over available steps. "—" indicates either either trajectory is too short or model not applicable (e.g., 2D model on 3D data).

more expensive than 2D or 1D solvers. Two of the datasets we explore, Post Neutron Star Merger (PNS) and Red Supergiant Convective Envelope (RSG) are among the most computationally expensive public simulation datasets to date having taken roughly 1.5M and 10M (Ohana et al., 2025) core hours to generate small numbers of trajectories. Emulation is complicated by log-spherical geometries with unique symmetries and space-time metric tensors. Despite these challenges, we can see that the finetuned WALRUS model is representing large scales well.

**Limitations**. While state-of-the-art from a machine learning perspective, scientific validation remains essential. For RSG, exterior layers develop artifacts despite faithful interior representation. For PNS, bulk dynamics are captured but sensitivity in physical processes yields incorrect estimates of quantities like heavy element production. These results nonetheless demonstrate potential for accurate emulation, motivating further refinement.

### 5.2. Cross-domain Analysis

**Experiment setting.** Having established strong transfer to new tasks, we now evaluate the claim that WALRUS is a cross-domain foundation model by exploring how effectively WALRUS performs across the full breadth of its pretraining data. As WALRUS was exposed to this data in pretraining, we elect to consider comparisons to an upper bound on the performance of prior foundation models on these datasets. We specialize WALRUS to each dataset through an additional 500K samples of finetuning. Baseline models are instead finetuned using the combined number of samples WALRUS has seen from each dataset in the precise orientation and sampling rate used for evaluation. For 2D data, WALRUS was exposed to approximately 4M samples per dataset during pretraining (though at varying sampling rates) and 500K during finetuning. Baselines are finetuned on 4.5M samples, regardless of their pretraining strategy.

**Analysis**. WALRUS achieves top performance with aver-

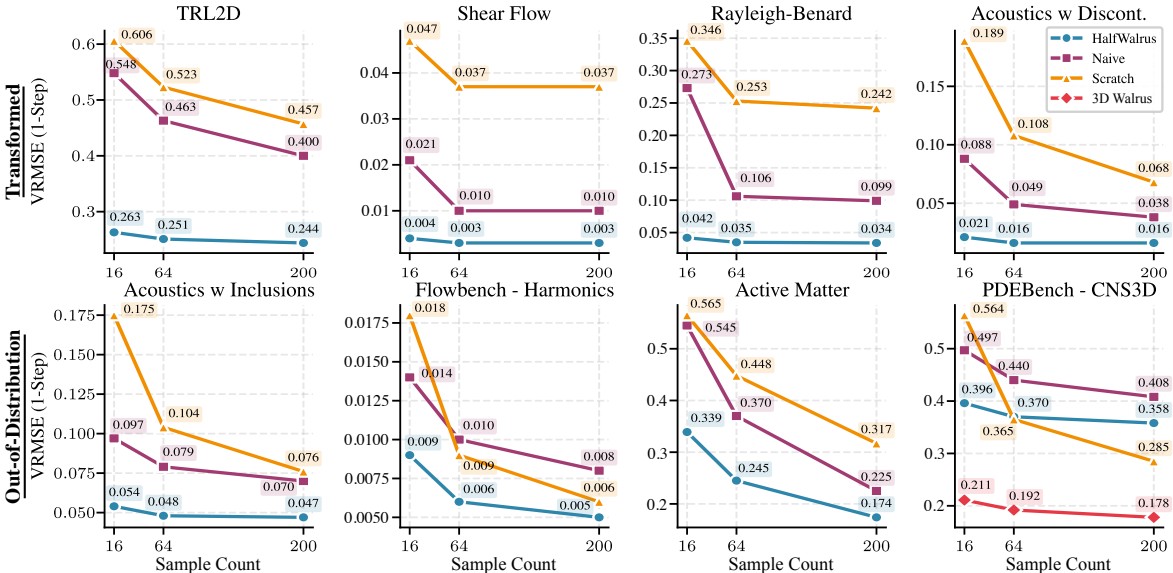

*Figure 5.* One-step VRMSE in limited data scenarios denoted by restricted numbers of samples. (Top) Transformed variants of in-distribution data. (Bottom) Previously unseen datasets. (Bottom-right) CNS3D is the first 3D dataset seen by any half-sized model. Full WALRUS trained with 3D included for comparison with the 2D-pretrained models used in all other panels.

age reductions of 52.2% (one-step, 18/19 tasks), 30.5% (20 steps, 17/19), and 6.3% (20-60 steps, 12/19) in Table 1. However, the variations in performance provide interesting insights into model architecture and training. DPOT's Fourier-based AFNO excels on linear acoustics/wave propagation, even outperforming WALRUS on longer horizons—consistent with numerical Fourier spectral methods' efficiency in smooth linear settings. Poseidon performs competitively on inviscid fluids, particularly MultiQuadrantsP, explained by its pretraining on Euler equations (4/6 tasks, including quadrant variants). Despite these domain-specific strengths, WALRUS's broader pretraining yields superior cross-domain performance.

### 5.3. Impact of Pretraining Strategy

**Experiment setting.** To assess whether diversity-focused pretraining matters beyond scale/architecture, we ablate augmentation strategies using half-size HALFWALRUS (Appendix Table 3) on restricted 2D data (Maze, Discontinuous, Gray-Scott, Rayleigh-Benard, Shear Flow, TRL2D, Staircase, Viscoelastics—chosen for anisotropic behavior). We compare: (1) HALFWALRUS with full augmentation (Section 4.1), (2) Naive 2D-only training, and (3) training from Scratch. All evaluate on transformed pretraining tasks and unseen out-of-distribution tasks using identical configurations.

**Analysis.** The results of this experiment paint a fascinating picture about the role of data diversity in pretraining and downstream performance. Looking only at pretraining metrics in Appendix Figure 7, it would appear that WALRUS's strategy of utilizing heavy augmentation in time and space,

including projecting two-dimensional data into 3D, has an overall negative impact. However, looking at the results for downstream tasks (Figure 5) we see a wildly different story which strongly affirms the need for diversity in pretraining and suggests that the community must be wary of experiments offering strong pretraining results on restricted pretraining collections. While the HALFWALRUS model which uses all augmentations discussed in Section 4.1 is expected to have a significant advantage in the Transformed category as this data falls within the HALFWALRUS training distribution, HALFWALRUS also robustly outperforms training from scratch and the naive pretraining approach on entirely new tasks. Especially notable is the performance of these entirely 2D-pretrained models on the 3D CNS task in the bottom right panel. Having never seen 3D data HALFWALRUS provides a boost over training from scratch in extremely small data regimes. However, this is a small boost and we see that insufficiently broad pretraining (excluding 3D data in this case) can actually become a hindrance as downstream data volumes grow. We include the finetuning performance of full WALRUS which was pretrained on 3D data for comparison. In this case, the 3D-pretrained model has a clear and overwhelming advantage suggesting the importance of including 3D data in pretraining.

## 6. Conclusion

**Limitations.** This work addresses several important, but specific limitations of current foundation models for transportive continuum dynamics - cost-adaptivity, stability, and efficient training on highly heterogeneous training data in their native resolution. Yet other obstacles still remain.

Training on non-uniform geometries while maintaining efficiencies is a natural next exploration direction. Similarly, settling the discrepancy between more expensive history-based in-context learning and faster but less flexible explicitly parameterized modes of operation (Appendix E) will likely require solutions that can interpolate between the two extremes while also handling the more complex case of partially observed or corrupted data. Additionally, WALRUS while having stochastic elements, is trained deterministically with full reconstruction loss. This could prove to be a representational bottleneck for poorly observed or stochastic systems. Diffusion models, particularly latent diffusion models, have shown great promise in ensuring stable rollouts at much higher compression levels (Rozet et al., 2025). These approaches, if generalized to the multiple physics setting without loss of accuracy, could offer significant advantages in run-time due to latent space operations and multi-step predictions while also providing probabilistic estimates.

**Summary**. WALRUS achieves state-of-the-art results (56/65 metrics across 26 tasks) through scaling data and parameter count, novel stabilization techniques, and incorporating adaptive compute. Strong performance stems from both architectural improvements and diversity-first training over 19 scenarios with aggressive augmentation enabled by efficient distribution strategies. Ablations confirm diversity benefits transfer despite higher pretraining loss. Yet despite this performance, more work remains for the development of physics foundation models. The distinct challenges of physical emulation where data, modeling, and computation interact in unique ways require continued advances in data management, distribution, and architectures designed for physical simulation. Close collaboration with domain scientists and numerical software developers will be essential as machine learning tools mature to support high-resolution, geometrically complex data for real scientific challenges.

### Acknowledgments

Polymathic AI gratefully acknowledges funding from the Simons Foundation and Schmidt Sciences, LLC. This work was supported in part by the AI2050 program at Schmidt Sciences (Grant G-25-70028). Dr. Mukhopadhyay thanks the Infosys-Cambridge AI centre for support. The team thanks Sophie Barstein for support with the accompanying blog post for the project. Compute for this project came from multiple sources and the authors would like to thank the Scientific Computing Core, a division of the Flatiron Institute, a division of the Simons Foundation for extensive computational support. Additionally, the results and models reported in this work were substantively aided through compute resources from the National AI Research Resource Pilot, including support from NVIDIA and NVIDIA's DGX Cloud product which includes the NVIDIA AI Enterprise Software Platform. In particular, the team would like to thank Mahidhar Tatineni at SDSC for technical support with NAIRR resources. The team would also like to thank the developers of APEBench (Koehler et al., 2024) for open sourcing the vape4d software used for several 3D visualizations in this paper.

### Impact Statement

This paper concerns the development of computational tools for emulating physical systems. The applications directly touched on in this paper are all purely scientific. We do acknowledge that physical emulation is a core part of CAE workflows used for weapons development, however, applying this work in those spaces would require significant adaptation including novel datasets and geometric adjustments and so the potential negative societal impact is minimal.

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

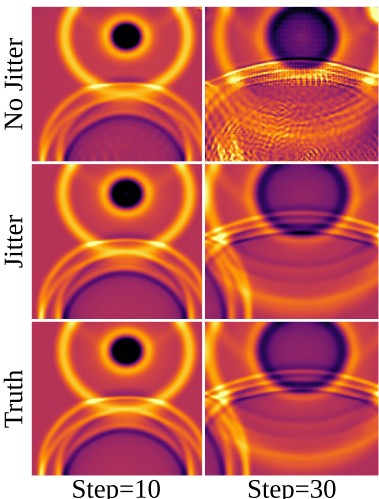

*Figure 6.* Patch jittering (middle) reduces the accumulation of high frequency artifacts (top) allowing for more stable long-term forecasts.

## A. Extended Details and Experiments

### A.1. Patch Jittering

For this analysis, we consider a continuous scalar-valued signal $U(x) \in \mathcal{L}^2(\mathbb{T})$ sampled uniformly as $u \in \mathbb{R}^N$, hidden representation $y \in \mathbb{R}^M$, and output $v \in \mathbb{R}^N$. We denote the downsampling rate by $P = \frac{N}{M}$. We denote the discrete Fourier transform of each of these signals by $\hat{u}$ which we index with $k \in \{0, \ldots, K\}$. We assume that $U(x)$ is a bandlimited signal such that $\hat{U}[k] = 0, \forall |k| > N/2$. By the Shannon-Nyquist sampling theorem, we can therefore represent $u$ exactly by its Fourier expansion $u[x] = \sum_{k \in \mathbb{Z}} \hat{u}[k] e^{-i2\pi kx[k]}$ and $\hat{U}(k) = \hat{u}[k]$. Note that we disregard the scaling factors in the Fourier transform in what follows as these can be arbitrarily absorbed into filter coefficients.

#### A.1.1. PROOF OF LEMMA 3.1

The strided convolution with filter $g$ can be exactly written in the frequency domain as:

$$\hat{y}[k] = \hat{g}[k]\hat{u}[k] + \sum_{j=1}^{P-1} \hat{g}[k + jM]\hat{u}[k + jM] \tag{5}$$

recalling that due to periodicity $\hat{u}[k] = \hat{u}[k + jN]$ and $\hat{y}[k] = \hat{y}[k + jM] \ \forall j \in \mathbb{Z}$, this amounts to the summation of the aliased frequencies at the new resolution after modulation by filter $g$. The "reconstruction" by transposed convolution with filter $h$ is then:

$$\hat{v}[k] = \overline{\hat{h}[k]}\hat{y}[k \mod P] \tag{6}$$

or repetition of the resolvable frequencies at the lower resolution followed by modulation by $h$. Thus, their composition can be simplified to:

$$\hat{v}[k] = \overline{\hat{h}[k]}\hat{g}[k]\hat{u}[k] + \sum_{j=1}^{P-1} \overline{\hat{h}[k]}\hat{g}[k + jM]\hat{u}[k + jM] \tag{7}$$

which given the smoothness constraints of small-kernel convolutions (McCabe et al., 2023b) leads to accumulation around the aliases of high magnitude modes.

#### A.1.2. PROOF OF PROPOSITION 3.1

We can reformulate this procedure probabilistically by randomly translating our input data and subsequently inverting the translation in our output data. Exploiting the Fourier shift property $\mathcal{F}u(x - s)[k] = e^{-i2\pi sk}\hat{u}[k]$, we can write this modified

version of Eq. 7 as:

$$\mathbb{E}_{s\sim\mathbb{T}}\Big[\hat{v}[k]\Big] = \mathbb{E}_{s\sim\mathbb{T}}\left[e^{i2\pi sk}\left[\overline{\hat{h}[k]}\hat{g}[k]\hat{u}[k]e^{-i2\pi sk} + \sum_{j=1}^{P-1}\overline{\hat{h}[k]}\hat{g}[k+jM]\hat{u}[k+jM]e^{-i2\pi s(k+jM)}\right]\right] \tag{8}$$

$$= \overline{\hat{h}[k]}\hat{g}[k]\hat{u}[k] + \sum_{j=1}^{P-1}\mathbb{E}_{s\sim\mathbb{T}}\left[\overline{\hat{h}[k]}\hat{g}[k+jM]\hat{u}[k+jM]e^{-i2\pi sjM}\right] \tag{9}$$

$$= \overline{\hat{h}[k]}\hat{g}[k]\hat{u}[k] + \sum_{j=1}^{P-1}\left[\overline{\hat{h}[k]}\hat{g}[k+jM]\hat{u}[k+jM]\int_0^1 e^{-i2\pi sjM}ds\right] \tag{10}$$

where each integral evaluates to 0 as $jM \neq 0$. This leaves us with:

$$\mathbb{E}_{s\sim\mathbb{T}}\Big[\hat{v}[k]\Big] = \overline{\hat{h}[k]}\hat{g}[k]\hat{u}[k] \tag{11}$$

or that the expectation of this randomized process is the un-aliased solution. In practice, this is proportional rather than exact due to the normalization factors in the DFT, though these can be absorbed by the filter coefficients without loss of generality.

In realistic settings, as this amounts to a simple Monte Carlo averaging procedure, the variance of this estimator has a slow $O(1/T)$ convergence rate (Caflisch, 1998). While this could likely be accelerated by a structured quasi-Monte Carlo method, the presence of nonlinearity and boundary conditions make any theoretical insights more instructive and less exact so we do not explore such approaches in this work. As such, in the next section, we turn to empirical exploration of sampling without averaging where we observe significant benefits at minimal cost.

### A.1.3. JITTERING ABLATION

Here we sample twenty validation trajectories for each pretraining dataset for the pretrained WALRUS model and compare the median VRMSE over full length autoregressive rollouts. We can see that the use of jitter shows a clear improvement in long term accuracy. The median improvement is 54% and the number of "high" long term errors (defined as median VRMSE $\geq 1$) shrinks from 10 to 3. Overall, we see improvement on 16/19 pretraining datasets (while we did observe improvement on MHD as well, this is outside of the range we consider reasonable prediction performance, so we do not include it in the numerator) with the few that do not improve being largely unaffected.

Jitter is primarily intended to aid with the stability of long-term rollout accuracy, particularly preventing divergence to values outside of the typical distribution of field values. Analyzing improvement through $L^2$ analysis can be tricky as the mean is sensitive to outliers. In Table 2, we show the median trajectory-averaged VRMSE for each dataset after sampling 32 trajectories. Here we can see that jitter shows a clear improvement in long term accuracy. The median accuracy improvement is 54% and the number of "high" long term errors (defined as median VRMSE $\geq 1$) shrinks from 10 to 3. This leads to an overall improvement on 89% of datasets with the few that do not improve being largely unaffected. An example of the type of improvement seen in practice is shown in Figure 6.

### A.2. Topology-aware Sampling Strategy

Uniform sampling of heterogeneous data can lead to training bottlenecks, especially in high dimensional spaces. Patch-based tokenization can produce large token counts in high dimensions. For example, a $256^3$ input processed with an effective patch size of 16 results in 4096 tokens per frame. Processing as few as three frames therefore requires sequence lengths of around $12K$ - larger than the initial pretraining window used in modern LLMs like LLaMA 3 (Grattafiori et al., 2024). This regime favors strategies like FSDP which communicates parameters across the network over over model parallelism which shares activations. Nonetheless, FSDP can lead to inefficiencies when training load is not balanced across the FSDP group.

PyTorch's FSDP utilizes synchronous collective communication primitives such as `AllGather` during the forward pass itself. These calls are blocking and require each rank to reach a given step before the communication operation can be completed. If workloads are not balanced within the sharding or FSDP group, the ranks running faster jobs will be sitting idle until the slower jobs catch up. Many foundation models for spatiotemporal dynamics to date have avoided this complication of load balancing by using only data at a single resolution and dimensionality. This, however, is not reflective of practical usage conditions and so it is vital to develop strategies for this dilemma.

| Dataset | Trajectory Lengths | Jitter | No Jitter |
|---|---|---|---|
| Maze | 202 | 0.34 | 1.07 |
| Inclusions | 102 | 0.18 | 1.30 |
| Discontinuous | 102 | 0.18 | 1.44 |
| Staircase | 50 | 0.01 | 0.27 |
| Active Matter | 81 | 0.98 | 0.98 |
| Gray-Scott | 1001 | 0.41 | 0.48 |
| Supernova (3D) | 59 | 1.20 | 1.28 |
| TGC (3D) | 50 | 0.67 | 1.12 |
| PlanetSWE | 1008 | 0.10 | 0.56 |
| MHD (3D) | 100 | $\geq 10$ | $\geq 10$ |
| Rayleigh-Benard | 200 | 0.81 | 0.84 |
| Shear Flow | 200 | 0.16 | 0.27 |
| FBHarmonics | 242 | 0.07 | 0.27 |
| RT Instability (3D) | 120 | 7.11 | 7.52 |
| MultiQuadrantsP | 101 | 0.62 | 0.60 |
| MultiQuadrantsO | 101 | 0.50 | 1.82 |
| TRL (2D) | 101 | 0.75 | 76.9 |
| TRL (3D) | 101 | 0.92 | 1.17 |
| Viscoelastics | 20* | 0.10 | 0.56 |

*Table 2.* Median validation full-trajectory averaged VRMSE at various time horizons for both jittered and un-jittered WALRUS. ($*$) denotes that for variable length trajectories, shortest length used as standard.

We address this issue in several ways. First, through *adaptive-compute tokenization* as discussed in Section 3.1. This ensures that at a given dimensionality, all datasets are converted to roughly the same number of tokens. For 2D data, we select this size as 32 per dimension without padding. For 3D, this is set to 16 without padding. Note that in Figure 3 this initially decreases throughput as, on average, this is increasing the total spatial tokens and thus increasing cost. The second is through the use of *differential batch sizes*. Linear projections, the dominant cost in the model, scale linearly with token count so to balance the tokens between 2D and 3D, we multiply the 2D batch size and length of time history by a factor of 2 each.

However, these steps alone are insufficient to reduce all discrepancies in workload. Encoder/decoder costs still vary by input size, though this can be partially accounted for by simply replicating these parameters on each device rather than sharding them as the hMLPs used are lightweight. Within the transform, while both 2D and 3D remain in the MLP-dominated regime, in moving from 2D to 3D, the larger spatial attention context results in a shift from linear layers forming 95% of the cost of a block to 80% which forms a significant growth in time spent in the attention operation.

Ideally, we would like each GPU to sample datasets independently to maximize batch diversity, however, these cost discrepancies can result in significant deadweight loss. As a compromise, we employ a sampling strategy designed for HSDP (Zhao et al., 2023) where all ranks within a sharding group are forced to sample from the same dataset every step. Each group, however, samples independently so that the total batch consists of many datasets. Combining this with the gradient accumulation as randomized load balancing trick used in (McCabe et al., 2023a), we have a system where the groups of nodes where `AllGather` operations act as a bottleneck are forced to sample data of the same resolution and dimensionality while the overall variance of time-per-step is reduced by averaging over multiple micro-batches between `AllReduce` operations prior to parameter updates, balancing sampling diversity and computational performance.

We can see the iterative impact of these changes in Figure 3. Under the given sampling scheme, these combined changes result in an increased throughput of 262% compared to naive usage of FSDP.

---

**Algorithm 1** Node Topology-aware Sampling for HSDP (single GPU view of one optimizer step)

---

**Require:** Datasets $\{D_1, \ldots, D_K\}$ with weights $\{w_k\}$ (held uniform in WALRUS); sharding group id $g$; gradient accumulation steps $M$; micro-batch size $B$; model parameters $\theta$, step $t$.

1: Zero local gradient accumulator: $G \leftarrow 0$
2: **for** $m = 1$ to $M$ **do**
3: $\quad k \leftarrow \text{WeightedSample}(\{w_k\}; \text{seed}: (t + m, g))$   (*shared across ranks in group $g$; independent across groups*)
4: $\quad b \leftarrow \text{DrawTrajectories}(D_k, B)$   (*independent per rank*)
5: $\quad$ Forward pass on $b$   (*blocking `AllGather` of $\theta$ within group $g$*)
6: $\quad$ Backward pass on $b$   (*blocking `AllGather` of $\theta$ within group $g$*)
7: $\quad G \leftarrow G + \nabla_\theta \mathcal{L}(b)$   (*local accumulation; no communication*)
8: **end for**
9: $G \leftarrow \text{AllReduce}(G)$   (*across data-parallel groups*)
10: $\theta \leftarrow \text{OptimizerStep}(\theta, G/M)$

---

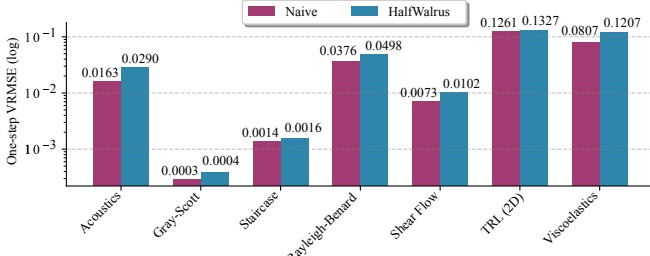

*Figure 7.* Pretraining VRMSE from HALFWALRUS and naively trained models. From pretraining alone, it appears that the less diverse "naive" strategy outperforms HALFWALRUS, but this trend is reversed on downstream tasks in Figure 5.

### A.3. Pretraining HALFWALRUS Metrics (Exp 5.3)

Figure 7 and Figure 5 show the misleading nature of evaluating pretraining performance for foundation models. While the pretraining metrics in Figure 7 suggest that heavier augmentation and 2D-3D representations have hurt model performance, this effect is limited to the pretraining setting. In downstream evaluations, Figure 5 shows that the diversity-first view of pretraining has actually resulted in significantly improved transfer performance even when evaluating within the scope of a single architecture and model shape.

## B. Model Details

WALRUS can be broken into the following stages shown in order in Figure 8:

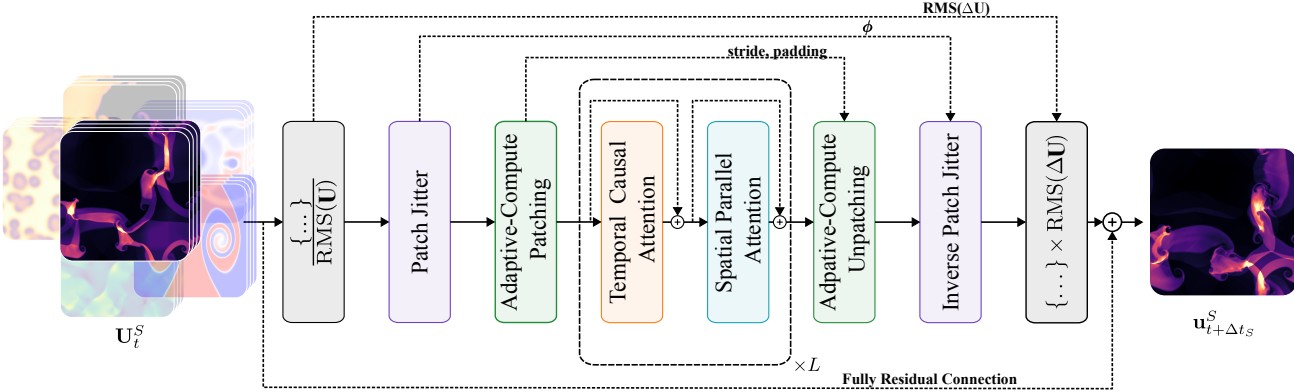

*Figure 8.* Stages of the WALRUS model. WALRUS uses recently developed adaptive-compute methods to balance cost between datasets on top of a modern transformer backbone and novel stabilization approaches.

- Normalization/De-Normalization

- Patch Jittering/Inverse Jittering

- Adaptive-Compute Patching/Reconstruction

- Split Space-time attention block

    - PaLM-parallel Attention-MLP (Space)
    - Vanilla Causal Attention (Time)

We will discuss each briefly in it's own section, though the high level sizing information can be found in Table 3. Normalization is performed outside of the model itself. Jittering and adaptive compute patching are components of the *encoder* and *decoder* blocks. The attention blocks are performed at constant resolution as part of the *processor* blocks.

*Table 3.* Model details for Walrus and HalfWalrus.

|  | **Walrus** | **HalfWalrus** |
| --- | --- | --- |
| Architecture | Space-time Split Transformer | Space-time Split Transformer |
| Parameters | $1.3 \times 10^9$ | $6.4 \times 10^8$ |
| Encoder | hMLP+StrideMod | hMLP+StrideMod |
| Time History (2D) | 6 | 6 |
| Base Token Shape (2D) | $32^2$ | $32^2$ |
| Time History (3D) | 3 | 3 |
| Base Token Shape (3D) | $16^3$ | $16^3$ |
| Projection dimension | 48 | 48 |
| Encoder dimension | 352 | 352 |
| Hidden dimension | 1408 | 1088 |
| MLP dimension | 5632 | 4352 |
| Space block | Parallel Attention | Parallel Attention |
| Space positional embedding | AxialRoPE | AxialRoPE |
| Time block | Causal Attention | Causal Attention |
| Time positional embedding | LearnedRPE | LearnedRPE |
| Attention heads | 16 | 16 |
| Activation | SwiGLU | SwiGLU |
| Normalization | RMSGroupNorm | RMSGroupNorm |
| Normalization Groups | 16 | 16 |
| Blocks | 40 | 30 |
| Drop Path | 0.05 | 0.05 |
| Optimizer | AdamW | AdamW |
| Learning rate | $2E-4$ | $1E-4$ |
| Weight decay | 1E-4 | 1E-4 |
| Warm-up Epochs | 10 | 10 |
| Cool-down Epochs | 10 | 10 |
| Scheduler | InvSqrt | InvSqrt |
| Gradient norm clipping | 10.0 | 10.0 |
| Micro-batch size per GPU (2D) | 2 | 2 |
| Micro-batch size per GPU (3D) | 1 | 1 |
| Micro-batch per epoch | 2000 | 2000 |
| GradAcc Steps | 4 | 4 |
| Epochs | 200 | 200 |
| GPUs | 96 | 8 |

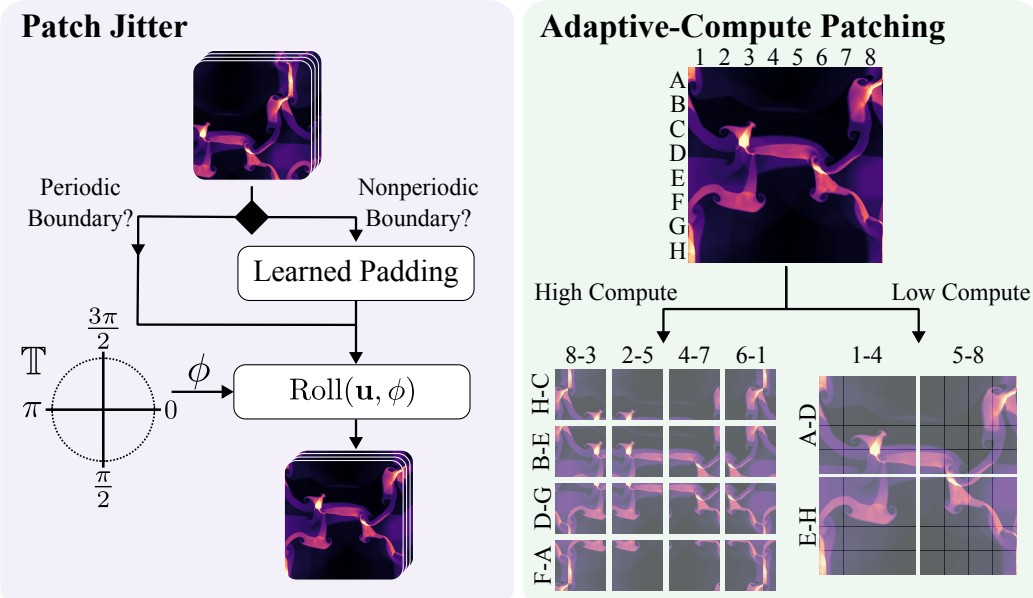

*Figure 9.* Stages of the WALRUS model. WALRUS uses recently developed adaptive-compute methods to balance cost between datasets on top of a modern transformer backbone and novel stabilization approaches.

### B.1. Reversible Normalization

During pretraining, we use per-trajectory normalization (McCabe et al., 2023a) as the pretraining process is intended to emulate data from a wide data pool where there may not be a sufficiently large number of trajectories to precompute statistics. However, given the nonlinear dynamics, scale can be a significant factor. During finetuning, therefore, when the model is assumed to have access to a small finetuning set, it can be advantageous to employ normalization computed over the full dataset.

As mentioned in Section 3, the normalization is both asymmetric and uses the RMS statistics rather than mean/standard deviation. This is largely to avoid drift due to the de-normalization of the step term. Given an input trajectory $\boldsymbol{U} \in \mathbb{R}^{T \times B \times C \times H \times W \times D}$ and using `torch`-style tensor conventions, this can be written as:

$$\mathrm{RMS}(\boldsymbol{U})_{:,b,c} = \sqrt{\frac{1}{T \times H \times W \times D} \sum_{(t,h,w,d) \in T,H,W,D} \boldsymbol{U}_{t,b,c,h,w,d}^2} \tag{12}$$

letting $\mathcal{M}$ represent the model, the full procedure can then be written as:

$$u_{t+1} = u_t + \mathcal{M}(\frac{\boldsymbol{U}_t}{RMS(\boldsymbol{U}_t)}) * RMS(\Delta\boldsymbol{U}_t) \tag{13}$$

### B.2. Patch Jitterer with Adaptive Compute Patching

This section focuses on concrete implementation while the theory of patch jittering is covered in Section A.1. As both jittering and adaptive patching (Figure 9) require controlled padding, these are tightly coupled in the WALRUS implementation which also depends on the boundary topology as described in Section D.2.

**hMLP** The lightweight encoder module used here is the hierarchical MLP (hMLP) stem of Touvron et al. (2022) which replaces ViT's single linear patch projection with a sequence of patchify-and-project stages interspersed with nonlinearies and normalization layers. Given input $\mathbf{x} \in \mathbb{R}^{C \times H \times W}$, tokens are produced by two successive convolutional downsampling layers with kernel sizes $(p_1, p_2)$ and strides $(s_1, s_2)$, each followed by normalization and a GELU nonlinearity:

$$\mathbf{h} = \mathrm{GELU}(\mathrm{N}(W_1 *_{s_1,p_1} \mathbf{x})),$$
$$\mathbf{z} = \mathrm{N}(W_2 *_{s_2,p_2} \mathbf{h}),$$

---

**Algorithm 2** Patch Jittering with Adaptive-Compute Patching

---

**Require:** Data $u$, $p_1, p_2$ encoder convolutional kernel sizes, $s_1, s_2$ encoder convolutional kernel strides, $b$ boundary identifiers.

1: $p_{eff} \leftarrow p_1 + s_1 \times (p_2 - 1)$
2: $s_{eff} \leftarrow s_1 \times s_2$
3: $pad_{stride} \leftarrow p_{eff} - s_{eff}$
4: **if** $b =$ "PERIODIC" **then**
5:    $pad_{total} \leftarrow pad_{stride}$
6: **else**
7:    $pad_{total} \leftarrow \text{int}(s_{eff}/2) + pad_{stride}$
8:    $u \leftarrow \text{ZeroPad}(u, \text{left}=pad_{total}, \text{right}=pad_{total}, \text{channel}=3)$
9:    $u[\text{padding region}, \text{channel}=b] \leftarrow 1.0$
10: **end if**
11: $j \leftarrow \text{randint}(0, pad_{total})$
12: $u \leftarrow \text{roll}(u, j)$
13: **if** $b =$ "PERIODIC" **then**
14:    $u \leftarrow \text{CircularPad}(u, \text{left}=\text{int}(s_{eff}/2), \text{right}=\text{int}(s_{eff}/2))$
15:    $u \leftarrow \text{ZeroPad}(u, \text{channel}=3)$
16: **end if**
17: **return** $u$

---

where $*_{s,p}$ denotes a stride-$s$, kernel-$p$ convolution and N is the normalization stage. In (Touvron et al., 2022), $p_i = s_i$ at each stage to ensure non-overlapping receptive fields, so each output token depends on a disjoint region of the input. However, with the adaptive compute procedure described below, the encoder essentially just becomes an extremely lightweight CNN.

**Adaptive-Compute Patching**. Our implementation is closely adapted from Mukhopadhyay et al. (2026) with the exception that we inherently try to equalize tokens seen per dataset during pretraining. To do this, we choose the kernel sizes $(p_1, p_2)$ and strides $(s_1, s_2)$ used in the convolutional downsampling layers to ensure the internal model resolution lines up with the settings in Table 3. Padding is then implemented based on the effective kernel and stride of the combined convolution operations as any intermediate normalization and nonlinear operations do not impact domain shape.

**Patch Jitter.** For periodic boundaries, jitter is implemented as a straightforward `torch.roll` operation that occurs before any padding is applied. However, in the case of non-periodic boundaries, jitter is instead implemented through padding and limited size `roll` operation. Due to the inherent translation equivariance of the convolutions used in patching, we can limit the size of the roll to the effective size of the convolutional kernel. Here the padding is chosen to minimize the additional interior tokens generated. The combined procedure is described in Algorithm 2.

After this padding stage, the encoder can be applied with the kernel and stride parameters without additional modification. subsequently to reconstruct the fields in grid space, the reverse procedure is followed: transposed convolutions using the same $p, s$ values perform upsampling, the jitter is inverted, and padded cells are cropped. The only complication that emerges is again the case of periodic boundaries where the domain is padded by $(k-s)//2$ on each side prior to performing the transposed convolution and subsequently truncated by $k - s$ to maintain periodicity.

### B.3. Processor Blocks

Each internal block of the transformer architecture consists of both a space and time mixing layer. These are performed sequentially and a high level visual representation of these blocks can be seen in Figure 10.

**Space.** Space and channel mixing are implemented using parallel attention (Wang, 2021) which implements the MLP and Attention in parallel rather than sequentially:

$$v' = \text{RMSGroupNorm}(u)$$
$$v = u + \text{MLP}(v') + \text{Attention}(v')$$

The Attention operation here is scaled dot product attention computed over the spatial domain such for a given time, all tokens in space attend to each other. Training stability is improved using QKNorm from (Dehghani et al., 2023). Position

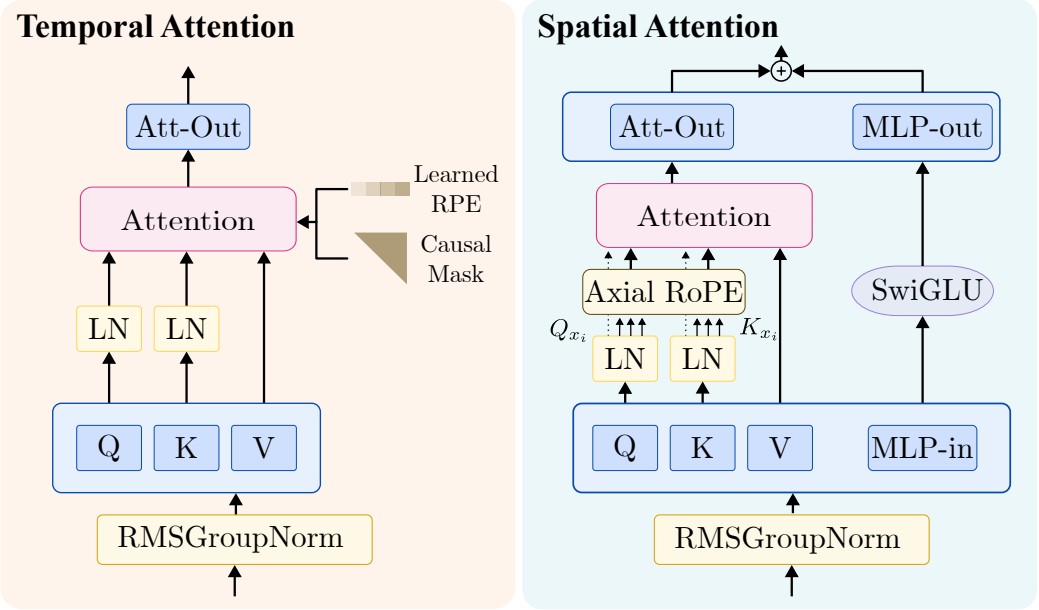

*Figure 10.* Stages of the WALRUS model. WALRUS uses recently developed adaptive-compute methods to balance cost between datasets on top of a modern transformer backbone and novel stabilization approaches.

encoding is performed using axial RoPE (Lu et al., 2023) interpolated onto different sized grids.

**Time.** In time, we use a standard attention implementation with causal masking applied independently to each spatial location such that for each point in space, this operation is causally masked attention over all time points. Time blocks use T5-style (Raffel et al., 2020) relative position encoding. While the structure of the objective is inherently causal and therefore does not require causal masking, we saw little difference in performance between the two approaches. Given the significant cost of utilizing the extra history tokens, we opted to use causal masking to enable KV caching in order to reduce inference costs for autoregressive forecasting. This is not currently implemented in the released code, but is vital for production deployment.

Note that while modifications exist to ensure RoPE is periodic, the conventional implementation is not. Therefore, to avoid boundary effects along periodic borders, we perform further random periodic rolls between each block. Rather than inverting immediately after, we use the group property of periodic translation and track the total magnitude of the roll during execution of the blocks. After all processor blocks have completed, we invert the full roll at that time.

### B.4. Misc.

In a large scale project, with the expense of evaluation and the known behavioral differences across scales, sometimes architectural decisions were made more from inertia, having worked well when originally implemented, rather than through careful testing. Here, we describe the cases where this occurred to bring to light areas that could benefit from further experimentation:

1. Encoder and processor using different activations - The encoder layers all use GELU activations (Hendrycks & Gimpel, 2016) while the processor layers use SwiGLU. This was an artifact of merging several implementations rather than an intentional decision. On small scale experiments, we saw little difference from correcting this, so it was left as is.

2. Additive corner handling - In the boundary padding examples, we discussed the one-dimensional case where boundary topology is denoted by appended one-hot encoded channels. However, in multiple dimensions, the padding will overlap in corners resulting in additive corner handling. Changing this to a cross-product of boundary tokens was not explored.

# C. Training Details

## C.1. Pretraining

WALRUS was pretrained using the 19 datasets in Table 6. Training was performed using the Adam optimizer (Kingma & Ba, 2014) with decoupled weight decay (Loshchilov & Hutter, 2017) and an inverse square root learning rate schedule with linear warmup (Zhai et al., 2022) and square root cooldown (Hägele et al., 2024). The base learning rate used was $2E - 4$ for WALRUS and $1E - 4$ for `HalfWalrus`. As the dataset is too large for many true epochs, we instead structure it in terms of logging intervals of 2000 micro-batches which we refer to as epochs for convenience. WALRUS was trained for 200 of these "epochs" over 96 NVIDIA H100 GPUs. This amounts to 38M micro-batches seen during pretraining. Given the differential batch sizes, this works out to about 4M examples per datasets for 2D data and 2M per dataset for 3D data. `HalfWalrus` was trained for 200 over 8 NVIDIA H100 GPUs. All training was performed in FP32 single precision as we observed significant accuracy degradation using BF16 while we observed training stability issues using FP16.

**Augmentations.** During pretraining, to maximize the diversity seen, we employ several forms of augmentation. These augmentations are chosen because they are physically valid for a significant number of datasets.

1. **Variable time-stride**: We sample trajectories spaced by varying $\Delta t$ where this ranges from 1 to 5 simulation steps. The model is forced to learn to infer the time step from history. This information is not provided explicitly.

2. **Tensor-aware rotations**: For all data defined on Euclidean domains, we perform additional augmentation using rotations sampled from the axis-aligned subset of SO(3). As mentioned in the main text, all geometric transformations are performed both on the spatial layout and on tensor-valued fields to ensure physical consistency.

For non-WALRUS models in experiments, pretrained weights were used from the public code released by the authors.

## C.2. WALRUS Finetuning

For all finetuning tasks, WALRUS was trained for an additional 50 pseudo-epochs (500K samples) on the new dataset exclusively at the time-striding and frame of reference used for evaluation. This was performed using the same optimizer and learning rate structure as pretraining but at half the original base learning rate. Finetuning was performed on one node with 4 NVIDIA H100 GPUs at single precision for both WALRUS and baseline models.

### C.2.1. ARCHITECTURAL CHANGES FOR FINETUNING

The following are architectural modifications we enable during finetuning to allow models to adapt to anisotropy in the problem domain.

**Learnable RoPE.** During finetuning, we replicate the RoPE frequency parameters in each dimension and make these parameters trainable. The model can therefore learn to bias the attention to different positions along each axis and account for any differences in the spatial discretization.

**Adding APE.** Models developed for complex domains often employ forms of learned absolute position embeddings. Analysis of these embeddings can reveal complex structure emerging in order to account for information that is not explicitly provided to the model. For instance, visualizing the channels in the position embeddings of a weather model might reveal a surface that appears like a low resolution map. The ability to learn these features can be invaluable for effective model performance. During finetuning, we therefore initialize a learned APE embedding layer.

As the RoPE adaptation has little downside, we enable it for all finetuning experiments. For APE, we use it only when the domain is anisotropic. Table 4 shows that the effect of APE is not universally beneficial. On both Euler–Periodic and Euler–Open datasets, models trained without APE consistently achieve lower next-step and rollout VRMSEs across almost all horizons.

In translationally invariant systems with little geometric forcing such as the Euler examples, the governing equations do not depend on absolute position, and injecting positional signals can introduce spurious biases. In this case, APE effectively adds artificial energy at every rollout step: when applied autoregressively it accumulates errors and degrades long-term forecasts. By contrast, in systems with fixed spatial heterogeneities—for example Rayleigh–Bénard convection, where unknown boundary forcing occurs at the top and bottom walls, or interface-dominated problems where the interface remains at a fixed location—APE provides crucial anchoring information that the model would otherwise lack. Thus, APE is extremely useful

*Table 4.* Effect of Absolute Positional Encodings (APE) on Euler datasets. Metrics are rollout VRMSE (mean ± std). Lower is better. Removing APE consistently improves performance.

| Dataset | APE | 1:3 | 3:9 | 9:27 | 27:83 |
|---------|-----|-----|-----|------|-------|
| Euler–Periodic | With APE | 0.0330 ± 0.0180 | 0.0734 ± 0.0336 | 0.287 ± 0.0859 | 0.921 ± 0.201 |
|  | No APE | **0.0316 ± 0.0171** | **0.0571 ± 0.0335** | **0.167 ± 0.0970** | **0.530 ± 0.234** |
| Euler–Open | With APE | 0.0289 ± 0.0132 | 0.05548 ± 0.0256 | 0.1665 ± 0.0843 | 0.6448 ± 0.4904 |
|  | No APE | **0.02588 ± 0.01145** | **0.04718 ± 0.02205** | **0.13503 ± 0.06864** | **0.53181 ± 0.3504** |

when it compensates for missing or unobserved conditions, but can hinder performance when absolute position is physically irrelevant.

## C.3. Finetuning Details for Baselines

In all experiments, baselines have been provided equivalent data volumes to the WALRUS model (500K for downstream tasks, 4.5M for 2D data included in pretraining, 2.5M for 3D data included in pretraining). How this data is grouped into batches was determined by hyperparameter search. For each downstream task, we selected a subset of datasets consisting of:

- euler_quadrants_periodic

- acoustic_scattering_discontinuous

- rayleigh_benard

- active_matter

The models were then trained at size B at a grid search of the following parameters

- learning rate - [1e-3, 5e-4, 1e-4, 5e-4, 1e-5]

- maximum batch size - [4, 8, 16, 32, 64]

where the number of total training steps was then computed from the data budget and batch size. For higher resolution systems, maximum batch size was decreased in power of two increments until the batch did not run out of memory. The best performing single hyperparameter set across all runs was then evaluated on the data subset at the full size model (L/H). If at full size, any of these runs diverged or performed worse than the smaller model on any datasets, the learning rate was decreased by 20% until this performance gap was eliminated.

Apart from the base learning rate, optimizer and scheduler details we selected to match WALRUS settings of AdamW with weight decay of $1E - 4$, gradient clipped to norm 10, and inverse square-root schedule with linear warmup and inverse square decay.

### C.3.1. MODEL SIZES

|  | Walrus | MPP AViT-L | Poseidon-L | DPOT-H |
|--|--------|------------|------------|--------|
| **# of parameters** | 1.29 B | 407 M | 628 M | 1.15 B |
| **Base operation (Space)** | Full Attention | Axial Attention | SWIN Attention | AFNO |
| **Base operation (Time)** | Causal Attention | Acausal Attention | None | Temporal-aggregation |

*Table 5.* Comparison of model sizes and core architectures between Walrus and baseline models.

### C.3.2. DPOT-H

While generally DPOT is configured to match pretraining settings, DPOT's public config files did not appear to use data normalization, we empirically found the model generally diverged when trained on unnormalized data as the training data covers a wide range of scales that may not have been seen during pretraining. We therefore used standard mean-std style

dataset normalization with DPOT with normalization computed over the full training set per individual dataset which we hereafter refer to as "global" normalization.

DPOT was trained using the normalized $L^2$ loss used in the source repository with a context length of 10 and at each step, the full field is predicted, ie $\boldsymbol{u}_{t+1} = \mathcal{M}(U_t)$. To adapt DPOT to arbitrary shapes, we employed linear interpolation on the absolute position embeddings and re-initialized embedding weights for new fields. As the DPOT architecture apart from APE are resolution-agnostic, this amounts to changing how several sizes were passed.

Hyperparameters selected for DPOT-H were learning rate $8E - 5$ and at maximum batch size 64.

### C.3.3. POSEIDON

Poseidon settings were configured to best match original training settings. Data was normalized using global dataset statistics and a normalized $L^1$ loss was used. At each step, the full field is predicted, ie $\boldsymbol{u}_{t+1} = \mathcal{M}(U_t)$.

Though there are no inherent architectural limitations, Poseidon's public code had several limitations preventing application to non-square grids. It was necessary to update several model components which used fixed $h = w$ to track $h, w$ separately and accordingly perform all resize operations to respect the rectangular shape. These were done in minimally invasive fashion without need for further training such that the model performance was found to be unchanged on square data up to numerical precision.

Empirically, we found that time-conditioned rollouts worked poorly on more complex dynamics and subsequently trained all reported models in autoregressive fashion. The "time-conditioning" fed into the model was kept at $\Delta t = .05$ to match the standard gap between timesteps in Poseidon's pretraining data so that the initial timestep is within the training distribution. As this approach was only used for finetuning, the time-gap observed is always constant and the model can adapt to the true time scale through finetuning.

Hyperparameters selected for Poseidon-L were learning rate $1E - 4$ and at maximum batch size 64.

### C.3.4. MPP AvIT-L

MPP settings were configured to best match original training settings. Per-trajectory normalization is used within the model, though the loss is computed in globally normalized space. At each step, the full field is predicted, ie $\boldsymbol{u}_{t+1} = \mathcal{M}(U_t)$.

The longer context length (16) used by MPP AvIT-L during pretraining was a significant cost during finetuning, but we found we were able to reduce this to the context used by WALRUS (6) with minimal loss of accuracy during finetuning. As metrics had already been computed for other models by the time this became apparent, the first prediction step was kept at T=17 for 2D systems for consistency. Field-specific embedding layers were re-initialized due to new fields that were not encountered during pretraining.

Hyperparameters selected for MPP-AViT-L were learning rate $1E - 4$ and at maximum batch size 32, though the longer context of MPP generally meant that batch size was more memory dependent.

## D. Data

All data used for pretraining was sourced from TheWell (Ohana et al., 2025) or Flowbench (Tali et al., 2024). TheWell contains diverse physical settings from across multiple domains while Flowbench fills one of the gaps in TheWell with an extensive collection of flow past obstacle examples. The full descriptions of each of these datasets and how they were generated can be found in the source documents.

For evaluation of transfer capabilities, we include additional datasets from the additional sources PDEGym (Herde et al., 2024), PDEBench (Takamoto et al., 2022), BubbleML 2.0 (Hassan et al., 2026), and PDEArena (Gupta & Brandstetter, 2023) to reduce the likelihood of results being impacted by subtle biases in the construction of the source datasets.

When available, we use the canonical train/valid/test splits provided by the development teams of these datasets. We break the data usage into two categories:

1. Pretraining data (Table 6) - Data used during pretraining WALRUS.

2. Downstream tasks (Table 7) - Data used for finetuning task which had never been seen by WALRUS during initial

| Dataset | Short Name | Source | CS | Resolution (pixels) | n_steps | n_traj |
|---|---|---|---|---|---|---|
| acoustic_scattering_discontinuous | Discontinuous | The Well | $(x, y)$ | $256 \times 256$ | 102 | 2 000 |
| acoustic_scattering_inclusions | Inclusions | The Well | $(x, y)$ | $256 \times 256$ | 102 | 4 000 |
| acoustic_scattering_maze | Maze | The Well | $(x, y)$ | $256 \times 256$ | 202 | 2 000 |
| active_matter | Active Matter | The Well | $(x, y)$ | $256 \times 256$ | 81 | 360 |
| euler_multiquadrants_periodicBC | MultiQuadrantsP | The Well | $(x, y)$ | $512 \times 512$ | 101 | 5 000 |
| euler_multiquadrants_openBC | MultiQuadrantsO | The Well | $(x, y)$ | $512 \times 512$ | 101 | 5 000 |
| gray_scott_reaction_diffusion | Gray-Scott | The Well | $(x, y)$ | $128 \times 128$ | 1 001 | 1 200 |
| helmholtz_staircase | Staircase | The Well | $(x, y)$ | $1 024 \times 256$ | 50 | 512 |
| MHD | MHD (3D) | The Well | $(x, y, z)$ | $64^3$ | 100 | 100 |
| planetswe | PlanetSWE | The Well | $(\theta, \phi)$ | $256 \times 512$ | 1 008 | 120 |
| rayleigh_benard | Rayleigh-Benard | The Well | $(x, y)$ | $512 \times 128$ | 200 | 1 750 |
| rayleigh_taylor_instability | RT Instability (3D) | The Well | $(x, y, z)$ | $128 \times 128 \times 128$ | 120 | 45 |
| shear_flow | Shear Flow | The Well | $(x, y)$ | $256 \times 512$ | 200 | 1 120 |
| supernova_explosion | Supernova (3D) | The Well | $(x, y, z)$ | $128^3$ | 59 | 1 000 |
| turbulence_gravity_cooling | TGC (3D) | The Well | $(x, y, z)$ | $64 \times 64 \times 64$ | 50 | 2 700 |
| turbulent_radiative_layer_2D | TRL (2D) | The Well | $(x, y)$ | $128 \times 384$ | 101 | 90 |
| turbulent_radiative_layer_3D | TRL (3D) | The Well | $(x, y, z)$ | $128 \times 128 \times 256$ | 101 | 90 |
| viscoelastic_instability | Viscoelastics | The Well | $(x, y)$ | $512 \times 512$ | variable | 260 |
| FPOHarmonics | FBHarmonics | The Well | $(x, y)$ | $512 \times 128$ | 242 | 400* |

*Table 6.* Pretraining dataset descriptions: coordinate system (CS), resolution of snapshots, n_steps (number of time-steps per trajectory), n_traj (total number of trajectories in the dataset). Step count may differ from other sources as we normalized step count to always include initial conditions. (∗) Removed flowbench_FPO_NS_2D_512x128_harmonics_Re614.hdf5 due to outlier trajectory.

| Dataset | Source | CS | Resolution (pixels) | n_steps | n_traj |
|---|---|---|---|---|---|
| FPOSkelenton | Flowbench | $(x, y)$ | $512 \times 128$ | 242 | 262 |
| PoolBoil Subcooled | BubbleML 2.0 | $(x, y)$ | $512 \times 512$ | 2001 | 44 |
| Conditioned Incompressible NS | PDEArena | $(x, y)$ | $128 \times 128$ | 56 | 6816 |
| CE-RM | PDEGym | $(x, y)$ | $128 \times 128$ | 21 | 1260 |
| CNS Turbulent | PDEBench | $(x, y, z)$ | $64^3$ | 21 | 600 |
| CNS Random | PDEBench | $(x, y, z)$ | $128^3$ | 21 | 100 |
| post_neutron_star_merger | The Well | $(\log r, \theta, \phi)$ | $192 \times 128 \times 66$ | 181 | 8 |
| convective_envelope_rsg | The Well | $(r, \theta, \phi)$ | $256 \times 128 \times 256$ | 100 | 29 |

*Table 7.* Pretraining dataset descriptions: coordinate system (CS), resolution of snapshots, n_steps (number of time-steps per trajectory), n_traj (total number of trajectories in the dataset).

pretraining, though Conditioned Incompressible NS from PDEArena was used during DPOT training and CE-RM from PDEGym was part of the same data collection as Poseidon's pretraining.

### D.1. Data transformations

The following elementwise transformations were applied to input fields independent of normalization. This was performed uniformly across all experiments and models. Note that since these are applied uniformly, all metrics are computed on transformed fields.

- acoustic_scattering_...

    1. density: Zero replacement - removed to avoid numerical complications from high dynamic range field which is deterministic transform of other provided field.

- post_neutron_star_merger

    1. density: $\log_{10}$ - reduce long tail of strictly positive field in accordance with domain conventions.
    2. temperature: $\log_{10}$ - reduce long tail of strictly positive field in accordance with domain conventions.
    3. pressure: $\log_{10}$ - reduce long tail of strictly positive field in accordance with domain conventions.
    4. entropy: $\log_{10}$ - reduce long tail of strictly positive field in accordance with domain conventions.
    5. internal_energy: $\log_{10}$ - reduce long tail of strictly positive field in accordance with domain conventions.
    6. boundary - Set second boundary to open instead of periodic to as this boundary is a mirror symmetry and therefore only periodic for scalars.
    7. all: interpolate(size=(..., 64), mode="trilinear", align_corners=False) - Interpolate dim from 66-64 to simplify ingestion.

- `supernova_explosion_128`

    1. `density`: $\log_{10}$ - reduce long tail of strictly positive field in accordance with domain conventions.
    2. `temperature`: $\log_{10}$ - reduce long tail of strictly positive field in accordance with domain conventions.

- `turbulence_gravity_cooling`

    1. `density`: $\log_{10}$ - reduce long tail of strictly positive field in accordance with domain conventions.
    2. `temperature`: $\log_{10}$ - reduce long tail of strictly positive field in accordance with domain conventions.

- `turbulent_radiative_layer_3D`

    1. `density`: $\log_{10}$ - reduce long tail of strictly positive field in accordance with domain conventions.
    2. `temperature`: $\log_{10}$ - reduce long tail of strictly positive field in accordance with domain conventions.

- `convective_envelope_rsg`

    1. `all`: time_stride=5 - Originally set to 5 during early experiments and maintained for continuity. Not necessary for performance.

### D.2. Boundary Handling

In the case of periodic boundary conditions, the computations described in Section A.1 can be performed exactly. As WALRUS is intended to be general-purpose and not constrained to settings where the boundary conditions are exactly known, we opt for a simple topological description of boundary conditions indicating whether a boundary is:

1. Open - The domain extends beyond the sub-domain we are currently viewing. We do not know what is beyond the boundary.

2. Closed - There is some form of barrier marking this as the limit of the domain. The model is unaware of what conditions are enforced at this barrier.

3. Periodic - There is not truly a boundary here and the neighboring points are on the opposite side of the domain.

For non-periodic boundaries, padding is implemented with additional channels containing binary masks for each topological boundary type. Open and closed boundaries are processed as separate channels so that the model can learn to treat each differently. For the case where downstream boundary conditions are known, we suggest employing ghost-node or extrapolation style padding as are commonly used in numerical packages (Clawpack Development Team, 2021; Holl & Thuerey, 2024) though the impact of this is not explored in this work.

## E. Modes of Operation for Dynamics Models

One important point when comparing emulation models is the mode of operation. In practice, there is a wide spectrum of information one can assume the model has access to. For instance, given full observation of all state variables and coefficients in a system governed by PDEs sampled at the generating resolution, one would be operating in a Markovian setting in which one timestep provides all necessary information to make the prediction. Foundation models such as Poseidon make this choice to greatly improve model efficiency. The Markovian framing can greatly accelerate the runtime performance of such models as it reduces disk IO and, depending on modeling choices, may alleviate the need for specialized temporal compression blocks or reduce token counts.

Models like WALRUS on the other hand, opt to use context information in order to improve generalization in cases where full information is unavailable. To demonstrate the advantages and disadvantages of this approach, we construct a simple incomplete information experiment where $\boldsymbol{u}(\boldsymbol{x}, t) \in \mathcal{L}^2(\mathbb{T}^2) \times [0, \infty)$ evolves according to the following linear PDE

$$\frac{\partial \boldsymbol{u}}{\partial t} = \nu_x \frac{\partial \boldsymbol{u}}{\partial \boldsymbol{x}} + \nu_y \frac{\partial \boldsymbol{u}}{\partial \boldsymbol{y}}.$$

where $\boldsymbol{u}(\boldsymbol{x}, 0) \sim \mathcal{N}(\boldsymbol{m}, \sigma I)$ is a 2D Gaussian, with the center $\boldsymbol{m} \in [0.25, 0.75]^2$ and an isotropic variance with $\sigma \in \mathcal{U}(0.05, 0.125)$. For a given PDE, the parameter $\nu$ is independently sampled form a uniform distribution, i.e., $\nu_x, \nu_y \sim$

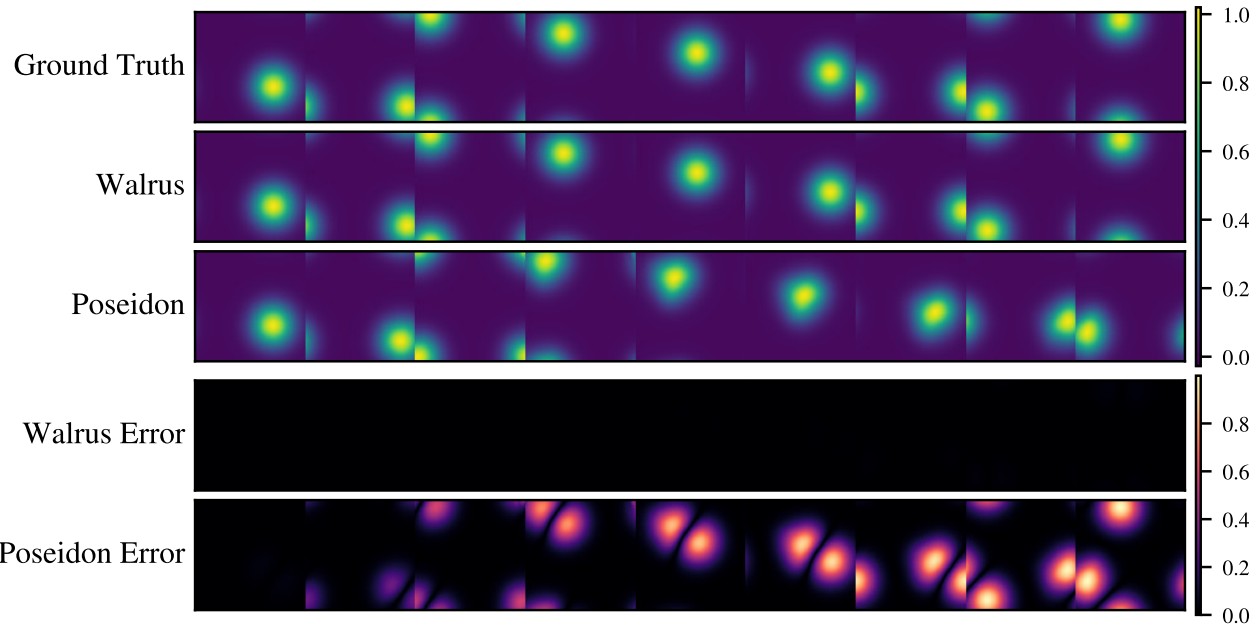

*Figure 11.* Comparsion between WALRUS and the Poseidon-L model on linear advection of smooth initial conditions. Without history, the task becomes degenerate as it is impossible to infer the missing system parameters.

$\mathcal{U}[0.05, 0.5]$. Linear advection on a periodic domain can be solved analytically as $\boldsymbol{u}(\boldsymbol{x}, t) = \boldsymbol{u}(\begin{bmatrix} x - \nu_x t \\ y - \nu_y t \end{bmatrix}, 0)$ which we do here to ensure accuracy of the solution function.

If we task both WALRUS and Poseidon to learn to emulate this system *without* any knowledge of $\boldsymbol{\nu}$, the result in Figure 11 shows that Poseidon is unable to make predictions while WALRUS easily learns the system. Without access to multiple steps, one-step models are forced to estimate a mean tendency. This pattern can be seen in many of the datasets within TheWell. This implies that for these one-step models to learn to make predictions in incompletely described systems, the initial conditions must be representative of the missing conditions.

Note that this isn't a statement on the core architectures used in either model. The SWIN (Liu et al., 2021) approach is an enormously successful backbone and from the relative performance in vision should be expected to be near interchangeable with the dense space-time ViT used here. WALRUS uses global attention to be able to adapt for very fast moving fields where transport may result in several-window jumps in a windowed base model, but this could also be controlled through other means. What this is a statement on is history vs Markovian analysis. When we restrict WALRUS to one frame in Figure 12, we can see very similar results between the two architectures. While here we see almost no drop-off going even to two frames, we still opt to use a larger frame count for the core model. This is a simple model where the true coefficient can also be recovered analytically from two frames given prior knowledge of the functional form.

**Negatives of fully preserved history-based approach.** It is important to mention that the history-based approach is not strictly superior. The efficiency difference between a model optimized for Markovian settings and one intended to run on longer histories can be significant. In Table 8, we show the timings of WALRUS compared with Poseidon measured on a single NVIDIA H100. While the comparison between WALRUS and Poseidon is not direct – WALRUS is twice as large, contains temporal-specialized layers, and built to handle multiple dimensionalities and a much broader class of behavior – it is illustrative of the types factors that can enter the decision making process.

Given these are single sample measurements, the GPU is under relatively low utilization and at small contexts, the GPU is able to absorb the additional FLOP budget of WALRUS sublinearly. However, we eventually see that this saturates as the context grows. Note that due to the small overall context length, the $O(N^2 D)$ cost of temporal attention is miniscule compared to the $O(ND^2)$ linear projections and MLPs rendering the practical asymptotics $O(N)$ in time so under high capacity, we would expect WALRUS to scale linearly with context. We can clearly see that smaller contexts offer a significant

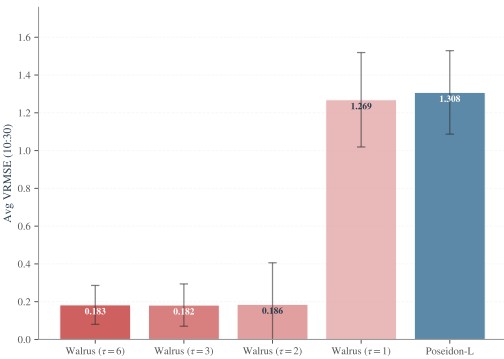

*Figure 12.* We can see the ability to infer missing information is a function of history rather than architecture as WALRUS also struggles as history is reduced to a single frame. Note that this is a simplified setting where the missing information can be analytically derived from two frames.

*Table 8.* Inference time per sample on the linear advection example, measured with 10 warmup and 20 timed forward passes on a single GPU. Note that irrelevant temporal operations are not disabled in the benchmarking code for context=1.

| Model | ms/sample |
|---|---|
| Walrus ($\tau = 6$) | $238.96 \pm 0.10$ |
| Walrus ($\tau = 3$) | $132.67 \pm 0.76$ |
| Walrus ($\tau = 2$) | $95.46 \pm 0.31$ |
| Walrus ($\tau = 1$) | $88.44 \pm 0.10$ |
| Poseidon-L | $72.71 \pm 1.35$ |

runtime benefit.

This is why it is important to note that the impact of this decision is problem-dependent. In well-understood cases where the dynamics are entirely driven by initial conditions, the one-step approach can prove to be an enormous efficiency advantage. Other foundation models like DPOT employ a middle path where the time dimension is entirely squashed by the encoder. This reduces the number of tokens fed to the processing blocks, but also places sole responsibility for extracting necessarily temporal patterns on the encoder alone. Finding an optimal approach which balances these needs is an open area of research.

## F. Evaluation and Additional Results

### F.1. Evaluation Metrics

#### F.1.1. VRMSE METRIC

The variance scaled mean squared error (VMSE): it is the MSE normalized by the variance of the truth. It has been used for the benchmarking of the Well (Ohana et al., 2025).

$$\text{VMSE}(u, v) = \frac{\langle |u - v|^2 \rangle}{(\langle |u - \bar{u}|^2 \rangle + \epsilon)}.$$

We chose to report its square root variant, the VRMSE:

$$\text{VRMSE}(u, v) = \frac{\langle |u - v|^2 \rangle^{1/2}}{(\langle |u - \bar{u}|^2 \rangle + \epsilon)^{1/2}}.$$

Note that, since $\text{VRMSE}(u, \bar{u}) \approx 1$, having $\text{VRMSE} > 1$ indicates worse results than an accurate estimation of the spatial mean $\bar{u}$. For all reported results, $\epsilon = 10^{-7}$.

| Dataset | VRMSE: One-step | | | |
|---|---|---|---|---|
| | MPP-AViT-L | Poseidon-L | DPOT-H | Walrus |
| **2D** | | | | |
| PDEG CE-RM | 0.1413 (±0.0469) | 0.0769 (±0.0300) | 0.1107 (±0.0347) | **0.0118** (±0.0049) |
| FB Skelenton | 0.0222 (±0.0052) | 0.0111 (±0.1030) | 0.0086 (±0.0056) | **0.0006** (±0.0005) |
| BML PoolBoilSubcool | 0.1012 (±0.0476) | 0.0829 (±0.0386) | 0.1954 (±0.0412) | **0.0617** (±0.0328) |
| PDEA ConditionedINS | 0.0456 (±0.0236) | 0.0291 (±0.0174) | 0.0404 (±0.0196) | **0.0040** (±0.0039) |
| **3D** | | | | |
| PDEB CNS 3D Turb | — | — | 0.2115 (±0.0181) | **0.0276** (±0.0057) |
| PDEB CNS 3D Rand | — | — | 0.2920 (±0.0654) | **0.0905** (±0.0324) |
| RSG Convective Envelope | — | — | 0.1121 (±0.0043) | **0.0587** (±0.0027) |
| Post Neutron Star Merger | — | — | 0.2394 (± 0.0396) | **0.2013** (± 0.0507) |

*Table 9.* OOD datasets: Median VRMSE ± std (one-step, averaged over all steps). Lower is better.

### F.1.2. SPATIALLY AVERAGED $W_1$ METRIC

The spatially integrated $s$-Wasserstein distance measures the discrepancy between a ground-truth conditional distribution and an approximated one, averaged (integrated) over the $M$ spatial points of the domain. In GenCFD (Molinaro et al., 2025), this was used to measure differences between state distributions. Here, the distribution is computed empirically by evaluating all states in a simulated trajectory. For ergodic systems with sufficiently long sampling periods, this is functionally equivalent; however, few systems here reach ergodic states. For each spatial point $x_i$, the one-dimensional $s$-Wasserstein distance between the (pointwise) ground-truth distribution $p_{\text{exact}}$ and the approximated distribution $p$ is computed from their cumulative distribution functions $F_{\text{exact}}$ and $F$, then integrated over space:

$$W_s(p_{\text{exact}}, p) = \left( \sum_{i=1}^{M} \int_0^1 \left| F_{\text{exact}}^{-1}(u(x_i)) - F^{-1}(u(x_i)) \right|^s \, \mathrm{d}u \right)^{1/s}.$$

We report the $s = 1$ variant, the spatially integrated 1-Wasserstein distance:

$$W_1(p_{\text{exact}}, p) = \sum_{i=1}^{M} \int_0^1 \left| F_{\text{exact}}^{-1}(u(x_i)) - F^{-1}(u(x_i)) \right| \, \mathrm{d}u,$$

where $F^{-1}$ denotes the quantile function (inverse CDF). Note that, since $W_1$ uses the full conditional distribution rather than only its mean or variance, a low value indicates that the entire generated probability distribution (and not just its first two moments) matches the ground truth. However, this does not evaluate the coherent structures in the data as each spatial point is evaluated independently and the metric is permutation invariant in time.

## F.2. Additional Results for Downstream Performance

### F.2.1. TABLE-FORMATTED LOSSES

Tables 9, 10, 11 show the finetuning dataset loss in table format for easier reference along with the standard deviation of the per-sample statistics. Note that per sample variation is not available for Post Neutron Star Merger data as there is a single validation trajectory.

### F.2.2. DETAILED LOSS OVER TIME PLOTS

Figure 13 shows the median VRMSE over each available step of the rollout for more fine-grained analysis. The shaded region represents plus/minus one standard deviation computed over samples in the data set. Generally, we see WALRUS has a sizable advantage in each downstream scenario. On the PoolBoilSubcool scenario, MPP-AViT-L eventually surpasses WALRUS though qualitatively speaking, the MPP model appears to learn to stop predicting bubbles to minimize loss, while

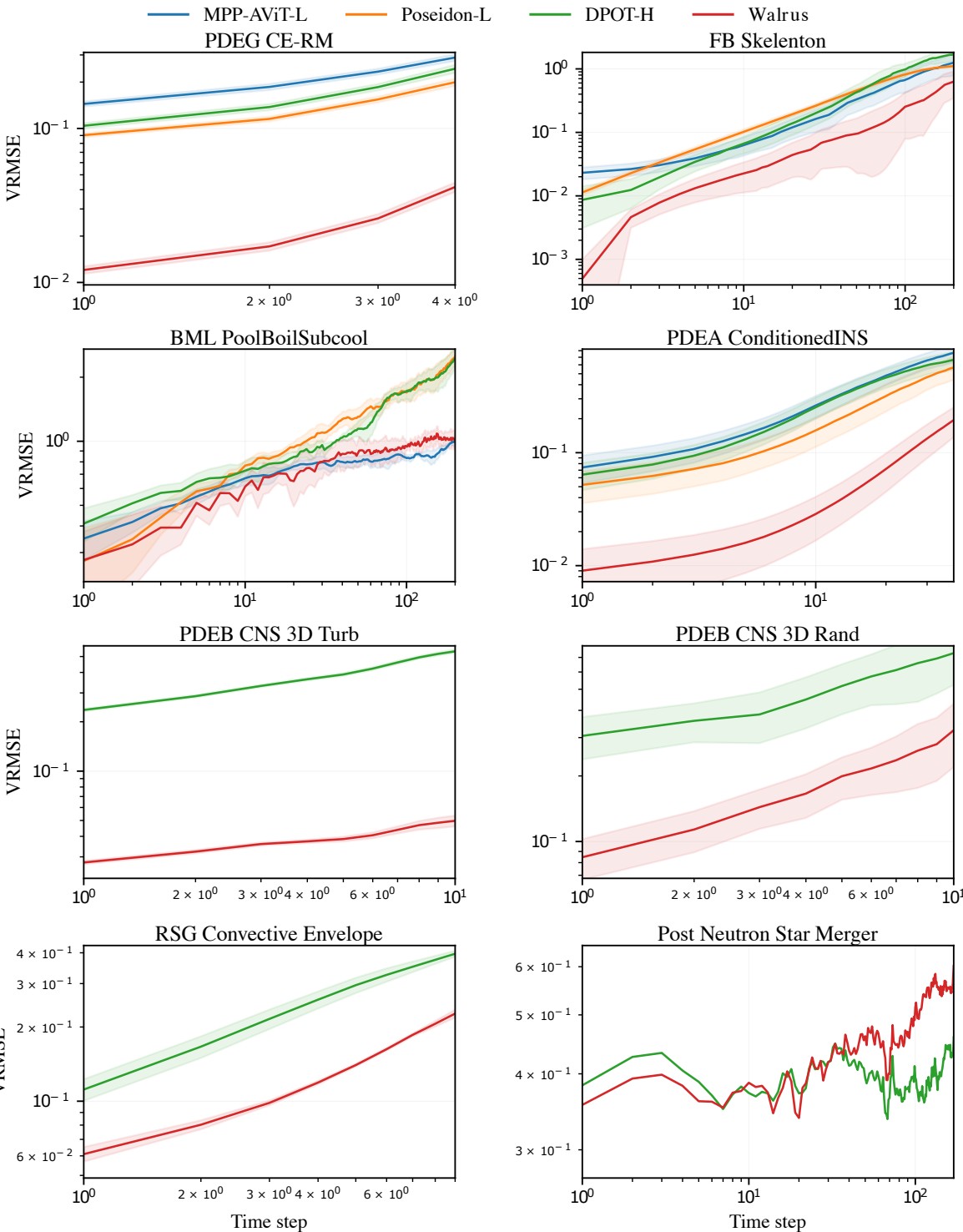

*Figure 13.* VRMSE over time for downstream tasks. Shaded region denotes $\pm\sigma$ where standard deviation is computed empirically over test samples at given step.

| Dataset | VRMSE: Avg $T \in [1:10]$ | | | |
|---|---|---|---|---|
| | MPP-AViT-L | Poseidon-L | DPOT-H | Walrus |
| **2D** | | | | |
| PDEG CE-RM | 0.2132 (±0.0097) | 0.1399 (±0.0062) | 0.1679 (±0.0084) | **0.0242** (±0.0099) |
| FB Skelenton | 0.0422 (±0.0115) | 0.0583 (±0.0052) | 0.0371 (±0.0117) | **0.0132** (±0.0056) |
| BML PoolBoilSubcool | 0.5446 (±0.0430) | 0.5549 (±0.0628) | 0.6182 (±0.0675) | **0.4656** (±0.0594) |
| PDEA ConditionedINS | 0.1608 (±0.0364) | 0.1006 (±0.0295) | 0.1488 (±0.0337) | **0.0178** (±0.0072) |
| **3D** | | | | |
| PDEB CNS 3D Turb | — | — | 0.4039 (±0.0085) | **0.0401** (±0.0018) |
| PDEB CNS 3D Rand | — | — | 0.5281 (±0.1474) | **0.2025** (±0.0592) |
| RSG Convective Envelope | — | — | 0.2772 (±0.0036) | **0.1422** (±0.0031) |
| Post Neutron Star Merger | — | — | **0.3877** (± —) | 0.3736 (± —) |

*Table 10.* OOD datasets: Median VRMSE ± std averaged over $T \in [1:10]$. Lower is better.

| Dataset | VRMSE: Avg $T \in [11:30]$ | | | |
|---|---|---|---|---|
| | MPP-AViT-L | Poseidon-L | DPOT-H | Walrus |
| **2D** | | | | |
| FB Skelenton | 0.1215 (±0.0437) | 0.1973 (±0.0164) | 0.1432 (±0.0495) | **0.0440** (±0.0273) |
| BML PoolBoilSubcool | 0.7496 (±0.0865) | 0.9468 (±0.0696) | 0.8485 (±0.0684) | **0.7095** (±0.0786) |
| PDEA ConditionedINS | 0.4834 (±0.0015) | 0.3226 (±0.0841) | 0.4571 (±0.0860) | **0.0777** (±0.0257) |
| **3D** | | | | |
| Post Neutron Star Merger | — | — | 0.3933 (± —) | **0.3873** (± —) |

*Table 11.* OOD datasets: Median VRMSE ± std averaged over $T \in [11:30]$. Lower is better. — = model not applicable. Spread not computable for PNS due to single trajectory.

### F.2.3. TRAJECTORY METRICS

In Table 12, we supplement the pointwise analysis with comparisons between the full trajectories. These results largely align with the long term pointwise metrics here. We can see that WALRUS is similarly able to capture local long-term distributions of physical values in the finetuning data perhaps with a larger advantage in most cases compared to pure pointwise evaluation.

### F.3. Additional Results for Cross-Domain Analysis

### F.3.1. DETAILED LOSS OVER TIME PLOTS

Figure 14 shows the VRMSE over each step. The shaded region represents plus/minus one standard deviation over the data set. Walrus without finetuning included for reference. We can see that despite seeing less total data per scenario with heavy problem-altering augmentation, the model is still able to compete with the finetuned models, though finetuning greatly reduces the rate at which error accumulates.

### F.3.2. EXPANDED TABULAR METRICS

For completeness, we include the VRMSE tables from 1 broken out into Tables 13, 14, 15 so that within-dataset variance can be included.

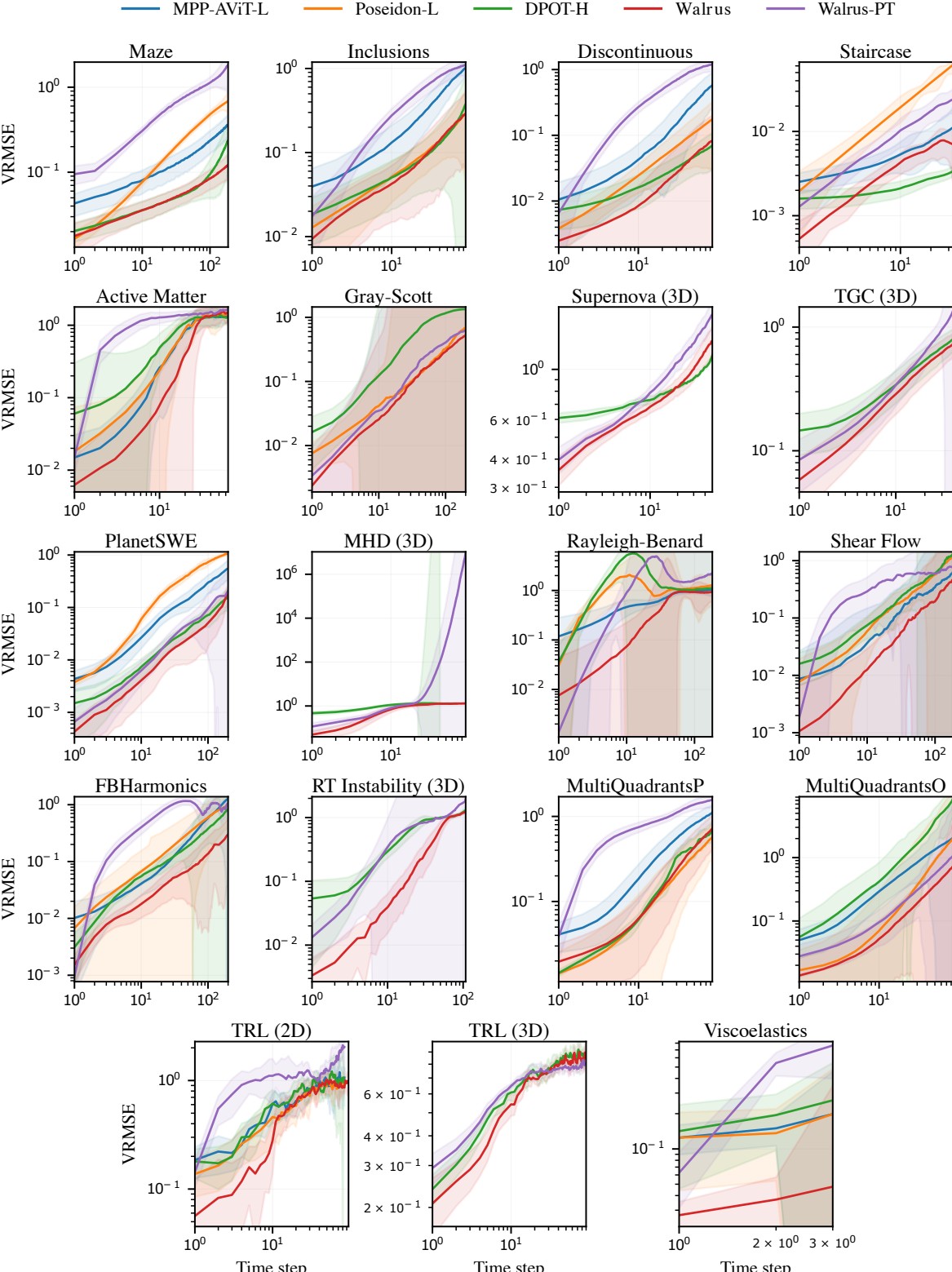

*Figure 14.* VRMSE over time for tasks in training distribution. Shaded region denotes $\pm\sigma$ where standard deviation is computed empirically over test samples at given step.

3028

| Dataset | Spat. Avg. Temporal $W_1$ (T=all) | | | |
| --- | --- | --- | --- | --- |
| | MPP-AViT-L | Poseidon-L | DPOT-H | Walrus |
| **2D** | | | | |
| PDEG CE-RM | 0.2330 (±0.0122) | 0.1548 (±0.0173) | 0.1941 (±0.0120) | **0.0301** (±0.0089) |
| FB Skelenton | 0.3682 (±0.1593) | 0.4517 (±0.0496) | 0.3606 (±0.0682) | **0.0652** (±0.0425) |
| BML PoolBoilSubcool | **0.1326** (±0.0573) | 1.1677 (±0.1708) | 0.4192 (±0.1050) | 0.1495 (±0.0173) |
| PDEA ConditionedINS | 0.1710 (±0.0221) | 0.1195 (±0.0225) | 0.1635 (±0.0221) | **0.0294** (±0.0058) |
| **3D** | | | | |
| PDEB CNS 3D Turb | — | — | 0.2208 (±0.0034) | **0.0266** (±0.0010) |
| PDEB CNS 3D Rand | — | — | 0.2378 (±0.0612) | **0.1044** (±0.0342) |
| RSG Convective Envelope | — | — | 0.0267 (±0.0107) | **0.0017** (±0.0052) |
| Post Neutron Star Merger | — | — | **0.0613** (±—) | 0.2244 (±—) |

*Table 12.* OOD datasets: Median spatially-averaged temporal $W_1$ (± std) over the full rollout (T=all). Lower is better. — = not applicable.

### F.3.3. TRAJECTORY METRICS

Table 16 shows the spatially averaged W1 distance between trajectories. We again see that WALRUS consistently predicts realistic distributions of physical states across a broad set of challenging problems. While there are isolated examples on which prior models outperform WALRUS the consistent ability to emulate a breadth of distributions shows the advantages of the larger, more broadly pretrained model.

## G. Finetuning rollout examples

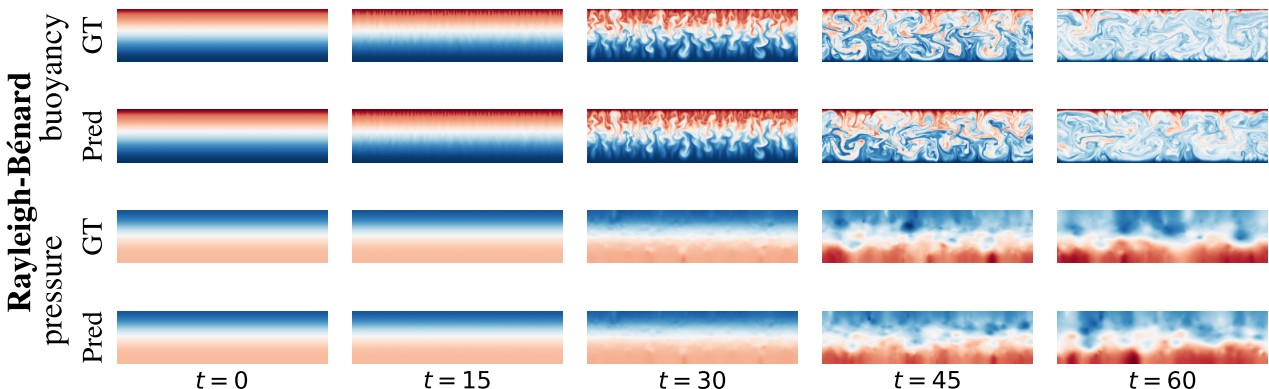

*Figure 15.* Rayleigh-Benard Convection in 2D.

| Dataset | VRMSE: One-step | | | |
|---|---|---|---|---|
| | MPP-AViT-L | Poseidon-L | DPOT-H | Walrus |
| *Biological and chemical behavior* | | | | |
| Active Matter | 0.0157 (±0.0106) | 0.0214 (±0.0914) | 0.0476 (±0.1301) | **0.0057** (±0.0059) |
| Gray-Scott | nan | 0.0048 (±0.5956) | 0.0061 (±0.0700) | **0.0001** (±0.0155) |
| *Acoustics and wave propagation* | | | | |
| Maze* | 0.0337 (±0.0115) | 0.0116 (±0.0080) | 0.0126 (±0.0045) | **0.0099** (±0.0055) |
| Inclusions | 0.0432 (±0.0262) | 0.0127 (±0.0114) | 0.0199 (±0.0136) | **0.0089** (±0.0066) |
| Discontinuous | 0.0097 (±0.0077) | 0.0035 (±0.0025) | 0.0055 (±0.0029) | **0.0021** (±0.0017) |
| Staircase | 0.0026 (±0.0007) | 0.0019 (±0.0015) | 0.0021 (±0.0011) | **0.0005** (±0.0002) |
| *Astrophysical and geoscience applications* | | | | |
| Supernova (3D) | — | — | 0.6417 (±0.0304) | **0.2462** (±0.1031) |
| TGC (3D) | — | — | 0.1002 (±0.0730) | **0.0466** (±0.0854) |
| PlanetSWE | 0.0035 (±0.0010) | 0.0035 (±0.0005) | 0.0013 (±0.0006) | **0.0004** (±0.0002) |
| *Plasmas* | | | | |
| MHD (3D) | — | — | 0.4734 (±0.0666) | **0.0580** (±0.0229) |
| *Viscous fluids* | | | | |
| Rayleigh-Benard | 0.0264 (±0.0549) | 0.0215 (±0.0473) | 0.0288 (±0.0382) | **0.0059** (±0.0265) |
| Shear Flow | 0.0071 (±0.0124) | 0.0090 (±0.0497) | 0.0162 (±0.0180) | **0.0012** (±0.0080) |
| FBHarmonics | 0.0104 (±0.0095) | 0.0068 (±0.1212) | 0.0031 (±0.0025) | **0.0005** (±0.0004) |
| RT Instability (3D) | — | — | 0.2165 (±0.0916) | **0.0565** (±0.0303) |
| *Inviscid fluids* | | | | |
| MultiQuadrantsP | 0.0418 (±0.0146) | **0.0142** (±0.0102) | 0.0397 (±0.0128) | 0.0194 (±0.0098) |
| MultiQuadrantsO | 0.0432 (±0.0253) | 0.0158 (±0.0264) | 0.0468 (±0.1344) | **0.0112** (±0.0086) |
| TRL (2D) | 0.1707 (±0.0868) | 0.1323 (±0.0780) | 0.1601 (±0.0759) | **0.0831** (±0.0638) |
| TRL (3D) | — | — | 0.2238 (±0.0430) | **0.1588** (±0.0490) |
| *Non-newtonian fluids* | | | | |
| Viscoelastics | 0.1013 (±0.1618) | 0.0878 (±0.1783) | 0.1398 (±0.1820) | **0.0295** (±0.1746) |

*Table 13.* Median VRMSE ± std (one-step, averaged over all steps). Lower is better. Dark shade = closer to best. — = model not applicable.

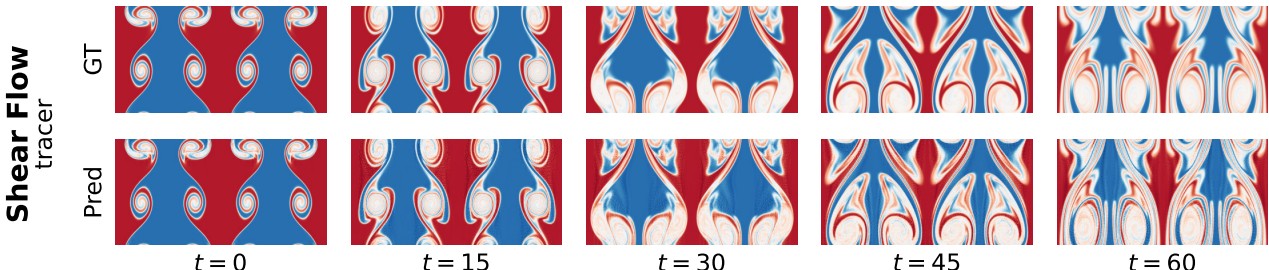

*Figure 16.* Shear flow in 2D.

| Dataset | VRMSE: Avg $T \in [1:20]$ | | | |
|---|---|---|---|---|
| | MPP-AViT-L | Poseidon-L | DPOT-H | Walrus |
| *Biological and chemical behavior* | | | | |
| Active Matter | 0.3195 $(\pm 0.2073)$ | 0.3355 $(\pm 0.2634)$ | 0.5272 $(\pm 0.2944)$ | **0.1262** $(\pm 0.1810)$ |
| Gray-Scott | nan | 0.0411 $(\pm 0.6886)$ | 0.1354 $(\pm 2.2831)$ | **0.0278** $(\pm 2.7619)$ |
| *Acoustics and wave propagation* | | | | |
| Maze* | 0.0797 $(\pm 0.0245)$ | 0.0785 $(\pm 0.0078)$ | 0.0350 $(\pm 0.0094)$ | **0.0345** $(\pm 0.0094)$ |
| Inclusions | 0.1362 $(\pm 0.0759)$ | 0.0510 $(\pm 0.0256)$ | 0.0500 $(\pm 0.0292)$ | **0.0430** $(\pm 0.0295)$ |
| Discontinuous | 0.0460 $(\pm 0.0356)$ | 0.0255 $(\pm 0.0130)$ | 0.0162 $(\pm 0.0084)$ | **0.0093** $(\pm 0.0101)$ |
| Staircase | 0.0053 $(\pm 0.0022)$ | 0.0201 $(\pm 0.0050)$ | **0.0022** $(\pm 0.0013)$ | 0.0040 $(\pm 0.0010)$ |
| *Astrophysical and geoscience applications* | | | | |
| Supernova (3D) | — | — | 0.7366 $(\pm 0.0252)$ | **0.6673** $(\pm 0.0503)$ |
| TGC (3D) | — | — | 0.3482 $(\pm 0.1000)$ | **0.2889** $(\pm 0.0895)$ |
| PlanetSWE | 0.0307 $(\pm 0.0113)$ | 0.0730 $(\pm 0.0121)$ | 0.0081 $(\pm 0.0032)$ | **0.0046** $(\pm 0.0028)$ |
| *Plasmas* | | | | |
| MHD (3D) | — | — | 1.0250 $(\pm 0.0625)$ | **0.6487** $(\pm 0.1685)$ |
| *Viscous fluids* | | | | |
| Rayleigh-Benard | 0.4109 $(\pm 0.1976)$ | 1.3819 $(\pm 2.4001)$ | 3.4868 $(\pm 48.4441)$ | **0.0992** $(\pm 0.1477)$ |
| Shear Flow | 0.0377 $(\pm 0.0423)$ | 0.0657 $(\pm 0.0430)$ | 0.0772 $(\pm 0.0373)$ | **0.0146** $(\pm 0.0204)$ |
| FBHarmonics | 0.0420 $(\pm 0.0190)$ | 0.0686 $(\pm 0.1059)$ | 0.0526 $(\pm 0.0153)$ | **0.0186** $(\pm 0.0071)$ |
| RT Instability (3D) | — | — | 0.3191 $(\pm 0.1289)$ | **0.0496** $(\pm 0.0294)$ |
| *Inviscid fluids* | | | | |
| MultiQuadrantsP | 0.2010 $(\pm 0.0872)$ | **0.0682** $(\pm 0.0642)$ | 0.0749 $(\pm 0.0629)$ | 0.0720 $(\pm 0.0501)$ |
| MultiQuadrantsO | 0.3050 $(\pm 0.1358)$ | 0.0846 $(\pm 0.2899)$ | 0.4801 $(\pm 0.1808)$ | **0.0587** $(\pm 0.0522)$ |
| TRL (2D) | 0.4796 $(\pm 0.1362)$ | 0.4117 $(\pm 0.1497)$ | 0.5134 $(\pm 0.1502)$ | **0.3393** $(\pm 0.1626)$ |
| TRL (3D) | — | — | 0.5753 $(\pm 0.0444)$ | **0.5216** $(\pm 0.0578)$ |
| *Non-newtonian fluids* | | | | |
| Viscoelastics | 0.1578 $(\pm 0.1113)$ | 0.1532 $(\pm 0.1486)$ | 0.1989 $(\pm 0.1591)$ | **0.0373** $(\pm 0.1121)$ |

*Table 14.* Median VRMSE $\pm$ std averaged over $T \in [1:20]$ from initial conditions. Lower is better. Dark shade = closer to best. — = model not applicable.

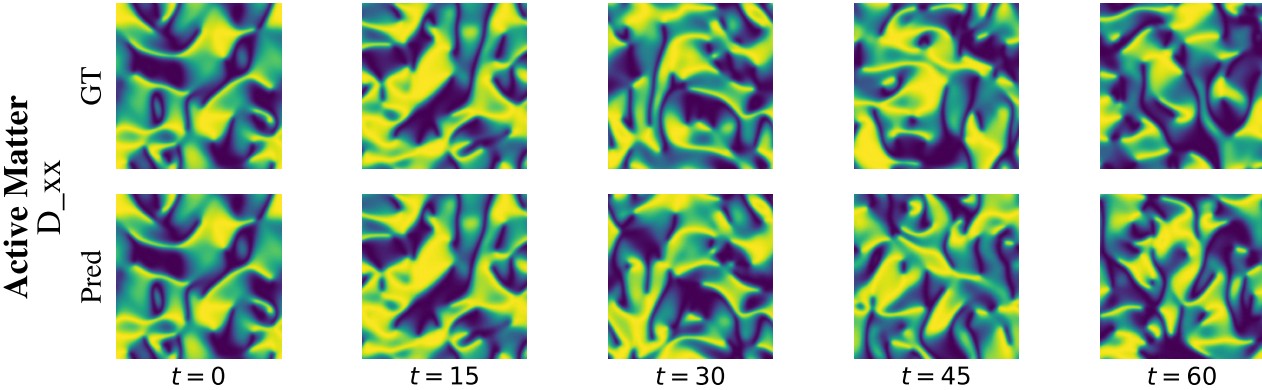

*Figure 17.* Active matter dynamics in 2D.

| Dataset | VRMSE: Avg $T \in [21:60]$ | | | |
|---|---|---|---|---|
| | MPP-AViT-L | Poseidon-L | DPOT-H | **Walrus** |
| *Biological and chemical behavior* | | | | |
| Active Matter | 1.2481 $(\pm 0.3363)$ | 1.3015 $(\pm 0.4767)$ | 1.2867 $(\pm 0.3227)$ | **1.2451** $(\pm 0.4126)$ |
| Gray-Scott | nan | 0.1295 $(\pm 14.1906)$ | 0.6571 $(\pm 21.2314)$ | **0.1268** $(\pm 26.8763)$ |
| *Acoustics and wave propagation* | | | | |
| Maze* | 0.1390 $(\pm 0.0432)$ | 0.2429 $(\pm 0.0195)$ | **0.0543** $(\pm 0.0139)$ | 0.0560 $(\pm 0.0139)$ |
| Inclusions | 0.4940 $(\pm 0.1797)$ | 0.1407 $(\pm 0.0674)$ | **0.1328** $(\pm 0.1003)$ | 0.1427 $(\pm 0.1080)$ |
| Discontinuous | 0.2310 $(\pm 0.1715)$ | 0.0890 $(\pm 0.0600)$ | 0.0398 $(\pm 0.0192)$ | **0.0385** $(\pm 0.0539)$ |
| Staircase | 0.0096 $(\pm 0.0056)$ | 0.0507 $(\pm 0.0110)$ | **0.0031** $(\pm 0.0011)$ | 0.0074 $(\pm 0.0027)$ |
| *Astrophysical and geoscience applications* | | | | |
| Supernova (3D) | — | — | **0.9669** $(\pm 0.0444)$ | 1.0963 $(\pm 0.1416)$ |
| TGC (3D) | — | — | 0.6809 $(\pm 0.1586)$ | **0.6197** $(\pm 0.1516)$ |
| PlanetSWE | 0.1213 $(\pm 0.0478)$ | 0.3452 $(\pm 0.0536)$ | 0.0317 $(\pm 0.0104)$ | **0.0207** $(\pm 0.0130)$ |
| *Plasmas* | | | | |
| MHD (3D) | — | — | 1.3055 $(\pm \infty)$ | **1.2256** $(\pm 0.4701)$ |
| *Viscous fluids* | | | | |
| Rayleigh-Benard | 0.7468 $(\pm 0.1288)$ | 0.9586 $(\pm 8.7464)$ | 1.2664 $(\pm 25.4808)$ | **0.6441** $(\pm 1.8700)$ |
| Shear Flow | 0.1814 $(\pm 0.2222)$ | 0.2408 $(\pm 0.1132)$ | 0.2880 $(\pm 3.5502)$ | **0.0810** $(\pm 0.1181)$ |
| FBHarmonics | 0.1785 $(\pm 0.1099)$ | 0.2526 $(\pm 0.2888)$ | 0.1546 $(\pm 0.0825)$ | **0.0603** $(\pm 0.0338)$ |
| RT Instability (3D) | — | — | 0.9321 $(\pm 0.3017)$ | **0.4776** $(\pm 0.1388)$ |
| *Inviscid fluids* | | | | |
| MultiQuadrantsP | 0.6860 $(\pm 0.2178)$ | **0.2807** $(\pm 0.1799)$ | 0.3839 $(\pm 0.1276)$ | 0.3408 $(\pm 0.1839)$ |
| MultiQuadrantsO | 1.1011 $(\pm 0.9423)$ | 0.6478 $(\pm 2.6427)$ | 2.7573 $(\pm 1.2301)$ | **0.2823** $(\pm 0.3932)$ |
| TRL (2D) | 0.8879 $(\pm 0.1795)$ | **0.8441** $(\pm 0.1913)$ | 0.9316 $(\pm 0.2156)$ | 0.8648 $(\pm 0.2265)$ |
| TRL (3D) | — | — | 0.8106 $(\pm 0.0612)$ | **0.7893** $(\pm 0.0571)$ |
| *Non-newtonian fluids* | | | | |
| Viscoelastics | — | — | — | — |

*Table 15.* Median VRMSE $\pm$ std averaged over $T \in [21:60]$ from initial conditions. Lower is better. Dark shade = closer to best. — = model not applicable.

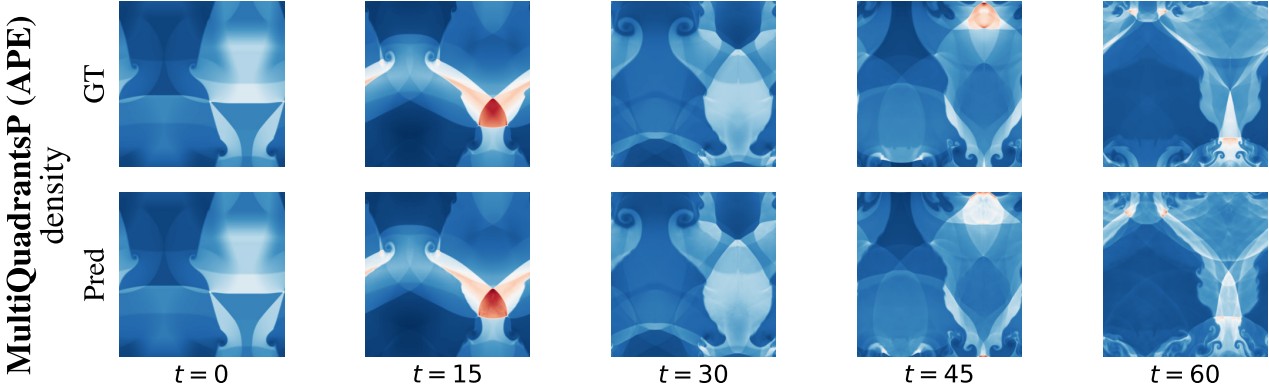

*Figure 18.* Euler multi-quadrants problem with periodic boundaries in 2D solved with absolute position embeddings (APE).

| Dataset | Spat. Avg. Temporal $W_1$ (T=all) | | | |
|---|---|---|---|---|
| | MPP-AViT-L | Poseidon-L | DPOT-H | **Walrus** |
| *Biological and chemical behavior* | | | | |
| Active Matter | 0.3180 ($\pm$0.1008) | 0.3282 ($\pm$0.1134) | 0.3613 ($\pm$0.1215) | **0.3129** ($\pm$0.1885) |
| Gray-Scott | — | **0.0748** ($\pm$0.5304) | 0.3434 ($\pm$0.1941) | 0.0824 ($\pm$0.8950) |
| *Acoustics and wave propagation* | | | | |
| Maze* | — | **0.0032** ($\pm$0.0137) | 0.0040 ($\pm$0.0059) | 0.0084 ($\pm$0.0047) |
| Inclusions | — | 0.0594 ($\pm$0.0167) | 0.0411 ($\pm$0.0198) | **0.0352** ($\pm$0.0232) |
| Discontinuous | 0.0613 ($\pm$0.0228) | 0.0563 ($\pm$0.0198) | 0.0228 ($\pm$0.0073) | **0.0182** ($\pm$0.0098) |
| Staircase | 0.0028 ($\pm$0.0012) | 0.0252 ($\pm$0.0048) | 0.0027 ($\pm$0.0005) | **0.0007** ($\pm$0.0003) |
| *Astrophysical and geoscience applications* | | | | |
| Supernova (3D) | — | — | 0.2460 ($\pm$0.0461) | **0.0569** ($\pm$0.0090) |
| TGC (3D) | — | — | 0.2036 ($\pm$0.0437) | **0.1743** ($\pm$0.0433) |
| PlanetSWE | 0.1077 ($\pm$0.0378) | 0.2104 ($\pm$0.0164) | 0.0303 ($\pm$0.0097) | **0.0233** ($\pm$0.0141) |
| *Plasmas* | | | | |
| MHD (3D) | — | — | — | **0.3939** ($\pm$0.0704) |
| *Viscous fluids* | | | | |
| Rayleigh-Benard | 0.5180 ($\pm$0.1018) | 0.4021 ($\pm$0.3129) | 0.3844 ($\pm$1.3932) | **0.2734** ($\pm$0.1212) |
| Shear Flow | 0.1896 ($\pm$0.3270) | 0.2931 ($\pm$0.2514) | 0.2643 ($\pm$1.1656) | **0.1224** ($\pm$0.6247) |
| FBHarmonics | 0.3096 ($\pm$0.2778) | 0.2876 ($\pm$0.4533) | 0.1022 ($\pm$0.4306) | **0.0370** ($\pm$0.0285) |
| RT Instability (3D) | — | — | 0.3750 ($\pm$0.0388) | **0.2644** ($\pm$0.0184) |
| *Inviscid fluids* | | | | |
| MultiQuadrantsP | 0.1652 ($\pm$0.0357) | 0.0866 ($\pm$0.0531) | 0.2513 ($\pm$0.0462) | **0.0762** ($\pm$0.0297) |
| MultiQuadrantsO | 0.4435 ($\pm$0.1393) | 0.3180 ($\pm$0.9788) | 0.3451 ($\pm$0.3711) | **0.0878** ($\pm$0.0676) |
| TRL (2D) | 0.2477 ($\pm$0.0959) | 0.2538 ($\pm$0.1409) | **0.1590** ($\pm$0.1292) | 0.2068 ($\pm$0.1000) |
| TRL (3D) | — | — | 0.0913 ($\pm$0.0194) | **0.0837** ($\pm$0.0292) |
| *Non-newtonian fluids* | | | | |
| Viscoelastics | 0.0414 ($\pm$0.0490) | 0.0472 ($\pm$0.0481) | 0.0919 ($\pm$0.0619) | **0.0136** ($\pm$0.0489) |

*Table 16.* Median spatially-averaged temporal $W_1$ distance ($\pm$ std) over the full rollout (T=all). Lower is better. Dark shade = closer to best. — = not applicable.

| Dataset | Short Name | Figure Ref |
|---|---|---|
| `acoustic_scattering_discontinuous` | Discontinuous | Figure 24 |
| `acoustic_scattering_inclusions` | Inclusions | Figure 23 |
| `acoustic_scattering_maze` | Maze | Figure 22 |
| `active_matter` | Active Matter | Figure 17 |
| `euler_multiquadrants_periodicBC` | MultiQuadrantsP | Figures 18 19 |
| `euler_multiquadrants_openBC` | MultiQuadrantsO | Figures 2021 |
| `gray_scott_reaction_diffusion` | Gray-Scott | Figure 26 |
| `helmholtz_staircase` | Staircase | Figure 27 |
| `MHD` | MHD (3D) | Figure 35 |
| `planetswe` | PlanetSWE | Figure 29 |
| `rayleigh_benard` | Rayleigh-Benard | Figure 15 |
| `rayleigh_taylor_instability` | RT Instability (3D) | Figure 40 |
| `shear_flow` | Shear Flow | Figure 16 |
| `supernova_explosion` | Supernova (3D) | Figure 41 |
| `turbulence_gravity_cooling` | TGC (3D) | Figure 39 |
| `turbulent_radiative_layer_2D` | TRL (2D) | Figure 28 |
| `turbulent_radiative_layer_3D` | TRL (3D) | Figure 42 |
| `viscoelastic_instability` | Viscoelastics | Figure 32 |
| `FPOHarmonics` | FBHarmonics | Figure 25 |

*Table 17.* Helper table for quick linking to specific rollout examples.

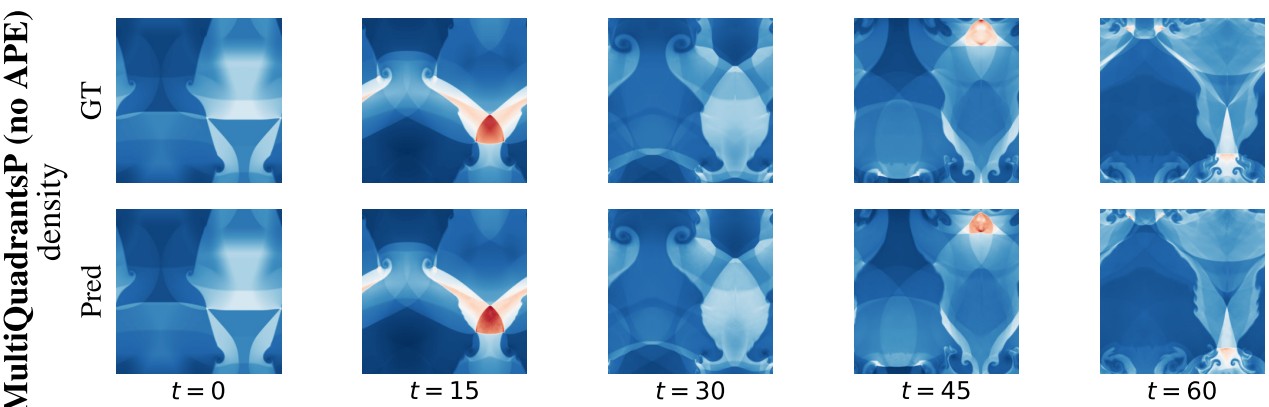

*Figure 19.* Euler multi-quadrants problem with periodic boundaries in 2D solved without APE.

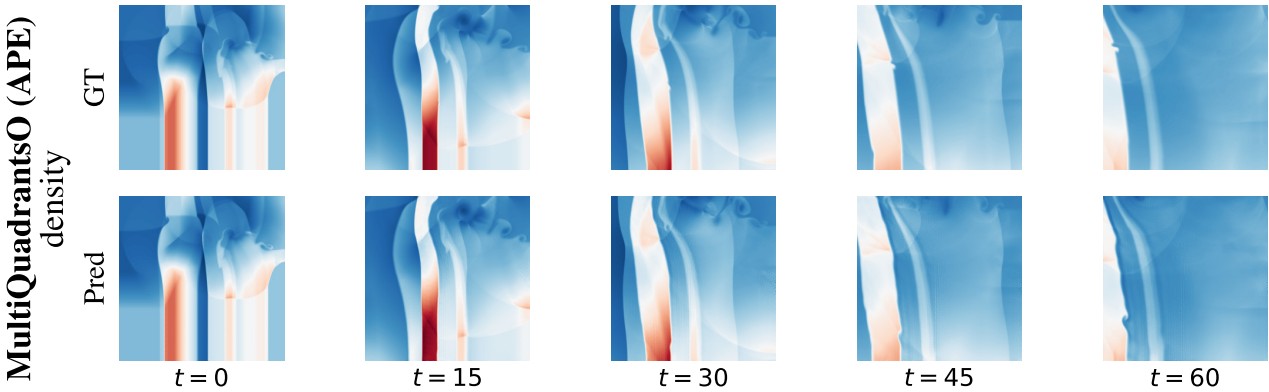

*Figure 20.* Euler multi-quadrants problem with open boundaries in 2D solved with APE.

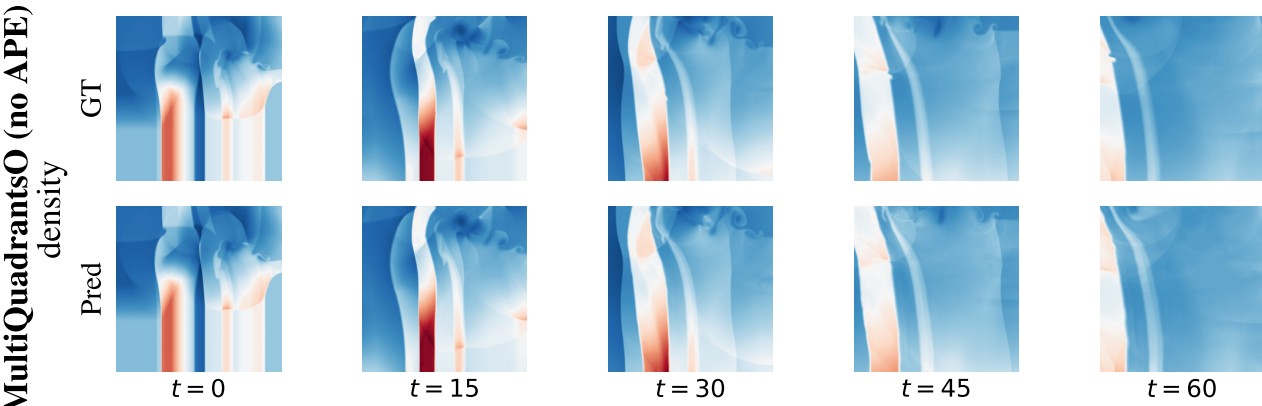

*Figure 21.* Euler multi-quadrants problem with periodic boundaries in 2D solved without APE.

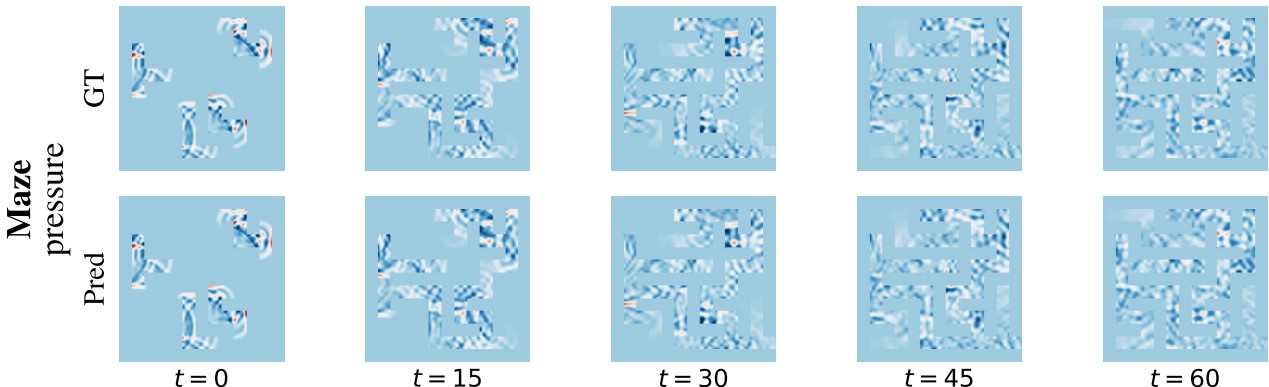

*Figure 22.* Acoustic scattering through a maze in 2D.

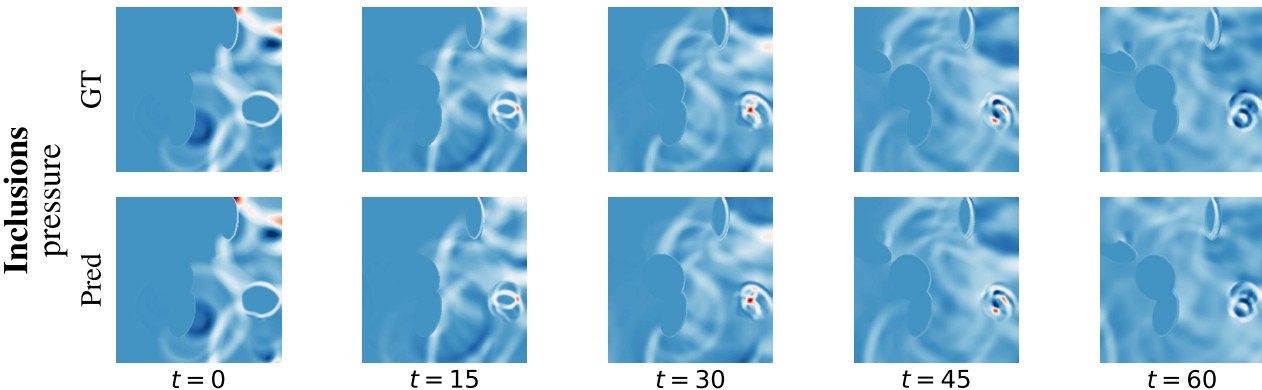

*Figure 23.* Acoustic scattering through medium with sharply varying inclusions in 2D.

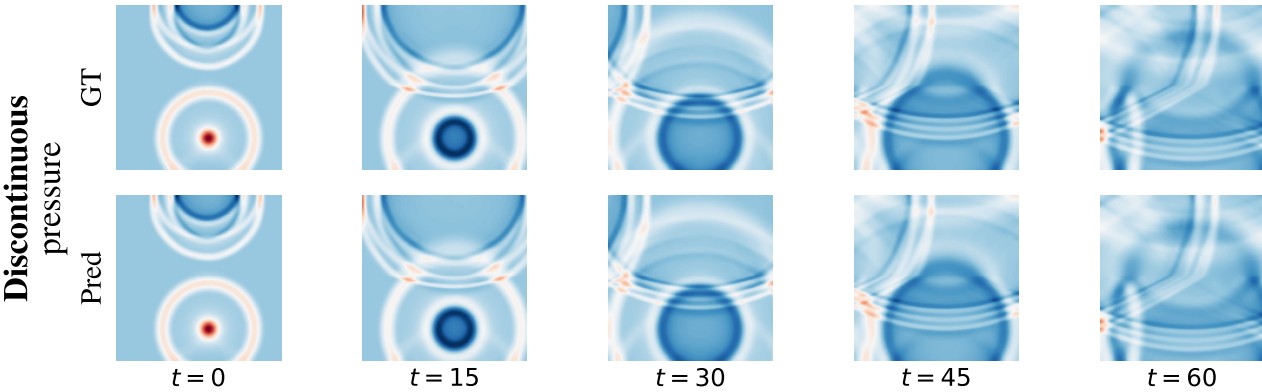

*Figure 24.* Acoustic scattering through medium with discontinuous density in 2D.

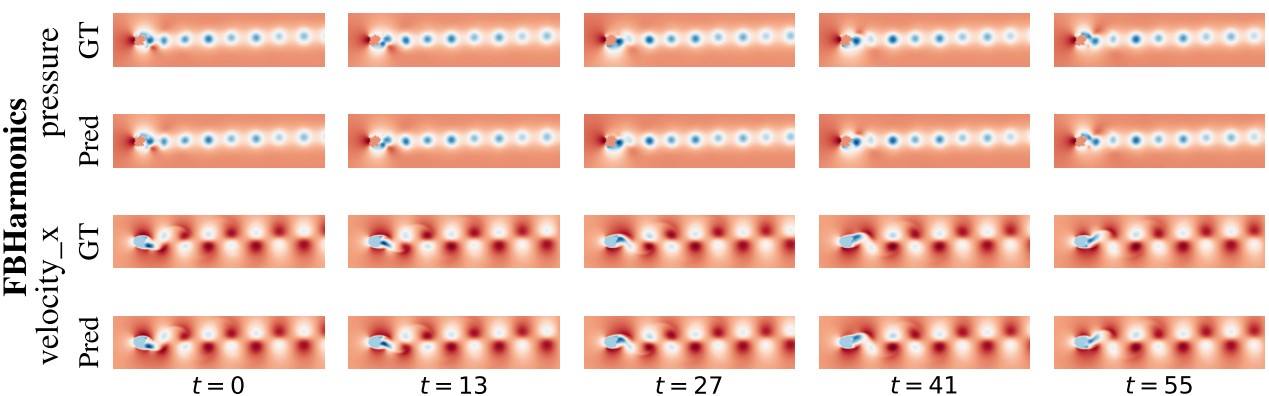

*Figure 25.* Flow around object with boundaries defined by spherical harmonics.

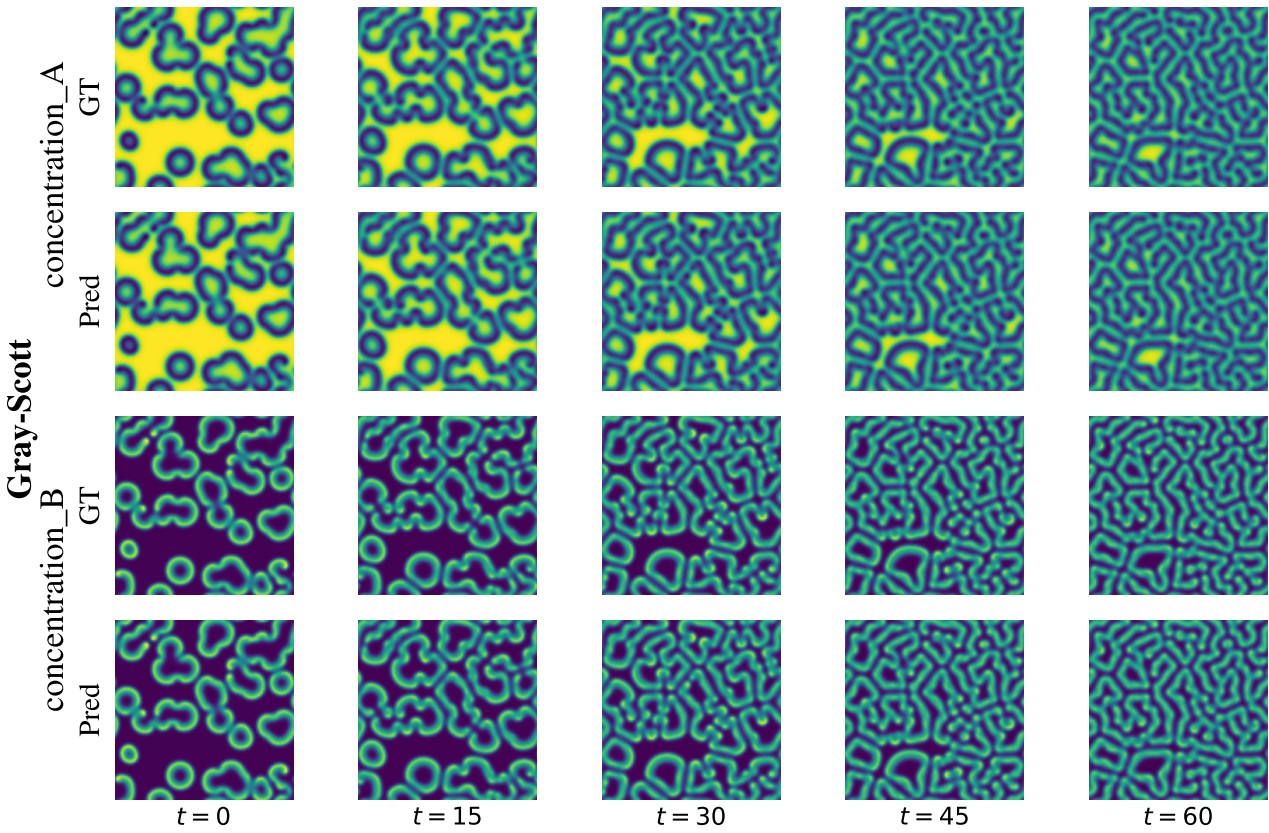

*Figure 26.* Gray-Scott diffusion reaction in 2D.

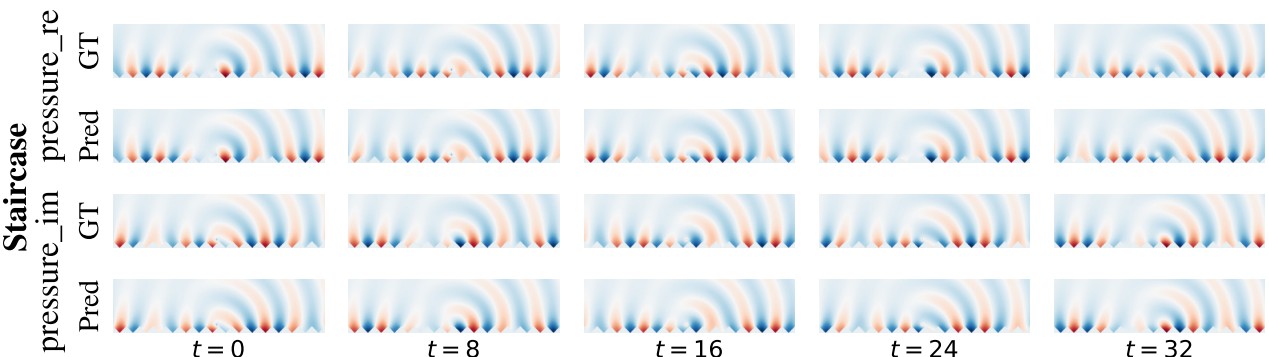

*Figure 27.* "Helmholtz Staircase" of wave propagation over infinite periodic surface in 2D.

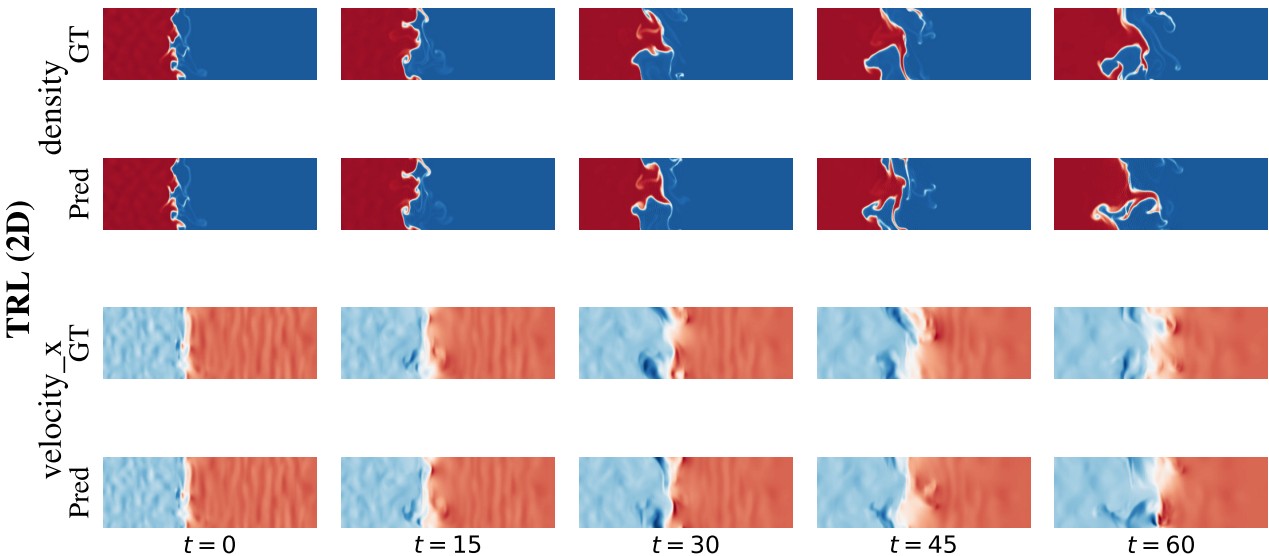

*Figure 28.* Turbulent radiative layer in 2D.

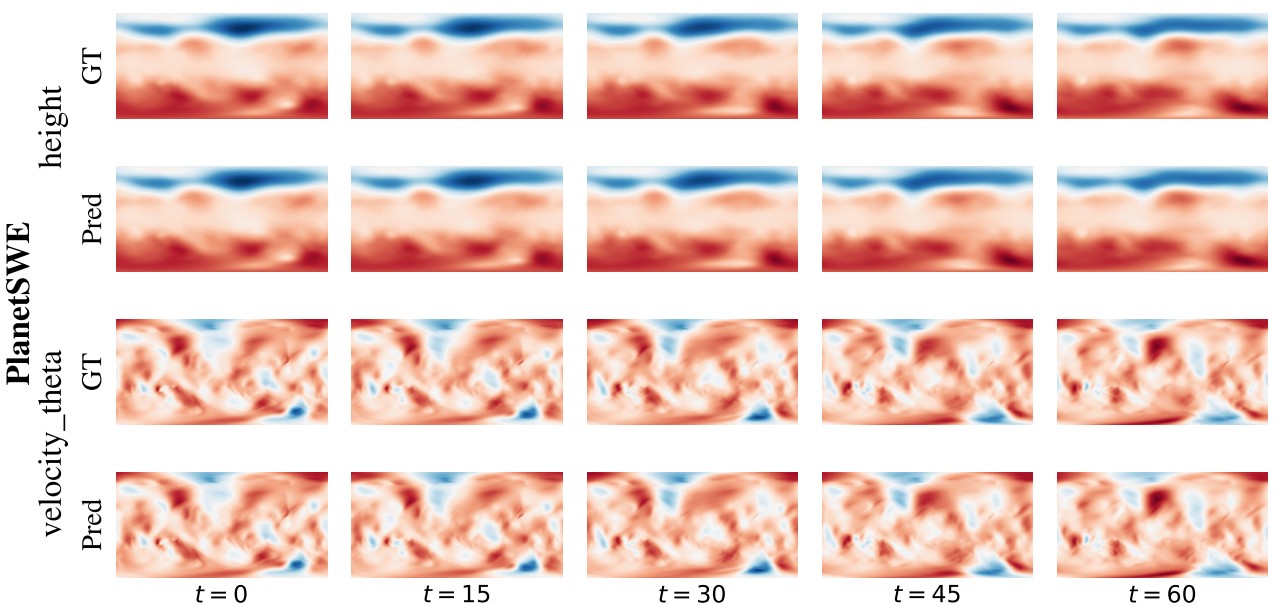

*Figure 29.* Shallow water equations over approximate earth topography (PlanetSWE).

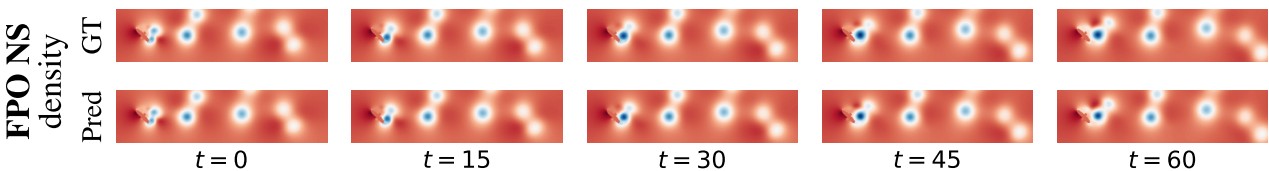

*Figure 30.* Flow around complex obstacle defined by skelenton in 2D.

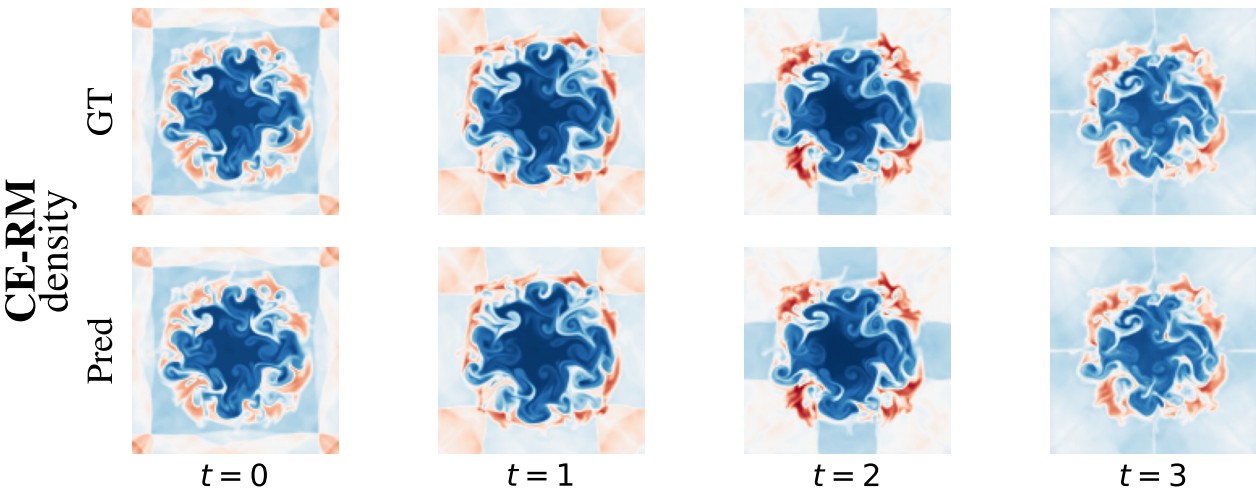

*Figure 31.* Compressible Euler equations initialized at Richtmyer-Meshkov instability in 2D. From PDEGym.

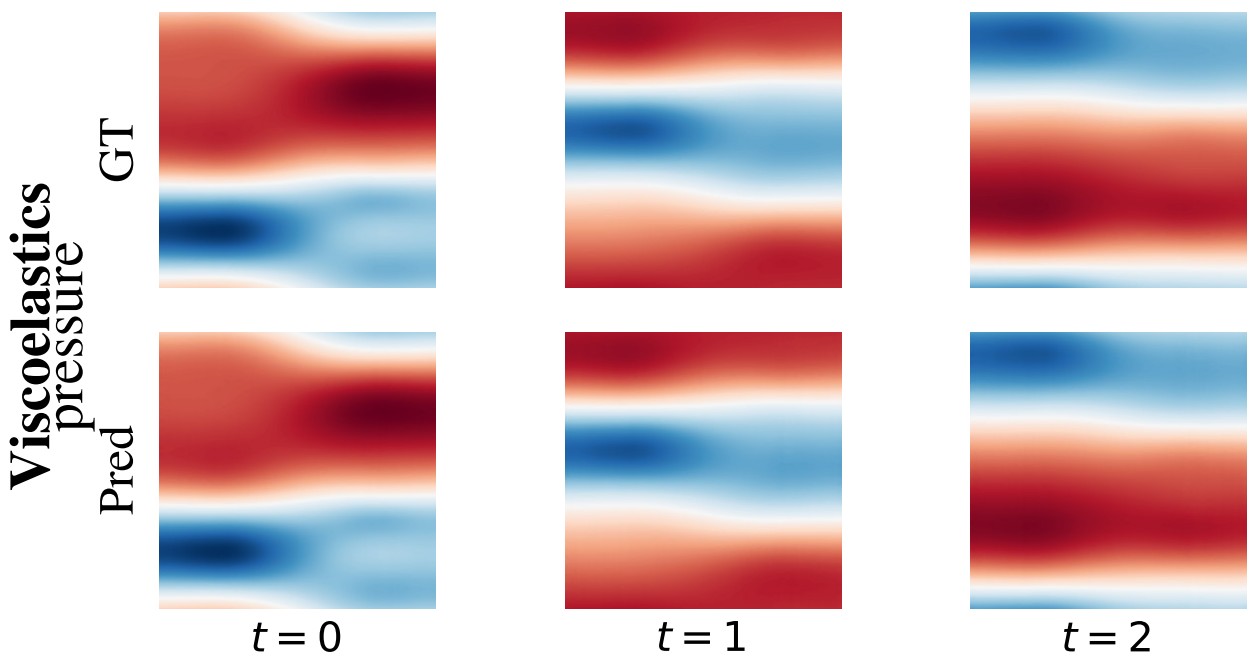

*Figure 32.* Instability in viscoelastic flow in 2D.

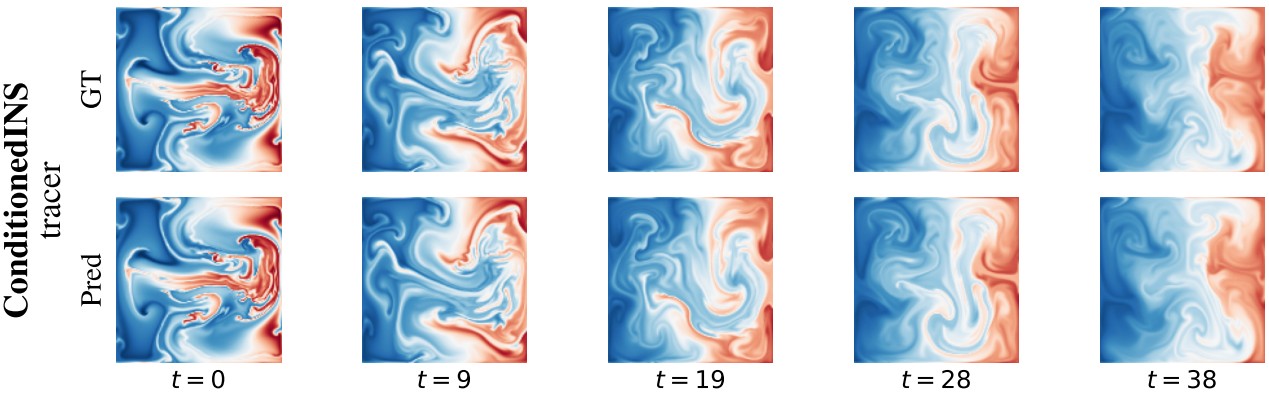

*Figure 33.* Particles embedded in buoyant incompressible flow in 2D. From PDEArena.

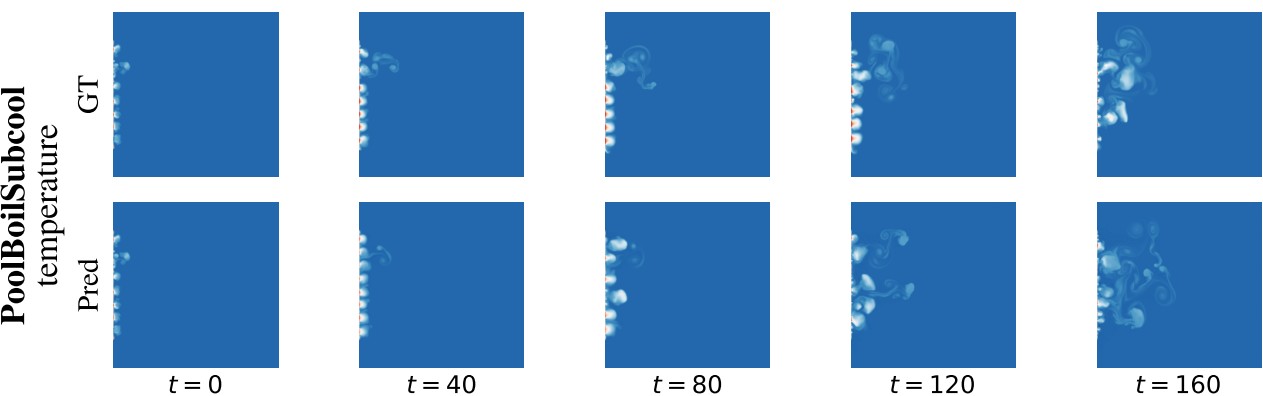

*Figure 34.* Multi-phase flow in which varying liquids are heated from below to form bubbles in 2D. From BubbleML 2.0.

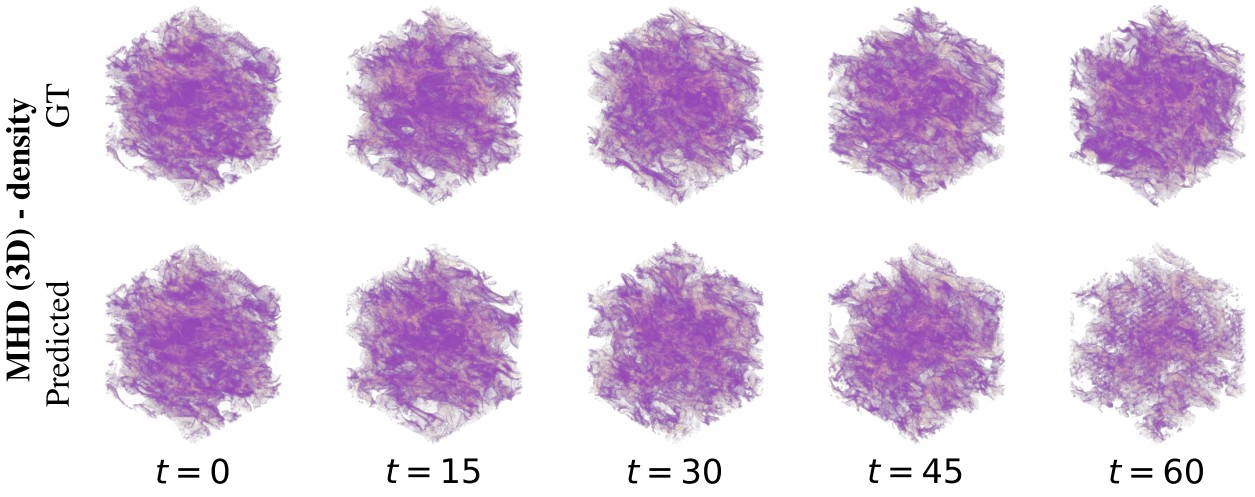

*Figure 35.* Magnetohydrodynamic turbulence in 3D.

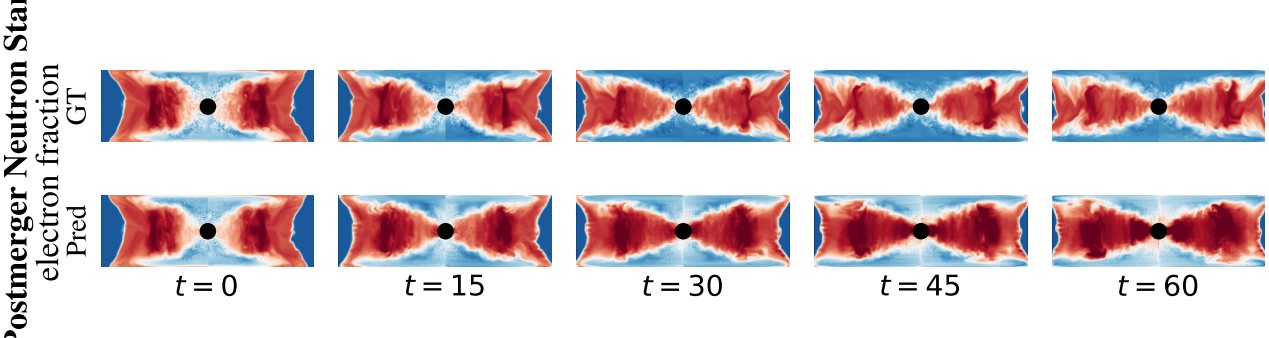

*Figure 36.* Aftermath of neutron star merger. Visualized slice in 2D, but simulated in 3D.

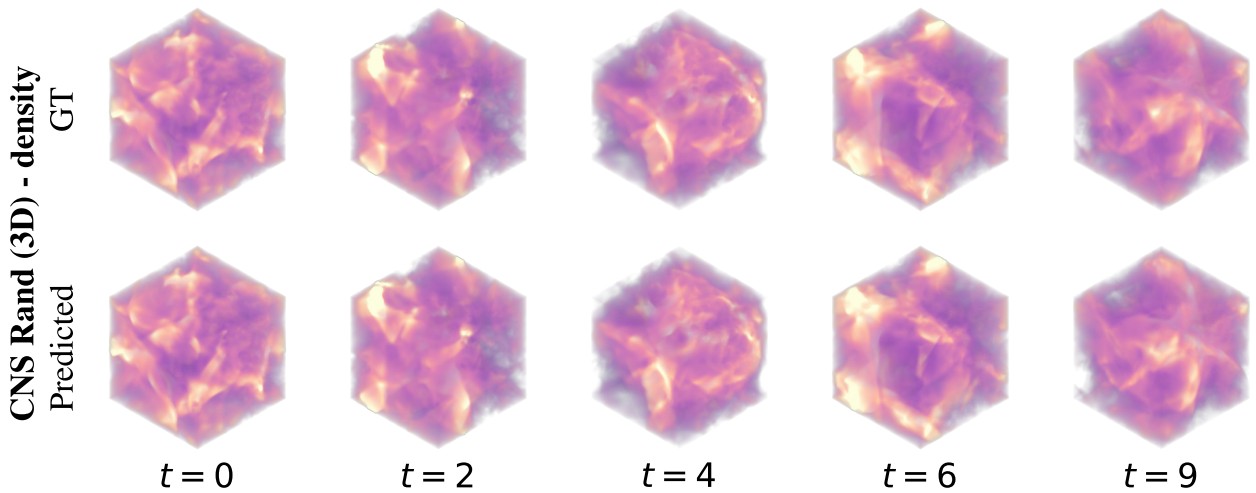

*Figure 37.* Compressible Navier-Stokes in 3D with "random" initial conditions. From PDEBench.

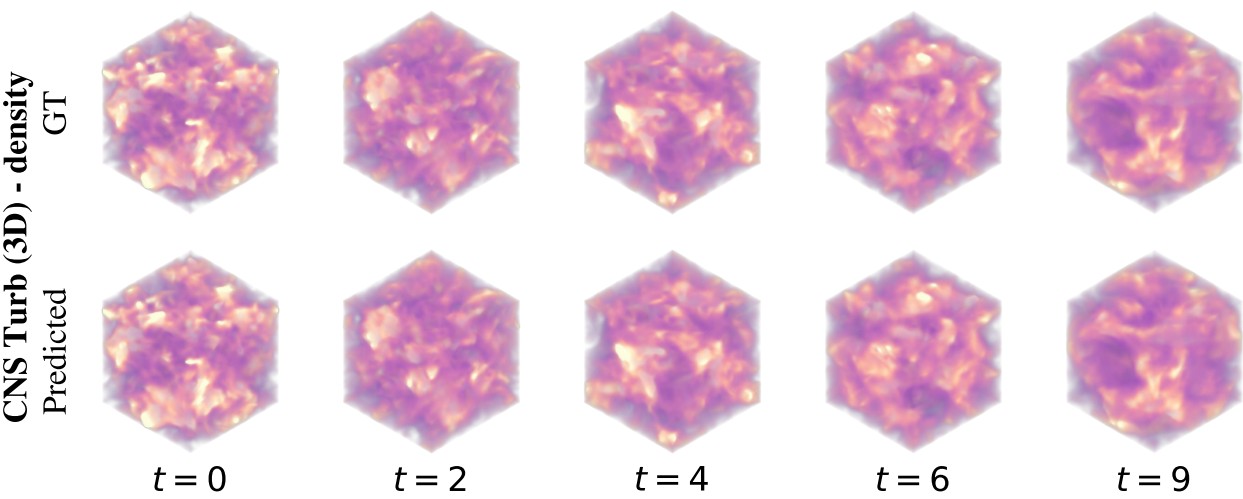

*Figure 38.* Compressible Navier-Stokes in 3D with "turbulent" initial conditions. From PDEBench.

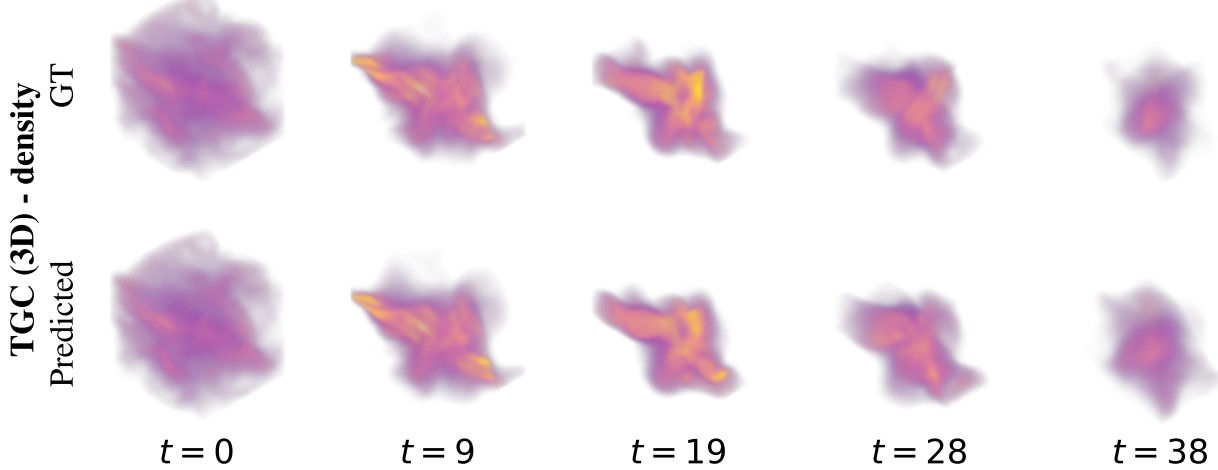

*Figure 39.* Cooling of the turbulent interstellar medium in 3D.

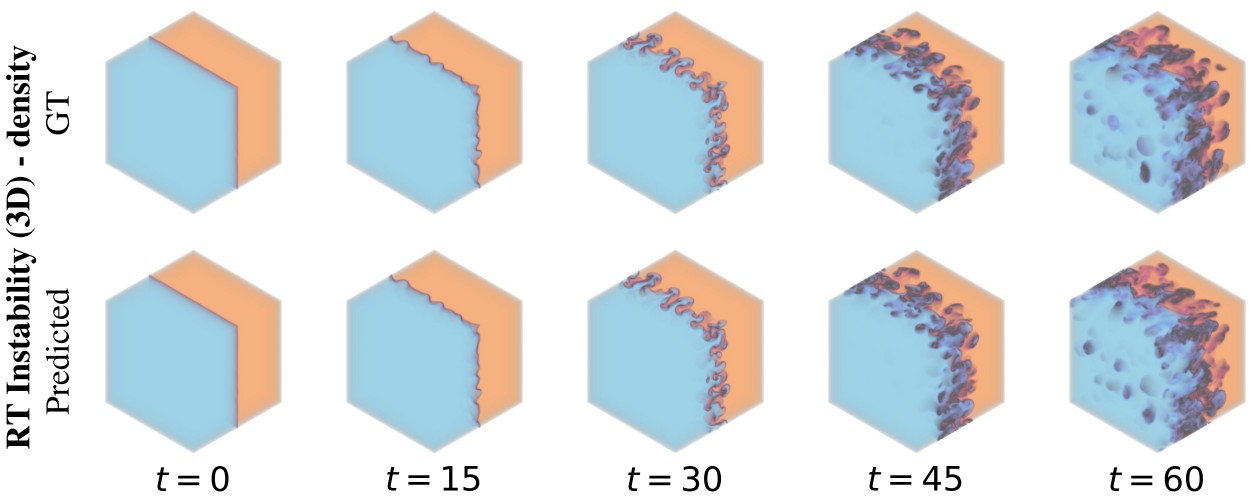

*Figure 40.* Rayleigh-Taylor Instability in 3D.

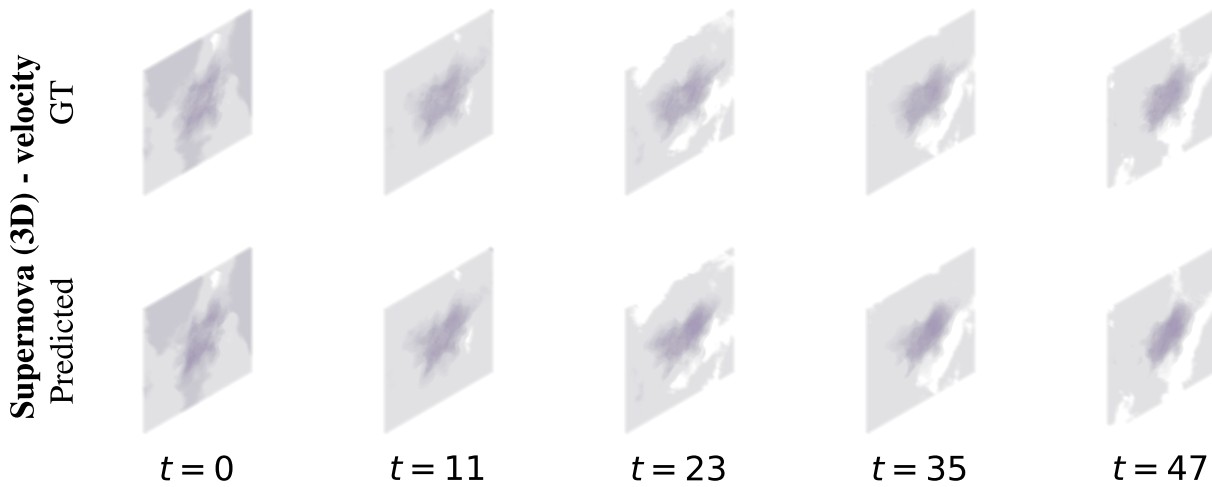

*Figure 41.* Supernova explosion in 3D.

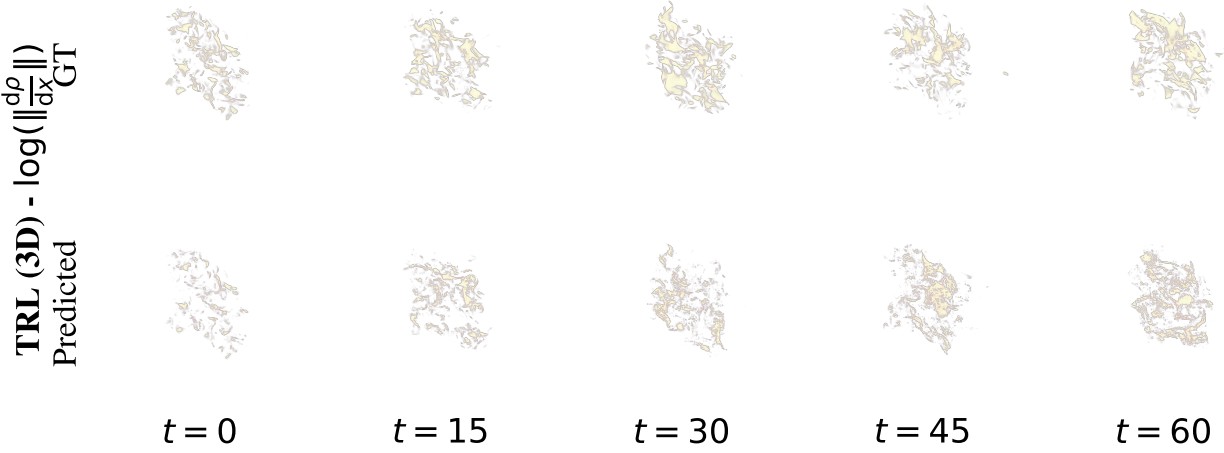

*Figure 42.* Turbulent radiative layer in 3D.

