# OpenReview forum: "Walrus: A Cross-domain Foundation Model for Continuum Dynamics"
_ICML.cc/2026/Conference — ICML 2026 spotlight_

### Official Review · Reviewer_sEh1 · 2026-02-20

**Soundness:** 2
**Presentation:** 3
**Significance:** 3
**Originality:** 2
**Overall Recommendation:** 4
**Confidence:** 4

**Summary:**

The authors position this paper as cross-domain foundation model for continuum dynamics and sort of claimed it as large-scale spatiotemporal emulator for heterogeneous 2D/3D, multi-res training and stabilize long rollouts. They have used diverse dataset (19) spanning astrophysics, geoscience, rheology, plasma physics, acoustics, and classical fluids with both 2D and 3D data and focussed on issues realted to enabling scalable, generalizable operator learning across diverse physical systems.

**Compliance With Llm Reviewing Policy:**

Affirmed.

**Key Questions For Authors:**

Same as Strengths And Weaknesses

**Limitations:**

Same as Strengths And Weaknesses

**Strengths And Weaknesses:**

Strengths:
proposed system is clearly defined as a sequence-to-sequence over spatiotemporal fields
empirical evaluation is extensive, covering multiple datasets
Patch jittering seems effective anti-aliasing trick
Reports consistent improvements over prior models
The paper is well written an I didnt find any glaring mistakes or presentation issues.

weakness:
limited analysis of out-of-domain generalization, bounding generalization errors across shifting topologies and extreme dynamical regimes

can you please create clear seggregation between Cross-domain and multiple dataset, currently appears closer to multi-task interpolation than true operator generalization

The whole performance os based on VRMSE, and limitations are qualitative, wouldnt it be better to show actually the performace also based on physical constraint?

I think you should also incude more ablations to understand the effects of scale, data, and architecture - But if it will take a lot of effort and run may be provide a theoritical justification, that should be fine as well.

can you please elaborate how its generalised when architecture does not hold any functional structure? like linear/non-linear/equivariance/etc.

No analysis is provided for error accumulation in autoregressive rollouts, especially in chaotic systems.

I feel like deterministic training objective is insufficient.


I have mixed feeling for the paper rating - probably would be more clear after discussion

---

> ### Author Rebuttal · Authors · 2026-03-31
>
> We thank the reviewer for their thoughtful feedback and kind words on the writing and empirical evaluation. Due to length restrictions, we group related points below and apologize for brevity.
>
> ---
>
> **Generalization and Cross-domain vs. Multi-task**
>
> G1 - Cross-domain and multiple dataset
>
> This is an interesting point, but we'd argue that the breadth of challenges here clearly covers multiple scientific domains. Each of the subheadings in Table 1 represent distinct research communities with their own tooling and norms. The downstream tasks in Section 5.1 involve new equations, boundary conditions, geometries, and regimes never seen during pretraining. The ablation in Section 5.3 (Figure 5, bottom row) shows pretraining advantages on new physics even in few-shot settings.
>
> If the concern is a lack of zero-shot extrapolation, we agree that this is not demonstrated. No foundation model in this space has demonstrated usable zero-shot performance on non-trivial tasks to date and we’re not claiming to have addressed this challenge. But this doesn’t mean progress isn’t being made. As noted in our response to gKDp, fine-tuning is standard even in vision and language.
>
> G2 - Generalization without functional structure
>
> The goal of foundation models is to imbue a large-scale model with the correct inductive bias through the training data rather than through modeling choices. This is why transformers have become so dominant in space: they’re flexible and hardware efficient. That said, there are a number of inductive biases still baked into the model. RoPE gives the spatial attention blocks a relative position bias. This means the attention operation is biased to be translation equivariant up to boundary effects. To a degree, patch jittering can also be seen as an approach for recovering some of the translation equivariance lost from downsampling in the encoder.
>
> ---
>
> **Evaluation and Metrics**
>
> E1 - Physical constraints beyond VRMSE
>
> VRMSE was chosen as it enables cross-dataset comparison and is the established Well benchmark metric. We agree physics-informed metrics would provide complementary insight. However, this is very domain specific and becomes unwieldy at the scale of 26 datasets. VRMSE is applicable to all of these systems at the time horizons explored and the normalization provides a simple point of reference for performance.
>
> We want to make the scope of our claims clear. What we’re describing in line 301R is that while the metrics chosen indicate that Walrus is a more accurate emulator than prior foundation models in this space, we do not claim that this performance has “solved” these problems for domain scientists. That requires more dedicated analysis based on subject matter expertise which often cannot be cleanly defined as a small set of loss functions.
>
> E2 - Error accumulation in autoregressive rollouts
>
> Autoregressive error accumulation is a major topic in the paper. Section 3.2 proposes patch jittering, an approach for mitigating autoregressive error accumulation with both theoretical and empirical evidence showing it reduces median long-horizon VRMSE by 54% and unstable rollouts from 10/19 to 3/19 datasets. The primary experiments show error across multiple time scales and this is covered more thoroughly in Appendix Figures 12-13 which show VRMSE over every rollout step with error bands. For chaotic systems, pointwise error growth is inevitable regardless of model quality; what matters is stability and physical plausibility, which we demonstrate in the rollout examples (Appendix G).
>
> ---
>
> **Ablations and Training**
>
> AT1 - Scale, data, and architecture ablations
>
> Section 5.3 directly ablates several of our claimed contributions via the HalfWalrus experiments. For architecture, we compare against fundamentally different backbones (AFNO in DPOT, SWIN in Poseidon, Axial Attention in MPP) in 5.1 and 5.2, providing indirect architectural comparison. Jitter is ablated in A.1. Our distribution strategy is ablated in figure 3.
>
> Transformer scaling with data and compute is well documented and we do not believe repeating this adds value. Studies on data composition or scaling dimensions for physical inference are interesting but typically require dedicated submissions to do well and so are out of scope for this paper.
>
> AT2 - Deterministic training objective
>
> We share this intuition and discuss it explicitly in Section 6 (Limitations). Latent diffusion approaches (Rozet et al., 2025) show promise for stable rollouts with probabilistic estimates. Generalizing these to the multi-physics setting is an active area we're excited about, but we believe the deterministic foundation here is a necessary first step that future probabilistic methods can build upon.
>
> ---
>
> Planned revisions: We will extend discussion of physics-based evaluation metrics and their domain specific challenges as well as make the generalization claims more concrete. We hope these responses address the reviewer's concerns.

---

> > ### Author Rebuttal · Reviewer_sEh1 · 2026-04-03
> >
> > My concerns have been adequately addressed

---

> > > ### Author Response · Authors · 2026-04-07
> > >
> > > Thank you for acknowledging our rebuttals. We would just ask that if we were able to address your concerns, you consider raising your score.

---

### Official Review · Reviewer_yTH7 · 2026-03-05

**Soundness:** 3
**Presentation:** 3
**Significance:** 4
**Originality:** 3
**Overall Recommendation:** 5
**Confidence:** 4

**Summary:**

The authors present Walrus, a 1.3B parameter cross-domain foundation model for continuum dynamics. The model architecture interleaves spatial and temporal processing blocks and employs several techniques to handle a variety of data shapes, dimensionalities, and resolutions. Walrus builds on previous contributions to the field of scientific machine learning at a large scale. It also introduced methodological contributions, "patch jittering," to mitigate tokenizer-induced chessboard or grid-pattern artifacts, a normalized training objective, and adaptive-compute tokenization as a combination of approaches from CSM, MPP, and hMLP. The model is trained on 19 diverse scenarios and evaluated against other foundation models. Across a diverse set of tasks and 56/65 metrics, Walrus is able to achieve state of the art results.

**Compliance With Llm Reviewing Policy:**

Affirmed.

**Final Justification:**

See rebuttal acknowledgment.

**Key Questions For Authors:**

1. What exactly does the topology-aware sampling sample? Does it operate on the input grid points, the temporal states, or the latent tokens?
2. Temporal Interpolation: How does the model perform when queried to predict time steps in between those it was trained on (i.e., temporal super-resolution)?
3. How many previous timesteps are consumed as context for each forward prediction? Can the authors provide a table detailing the number of steps used in each problem, if they are different?
4. What is the actual inference speed of Walrus? The text mentions that CSM allows users to "trade accuracy for speed dynamically," but concrete latency metrics are needed to substantiate this claim.
5. The downsampling/upsampling factors are modulated at runtime to maximize token usage. Is this restricted to dyadic/power-of-two input sizes, or can it genuinely handle arbitrary, non-standard resolution grids? I see that post_neutron_star_merger uses 192x129x66, so it must, but it is not clear to me how this works. Can you also clarify the exact mechanisms of hMLP and MPP in this context?
6. Walrus predicts the difference between inputs and outputs rather than the absolute output state. What was the rationale behind predicting this output as opposed to predicting the field itself?
7. When augmenting 2D data for 3D processing, why is zero-padding used instead of periodic repetition or boundary clamping? Has an ablation been performed to justify this choice? Also, is there any study with and without the random rotations? Even if the authors have some general understanding which has not been fully quantified, I think this would be valuable discussion.
8. Are the randomly sampled time strides mathematically or functionally analogous to the "all2all" training methodology utilized in Poseidon? I.e. do they provide quadratic growth in the number of possible training samples?
9. How exactly is the prediction lead-time incorporated into the model? Is it concatenated as a feature, injected via positional encoding, or used to modulate the normalization layers? Or, is this learned implicitly based on the given trajectory?
10. How does the model behave if only 1 initial condition (not an initial trajectory) is provided? Does the model support a "zero-history" mode, or must the history be padded/simulated? Table 1 discusses "one-step" but as far as I can tell, this term is never defined in the manuscript: is this the model performance with only 1 input timestep, or is this simply the performance of each model taking the true previous timestep? If you feed predictions back autoregressively in this low-context regime, how quickly does the rollout degrade? This could be done for the experiment in Figure 11, but other (more challenging) problems for a direct comparison with Poseidon are also requested.
11. Most existing state-of-the-art models for continuum dynamics (like Poseidon) function effectively with a single snapshot. Could the authors provide a quantitative study showing that the accuracy gain as a function of the number of context snapshots? This would be helpful to understand more fundamental questions, e.g. at what point do we see diminishing returns in accuracy?
12. Can the authors provide a full table of results for the HalfWalrus and Naive models, for all experiments in Table 1? Also, is the Naive model named so, only because it does not use 3D data?
13. Can the authors provide a more clear motivation for the non-Markovian setup of this work? I feel that this problem setting is not fully motivated in the manuscript.
14. Out of curiosity, is there any particular reason for the name Walrus?

I would be happy to raise my score if the authors address my questions and concerns, particularly those asking for more quantitative investigation and more detailed studies of the ablative results.

**Limitations:**

I feel the limitations due to the non-Markovian architecture have not been fully addressed in this work. The authors highlight many advantages, but it remains unclear how Walrus performs in a Markovian setting, and this deserves investigation.

**Strengths And Weaknesses:**

**Strengths**
- The authors train and evaluate on a very broad set of PDEs in both training and inference. This also includes problems which exhibit very challenging or chaotic dynamics, posing open problems for scientific machine learning.
- For comparisons with other foundation models, the authors provide fair comparisons in terms of the data processed by each model. For in-distribution tasks, other foundation models are trained on the same number of trajectories as Walrus has seen during both pre-training and fine-tuning.
- Patch jittering, while not theoretically complex, provides a simple and effective solution to a known problem in vision transformers.
- From a systems and engineering perspective, Walrus constitutes a significant undertaking. The distributed training of such a large model on a variety of both 2D and 3D data helps to push forward the state of the art.

**Weaknesses**
1. Operational rigidity (Non-Markovian Construction). Unlike many competing foundation models and neural operators, which are Markovian and only require a single state to predict the future, Walrus requires a sequence of historical snapshots. The authors provide some discussion on this in the appendix, demonstrating that this improves results, especially in scenarios with uncertainty in parameters. I recognize that the authors are presenting a new approach for *continuum dynamics* where multiple previous snapshots are often available, but this requirement nonetheless hinders the generality of the approach.

1.a Initialization Constraint. As a result, it seems that Walrus cannot be used to start a simulation from a single physical initial condition without a "warm-up" period or synthetic historical data.

1.b Data Overhead. The requirement quadruples (assuming 4 previous snapshots) the data that must be held in memory during each training or inference step, which may offset any speed advantages and increase training times.

1.c Historical Sensitivity. In real-world scenarios where historical data may be sparse, noisy, or sampled at inconsistent intervals, the model's reliance on a precise temporal context sequence could possibly lead to brittleness that single-snapshot models can avoid.

1.d There is no clear ablation on Walrus to demonstrate how much performance improves by the introduction of additional snapshots in the context trajectory.

1.e In its current form, the manuscript does not adequately explain how many time steps are used for each problem. Likewise, these details remain sparse from the experimental setup and discussion, which make it difficult to evaluate the fairness of the comparisons with Markovian baselines.

2. Metrics. I find VRMSE to be a good choice for many problems; however, for turbulent problems (especially in 3D) this metric provides little to no information which helps the audience to distinguish performance after predictions have diverged from the true solution. There is still value in statistical solutions and measuring how well features of the solutions are maintained by the model, besides L1 or L2 norms. The paper "Generative AI for fast and accurate statistical computation of fluids" (Raonic et al., arxiv 2409.18359v2) provides a great deal of discussion on metrics which are suitable for the turbulent dynamics exhibited in many problems. Currently, a more descriptive set of metrics is lacking from this work.

3. Baseline Comparisons. Walrus is not only significantly larger that the baselines (e.g. 1.3B vs 628M parameters for Poseidon-L), but is also pretrained on a much larger scale. While I don't deny that the large-scale training is a strength, I do believe that the authors have not provided the results to make it clear whether the improved performance comes as a result of algorithmic contributions or is simply a result of scaling up. The authors provide several ablations which modify the training data inputs and total number of parameters. These are critical to understanding the behavior of the model, and detailed results should be added to the main table for full context.

4. Lack of Scaling Laws. For foundation models in particular, scaling behaviors are a vital and necessary component of the evaluation. Some initial work has been done (again in the ablations mentioned in my previous point), but has not been full fleshed-out. I would request the authors to refer to Poseidon for a deeper look into their scaling studies, and request a more detailed evaluation in this area, in particular the scaling with respect to model size, pretraining dataset size.

5. Restriction to Euclidean Grids. The model relies on Euclidean grids (or is exclusively evaluated on them). Many large-scale and high-fidelity scientific simulations rely on unstructured meshes and complex geometries. I see much work on foundation models moving toward this direction, but in its current form, the paper does not make it clear how these inputs may be reconciled with the proposed approach.

6. Limited Algorithmic Novelty and Lack of Clarity. Beyond patch jittering, the methodological contributions are somewhat incremental. The architecture seems to be largely a standard transformer coupled with existing compute-adaptive techniques and scaled up with more parameters and data. Furthermore, the authors rely heavily on other works, the details of which are not adequately explained within the manuscript. The need to refer to several references and publications to start understanding how Walrus actually functions makes the paper difficult to follow. I would ask the authors to revise the methods section and include more explanation on the underlying concepts they employ from works such as hMLP.


**Minor Comments**
-Line 301 reads "medium (1-10) and medium (11-20)" which I suspect is a typo and should be "short ... medium"
-Line 365 the reference to table 3 appears to be hardcoded, or at least the link to the table does not work.
-The author list for Llama 3 is exceptionally long. I believe this can be shortened within ICML guidelines.
-Table 7 says "pretraining dataset description" but I believe this is the fine-tuning dataset table.

---

> ### Author Rebuttal · Authors · 2026-03-31
>
> We thank the reviewer for their exceptionally detailed feedback. Due to length restrictions we group points and apologize for extreme brevity.
>
> **Non-Markovian Architecture (1, 1a-1e, 10/11)**
>
> Appendix E describes the trade-offs: Markovian models are more efficient but cannot reliably operate in imperfectly observed domains when unknowns cannot be inferred from initial conditions alone. This makes them well suited to simulation acceleration but less reliable for forecasting from observations or reanalysis. Most weather models (GraphCast, FuXi, FCN) use multiple history steps for this reason. We demonstrate concretely in Figure 11: on linear advection with unknown velocity, the Markovian model cannot make meaningful predictions while the non-M model infers the missing information.
>
> *Context length (1e):* 2D uses 6 frames, 3D uses 3 (Table 3). "One-step" means each model receives its full context of true states and predicts one step forward. Baselines receive native context lengths (MPP: 6, DPOT: 10, Poseidon: 1) as in C.3. We will clarify this in main text.
>
> *Initialization (1a):* Walrus can be finetuned as a Markovian model as the architecture is not dependent on fixed shape. However, most experiments here are in the incomplete information setting where this is shown to be inadvisable in Section E.
>
> *Historical sensitivity (1c):* Yes, if the historical data is flawed, this will introduce error. However, if one-step is insufficient for prediction as in App E, a source of error is preferable to a degenerate objective.
>
> *Context ablation (1d):* We agree this is valuable and will add results to Appendix E which is where this is explored.
>
> ---
>
> **Metrics (2)**
>
> We generally agree, though in this case we analyze within predictability limits. Figures 12/13 show few systems reach asymptotic error until near trajectory endpoints. For longer evaluations of turbulent systems, statistical metrics as in Raonic et al. would be a valuable addition.
>
> ---
>
> **Baseline Comparisons and Scaling (3, 4)**
>
> Sections 5.1/5.2 compare leading foundation models as a downstream user would encounter them, all using the largest public checkpoints. DPOT-H is comparable in size (1.15B vs 1.3B) and Walrus still substantially outperforms it. Sections 5.3, A.1.3, and Figure 3 analyze our specific contributions through controlled study.
>
> On scaling laws: transformer scaling with data and parameters is extensively documented. If this were an initial proposal for physics foundation models, confirming scaling properties would be novel, but that foundational work has been established. We believe to say something new, we need more targeted studies on data composition and scaling dimensions specific to physical inference, but these warrant dedicated investigation.
>
> ---
>
> **Euclidean Grids (5)**
>
> Correct, discussed as a key limitation in Section 6. Extending to unstructured meshes while maintaining efficiency is a natural future direction.
>
> ---
>
> **Novelty and Clarity (6)**
>
> Patch jittering provides a theoretically grounded solution with strong empirical impact (54% VRMSE reduction). The engineering contributions yielding 262% throughput improvement are also non-trivial. Other choices synthesize existing tools, but demonstrating that this integration enables a qualitative leap in cross-domain performance is itself a contribution. We will improve self-containedness by adding hMLP to appendix and fixing link for CSM.
>
> ---
>
> **Rapid Remaining Key Questions**
>
> - 1 - Refers to network topology. All GPUs in a sharding group sample the same dataset to avoid FSDP deadweight loss. Justification in A.2.
>
> - 2 - This is a discrete-time model; temporal super-resolution is not explored.
>
> - 4 - Will add latency numbers. CSM adjusts token count, deterministically changing FLOPs.
>
> - 5 - Not limited to powers of two, but our filter sizes (8,4) constrain practical downsampling ratios. PNS's awkward dimension (66) is resized to 64 by interpolation (D.1).
>
> - 6 - Heuristic from early experiments, inspired by GraphCast's similar approach.
>
> - 7 - Applies to tensor-valued fields, not boundaries. The new boundary is periodic, which we will clarify in text.
>
> - 8 - These differ substantially due to discrete-time autoregressive vs continuous-time large-step prediction.
>
> - 9 -  Learned implicitly from trajectory context. Would need to add to handle non-uniform history.
>
> - 12 - 5.3 ablates training strategy at controlled scale. Applying HalfWalrus to Table 1 tasks would conflate training strategy effects with scale and data volume differences. The controlled setting in 5.3 is chosen to isolate these factors.
>
> - 14- They are one of the keystone species of the Arctic Ocean.
>
> ---
>
> **Minor corrections:** Thanks. Will fix L301 typo, Table 3 reference, and Table 7 caption.
>
> **Planned revisions:** Clarify context definitions, add latency numbers, add context-length ablation to Appendix E, and add hMLP to appendix and fix self-link for CSM. We hope these address the reviewer's concerns.

---

> > ### Author Rebuttal · Reviewer_yTH7 · 2026-04-02
> >
> > I would like to thank the authors for the exceptionally thorough rebuttal. I appreciate the effort to concisely answer my extensive line of questions. The rebuttal clarifies several of my core methodological concerns. As I stated before, I will raise my score to 5 (accept).
> >
> > However, two important methodological caveats have become clear based on your responses. I would deeply appreciate if the authors consider these two points for the final manuscript.
> >
> > * **Baseline Fairness Regarding Context Length** I understand the reasoning behind using the "native" context length, but I think we must consider how I practitioner might apply any particular model in a real-world setting. It would be straightforward to fix these to be the same across all models. Poseidon in particular seems to be at a significant disadvantage with a context length of only 1. The channel projection encoder and decoder are usually trained from scratch for this model. A straightforward modification would be to take all 6 input time steps and simply concatenate them along the channel dimension. Let me be clear as well -- I do not disagree with the authors that the non-Markovian setting does not merit investigation. In fact, it is the more intuitive approach for many engineering / scientific simulations, especially those under missing information. Weather is a very prominent example. Likewise, it is not that I suspect Poseidon will suddenly match the performance of Walrus with the larger context window. It is simply that I feel this topic was not discussed in sufficient detail in the current manuscript. My recommendation to the authors is to highlight this discussion more explicitly in the manuscript, and additionally, to provide a small scale experiment where the number of input trajectories is fixed among all models. I believe this discussion would be beneficial for both those of us who study these models in more applied settings, as well as those who work on the development of large-scale foundation models.
> >
> > * **Metrics for Long Rollouts** This point is not sufficiently clear to me. The response seems to contradict itself. On the one hand, the rollouts in these experiments are significantly longer than those of previous works. On the other hand, the authors state that errors are primarily measured before asymptotic error is reached. Judging from the figures for many examples in the appendix, the latest time steps show obvious divergence in overall dynamics, although they may represent probabilistic solutions to the equations. I see the lack of smoothing in the long-rollout case to be a *possibly* **very valuable** contribution, but the evidence to support this is currently lacking. As the authors, you are in the best position to simply compute these metrics over the predictions at the last timestep. I reiterate, using a probabilistic metric in this evaluation would be valuable.
> >
> > Assuming the promised additions are integrated into the CRV, I believe Walrus is a strong contribution to the field. I am raising my score accordingly, and I sincerely hope that the authors consider the two points listed above for the final manuscript.

---

> > > ### Author Response · Authors · 2026-04-07
> > >
> > > Thanks for the insightful followup and for the kind comments. We have a bit more space here, so we can go into more detail on these follow-ups. These are both good points and we'd like to address them to the extent we believe is possible with this submission.
> > >
> > > **Baseline Fairness Regarding Context Length** - Generally this is something that we agree on in principle but may disagree on implementation. In general, what we think is viable and will commit to adding is additional context length ablation in Appendix E (where we focus on the Markovian discussion) that shows that Walrus also cannot solve the toy problem without history.
> > >
> > > The reason for this is that we feel the choice of how to extend Markovian models to non-Markovian settings introduces new degrees of freedom and results would then be very dependent on how we chose to modify the base architecture. While stacking the timesteps as additional channels is indeed an option, this still isn't entirely fair given Poseidon was not pretrained to infer dynamics from varying timesteps and the other models use more complex mechanisms. Walrus and MPP use temporal attention while DPOT uses a fixed context length downsampling block. We agree Poseidon would likely perform better, but releasing such numbers would be interpreted as experimental evidence of Poseidon's capabilities in the non-Markovian setting. Given we think there is a lot of space for exploration there, we are not comfortable making such claims even implicitly. Instead, the current submission shows results with the standard model and explains the gap in what's possible with and without history. As we mentioned before, we will update this section to also go into more detail on the advantages of this approach as well.
> > >
> > > On what we can do - since space and time operations are factorized in Walrus and temporal mixing is done via attention, Walrus is able to adjust the history length during finetuning without any architectural changes. This isn't an entirely fair comparison either for general purposes as Walrus was pretrained to infer dynamics in-context from history which wouldn't be available in history-free finetuning, but this makes it possible to show that the simple missing information task in Appendix E is impossible for Walrus without history as well and that this isn't a statement about any particular architecture.
> > >
> > > **Metrics for Long Rollouts** - There are two points we want to clarify here before going into general discussion:
> > > 1. We do not make the claim that Walrus has reduced smoothing. Our claims of improved stability rest on a decrease in aliasing-induced divergence. As with most ML approaches, we feel qualitative evaluation of solutions still shows some level of smoothing compared to the numerical ground truth. This is visible in the appendix rollouts in many of the turbulent mixing problems (RB, RTI). We agree with the common hypothesis that explicit probabilistic modeling is likely the most promising approach to date for reducing smoothing effects, though that is not explored here.
> > > 2. What we mean by "within the limits of predictability" is that it does not yet appear that most of these losses have plateaued to the point where it is impossible to consistently outperform predicting a random state from the true distribution over states. For chaotic systems reaching a stationary distribution specifically, minor perturbations lead to pointwise RMSE plateauing around $\sqrt 2 \sigma$ where $\sigma$ is the standard deviation of the distribution of field values so de-correlated trajectories reaching stationarity will typically end up around this value.
> > >
> > > Generally the datasets used are a mix of stationary systems and "transitional" systems. For stationary systems (RB after initial mixing, MHD, AM after initial mixing), we completely agree on the value of distributional metrics and their necessity once pointwise relationships decorrelate. The "transitional" systems (RTI, acoustics, SPN) are ones whose initial conditions are either perturbations of metastable states or systems with an initial forcing. Generally these would eventually converge to a new equilibrium, but would require significantly longer burn-in times, so currently time-averaged distributional metrics are heavily dependent on which windows are being averaged.
> > >
> > > Based on this discussion, we do plan to add the averaged W1 suggested in Raonic et al computed over time rather than samples to the appendix. For stationary systems, this would be similar for sufficiently long rollouts due to ergodicity and remains well-defined and captures a quantity of interest for non-stationary systems. We believe it provides interesting complementary analysis to the pointwise metrics currently included.
> > >
> > > ---
> > >
> > > Again, we'd like to thank you for the detailed review and discussion. It's really made the paper stronger.

---

### Official Review · Reviewer_tGkq · 2026-03-10

**Soundness:** 3
**Presentation:** 3
**Significance:** 2
**Originality:** 3
**Overall Recommendation:** 5
**Confidence:** 4

**Summary:**

This paper addresses a long-standing challenge in machine learning–based physical simulation: maintaining data consistency and spatiotemporal continuity across both 2D and 3D problems. The authors propose **WALRUS**, a Transformer-based architecture that leverages spatiotemporal attention mechanisms to learn from heterogeneous 2D and 3D data across diverse physical scenarios.

The framework integrates several technical components, including Patch Jittering, 2D-to-3D augmentation, Adaptive-Compute Tokenization, and Topology-Aware Sampling, aiming to improve robustness and generalization across varying geometries and resolutions.

Extensive experiments are conducted on more than ten distinct physical simulation settings, accompanied by detailed ablation studies. The empirical results demonstrate consistent performance gains over baseline methods, supporting the effectiveness of the proposed design choices.

**Compliance With Llm Reviewing Policy:**

Affirmed.

**Final Justification:**

The paper’s main strength is its empirical performance: the experiments are broad, convincing, and clearly presented, which makes the work sound and significant.

My main concern was originality. The rebuttal mainly improved my view on this point by clarifying which parts are novel, which are integrative, and why patch jittering is a meaningful contribution in this setting. I still see the paper’s strongest aspect as its experiments, but the rebuttal addressed my main concern sufficiently and led me to raise my score from 4 to 5.

**Key Questions For Authors:**

1.What do the authors consider to be the primary downstream tasks enabled by WALRUS? Beyond accurate reconstruction of 2D and 3D physical fields, it would be helpful to understand the intended practical use cases of the model. For example, could it be applied to surrogate modeling for accelerated simulation, inverse problems, real-time control, design optimization, or data assimilation? A clearer discussion of concrete application scenarios would better position the contribution within the broader landscape of scientific machine learning.

2.How do the authors characterize the principal methodological contribution of this work? Given that several components (e.g., Patch Jittering, augmentation strategies, adaptive tokenization mechanisms) have precedents in related domains, it would be valuable to clarify whether the novelty lies primarily in the architectural design, the theoretical framing, the integration strategy, or the empirical validation across heterogeneous physical settings. Explicitly articulating this would help distinguish the work from being perceived as a combination of existing techniques and better highlight its conceptual contributions.

**Limitations:**

yes

**Strengths And Weaknesses:**

Strengths:

 1.The experimental section is thorough and well-structured, including extensive analysis and detailed ablation studies that systematically evaluate the contribution of each component of the framework. The architectural diagrams and qualitative result visualizations are clear and informative, facilitating understanding of the proposed method. Additionally, the paper provides formal proofs for the stated lemmas, which strengthens the theoretical rigor of the work.

2.The simulation tasks considered are representative and practically relevant within the domain of physical modeling. The datasets span a diverse range of physical scenarios, and the empirical results demonstrate that the proposed approach achieves consistent performance improvements across different domains. This breadth of evaluation enhances the credibility and general applicability of the method.

Weaknesses:

1.Several core components of the framework—such as Patch Jittering and related augmentation strategies—have been explored in other domains. While their integration into a unified framework for physical simulation is valuable, the overall contribution appears to be more of a synthesis of existing techniques rather than the introduction of fundamentally new methodological insights. This somewhat limits the perceived level of innovation.

2.The theoretical analysis associated with the Proposition establishes the expectation behavior of patch jittering, demonstrating aliasing cancellation in expectation. However, it does not provide a corresponding variance analysis. Since the argument relies on expectation-based cancellation, a discussion or bound on the variance would strengthen the theoretical guarantees and provide a more complete understanding of the stochastic behavior of the method.

---

> ### Author Rebuttal · Authors · 2026-03-30
>
> First off, we’d like to thank the reviewer for their time and effort in reviewing our work. The reviewer made some strong points and we hope to use them to strengthen the submission. Due to the length limitations in the rebuttal process, we’re going to group what we felt were related points for response.
>
> ---
>
> _Downstream tasks_
>
> Walrus is a foundation for training general purpose surrogate models for time-varying physical phenomena. Once the model is finetuned, it can be used for any task one would normally use a differentiable surrogate for. Either the forward problem of accelerating simulation or various inverse problems like assimilation, parameter inference, or design, though in design the task must be able to be parameterized in a way that the model can ingest. That said, as the paper focusing on the quality of the surrogate itself, the primary concrete examples are focused on acceleration. For instance, if we look at two downstream tasks, RSG convective envelope and Post neutron star merger, both modeling enormously complex problems in astrophysics, the datasets used for generation in this case took several weeks to months to generate according to the Well documentation () while here the forecasts took several minutes.
>
> ---
>
> _Methodological Contributions and Novelty_
>
> Thank you for this comment. This is definitely an area where we can make things clearer.
>
> In general, this is a space where foundation model research is still fairly young and still faces a wide range of challenges. Our work focuses on a particular subset of these and then demonstrates how addressing these issues leads to improvements upon prior foundation models. Given Walrus is the leading model on 56/65 metrics on an extremely diverse evaluation suite, we'd argue that the impact is considerable.
>
> The two areas we really push on are pretraining diversity (and the accompanying engineering challenges) and stability. Some of these are addressed by synthesis of existing ideas, but others we would argue are addressed through novel solutions. We’d break them the relative as follows:
> - Novel: patch jittering, sampling strategy
> - Integration: adaptive tokenization, Asymmetric Input/Output Normalization, 2D->3D augmentation
> - Established: Other architectural components (hMLP, space/time attention components), variable time striding
>
> Jitter in particular we see as having a significant impact and novelty. Jitter is not simply using a random shift or crop augmentation during training. It uses independently sampled jitter factors at every autoregressive step to reduce the coherence of autoregressive error accumulation. To our knowledge, the only similar idea in the literature comes from DeepDream where this was used heuristically and in an extremely different context. To our knowledge, our work is both the first usage of a similar method in autoregressive generation and the first theoretical description of it.
>
> Our overall sampling strategy, on the other hand, is largely engineering and distributed system design, but efforts at this scale would not be possible without such contributions as we show that our approach led to a 262\% improvement over naive usage of torch FSDP. We believe these types of efficiency gains, particularly given the fact that the code is included in the submission, will also help other researchers in the field advance their own ideas.
>
> ---
>
> _Variance Reduction_
>
> We'd be happy to add this to the appendix. Since this boils down to averaging over an ensemble of independent samples, for k samples, the variance reduction is O(1/k). In the paper, since we found that 1-step with its negligible increase in cost works well in practice for mitigating long term error, we avoided focusing on asymptotics and instead focused on empirical analysis which shows the performance at the level we’re actually reaching in practice. (Appendix A.1.3)
>
> ---
>
> Once again, we’d like to thank you for the valuable discussion. If we’re able to address your concerns by including a broader discussion of specific applications, adding the variance bound, and clarifying which components are novel vs synthesized, we’d ask that you consider raising your score.

---

> > ### Author Rebuttal · Reviewer_tGkq · 2026-04-03
> >
> > My concerns have been adequately addressed. Adjusting my score accordingly.

---

> > > ### Author Response · Authors · 2026-04-07
> > >
> > > Thank you for the review and openness to discussion! If any other concerns pop up, please let us know and we'll do our best to address them either in text if this format allows for that or in future drafts if it does not.

---

### Official Review · Reviewer_gKDp · 2026-03-11

**Soundness:** 3
**Presentation:** 3
**Significance:** 3
**Originality:** 3
**Overall Recommendation:** 4
**Confidence:** 3

**Summary:**

The paper proposes large-scale pre-trained modeling for solving multiple continuum dynamics problems with a single pre-trained model. The paper aims to advance the neural solvers by addressing challenges in current approaches: limited to 2D problems, typically on fixed resolution, and homogeneous physics. To address these, the paper proposes several computation methods including patch jittering, 2D-to-3D augmentation, adaptive-compute tokenization, and topology-aware sampling. The model is tested on several cross-domain physics problems.

**Compliance With Llm Reviewing Policy:**

Affirmed.

**Final Justification:**

I thank the authors for the additional effort in addressing the concerns. I look forward to seeing these clarifications and revisions incorporated into the final version.

**Key Questions For Authors:**

1. Could the authors provide justifications on the design choices? for example, why RoPE (is this physically more meaningful?), T5-style temporal encoding, hMLP, etc?

2. It seems that the authors have some explanations on FNO/AFNO model's good performance on linear acoustics/wave propagation. Are there any expectation, e.g., Walrus would be expected to perform better on this type of physics problems?

3. Are there some studies/considerations on the pre-training methods? What's shown in the paper is the best set up found empirically?

**Limitations:**

The authors adequately discussed the limitations in Conclusion.

There is no expected negative societal impact.

**Strengths And Weaknesses:**

Strengths:

- Large-scale pretrained models: This is one of the important topics in surrogate modeling and the paper aims to provide an advanced method in this domain.

- Extensive experiments: The paper considers a set of diverse continuum dynamics problems ranging from biological/chemical problems to fluid problems.

- The paper also presents comparisons against the competing methods.

Weaknesses:

- Comments on experimentations: The paper is primarily empirical, and many of its claims rely on experimental results. As such, several questions regarding the experimental design, controls, and reporting need to be clarified to properly assess the strength of the conclusions.

    - Table 1 (and the text for Table 1) suggest that the overall trend is that the difference between Walrus and the best one among the baselines gets smaller as the prediction horizon gets longer. It would be important to examine what happens at even longer horizons (could the advantage of WALRUS persists or disappears?).

    - Relatedly, in 5.3. Impact of pretraining strategy, the impact is studied with one-step prediction only. Can the authors verify the same results would be expected for a longer prediction horizon?

    - Missing error bars or statistics? Are the reported results averaged over multiple runs with different random seeds, and if so, what is the variance across runs? Additionally, how sensitive is the model’s performance to initialization?

    - While the paper presents several ablations, it is unclear whether tensor-law–aware augmentation and variable time striding are ablated independently. If these components are important to the claimed performance gains, it would be helpful to isolate and quantify their individual contributions; otherwise, it remains unclear how much they actually matter.

    - Given that the paper positions WALRUS as a foundation model, one might expect to see clearer evidence of zero-shot adaptation/in-context learning/adaptation to distribution-shift capabilities. Without such evaluations, the use of the term “foundation model” may appear somewhat stronger than what is empirically demonstrated, since most reported gains rely on fine-tuning rather than direct transfer.

    - Since the baseline models are pretrained on different datasets, it would be helpful to clarify in what sense the comparisons are intended to be meaningful. Differences in pretraining corpora may influence downstream performance, so it would strengthen the evaluation to more explicitly discuss how this factor affects the interpretation of the results.

- About writing (not the major concern, but it feels strongly that there should be some improvements)

    - The paper is not fully self-contained. Key components such as compute-adaptive tokenization are referenced without sufficient high-level explanation. While not all details are needed, the paper should briefly (in high level) describe what this mechanism is and how it operates conceptually, rather than assuming prior familiarity.

     - Similarly, the description of topology-aware sampling is not sufficiently clear or self-contained. While the paper outlines the motivation and reports throughput gains, the conceptual mechanism and its practical implications are not explained at a high level, making it difficult for readers to fully understand how it differs from standard distributed sampling strategies.

---

> ### Author Rebuttal · Authors · 2026-03-30
>
> We thank the reviewer for their thorough and constructive feedback. Due to length restrictions, we group related points below.
>
> **Evaluation and Metrics**
>
> EM1 - Longer horizons / EM2 - One-step in 5.3
>
> Great question. We'd first note that our evaluation horizons (60-200 steps, Appendix F) are considerably longer than prior work (Poseidon: 20, MPP: 5). The narrowing gap is common for physical dynamics as early errors result in "correct" subsequent steps also changing. For chaotic systems, as we mention on line 304L, any two trajectories eventually decorrelate, so all effective models converge to similar loss. Section 5.3 focuses on one-step precisely to isolate the pretraining strategy from these convergence effects. This is the direct training objective and provides the cleanest signal for evaluating the optimization procedure.
>
> EM3 - Error bars
>
> Error statistics over test samples are included as shaded regions in Appendix Figures 12-13 and described in Appendix F. Multi-seed training runs are unfortunately infeasible at this scale, which is standard for foundation model research. We're happy to break main-text tables into expanded versions with per-sample statistics in the Appendix.
>
> ---
>
> **Training and Architectural Details**
>
> T1 - Augmentation ablations
>
> Good point, we should clarify why these are ablated together. Combined, these expose the model to both shifts in the visible scales and in the reference frame - two important physical dimensions. These are both important parts of the training pipeline and are ablated collectively in Section 5.3 via HalfWalrus vs. Naive, where augmentation robustly outperforms naive pretraining on both transformed and out-of-distribution tasks (Figure 5).
>
> T2 - Compute-adaptive tokenization / T3 - Topology-aware sampling
>
> We appreciate this feedback on clarity. Adaptive tokenization adjusts convolutional strides in order to control the tokenized resolution (details in Appendix B.2. We'll add the missing cross-reference). For topology-aware sampling, the core mechanism is: within each FSDP sharding group (1 node in our case), all ranks (GPUs) sample the same dataset (matching resolution/dimensionality to avoid AllGather bottlenecks), while individual groups sample independently to preserve diversity. By accumulating over multiple microbatches, we can further reduce the per-batch variance before AllReduce. We agree this deserves better exposition and will add an algorithm box to Appendix A.2.
>
> T4 - Design choices (RoPE, T5 RPE, hMLP)
>
> For components outside our specific contributions, we preferred established approaches. RoPE is generally the top performing relative position encoding in vision and language. T5-style RPE marginally outperformed on our short temporal dimension (3-6 steps) in early experiments, though not by a degree we believe to be meaningful. hMLP follows Touvron et al. (2022) and was effective in MPP (McCabe et al, 2023) which uses a similar overall structure.
>
> ---
>
> **Foundation Model Scope and Comparisons**
>
> F1 - Zero-shot performance
>
> We use "foundation model" per Bommasani et al. (2021): models trained on broad data at scale that can be adapted to downstream tasks. We're unaware of any physical emulation foundation model achieving usable zero-shot performance on non-trivial systems. We believe that would be an enormous milestone. Our experiments do demonstrate strong transfer: 63.6% average one-step loss reduction on held-out tasks (Section 5.1), and clear few-shot advantages from pretraining diversity (Figure 5).
>
> F2/F3 - Corpus composition and baseline fairness
>
> We agree corpus analysis is valuable but believe it warrants its own dedicated study, as is common in vision and language. Our comparisons address the practical question: how does WALRUS perform against the best available models? Importantly, baselines receive a considerable advantage: they get the combined pretraining + finetuning volume (4.5M for 2D, 2.5M for 3D) in the exact orientation and sampling rate used for evaluation, while WALRUS saw the pretraining data under heavy augmentation at varying time strides. Despite this, WALRUS is the top model across domains with no clear areas of weakness.
>
> F4 - Inductive bias and pretraining method
>
> The transformer backbone trades domain-specific inductive bias for generality. While Fourier-based architectures like DPOT excel on smooth linear dynamics, such advantages tend to be fragile across domains. For pretraining, no clear winner has emerged among competing approaches (autoregressive, in-context, masked denoising), so we opt for the autoregressive strategy as it is one of the more common approaches and reflects our targeted downstream task of surrogate modeling.
>
> ---
>
> **Planned revisions:** We will add a main-text cross-reference for adaptive tokenization, an algorithm box for topology-aware sampling in Appendix A.2, and expanded per-sample statistics in appendix tables. We hope these responses and revisions address the reviewer's concerns.

---

> > ### Author Rebuttal · Reviewer_gKDp · 2026-04-03
> >
> > Thank you very much for providing the response. Several concerns have been resolved. I'd like to some follow-up questions/comments.
> >
> > EM1. Thank you for the clarification and for pointing to the longer-horizon results. The explanation regarding trajectory decorrelation in chaotic systems is helpful.
> >
> > As a follow-up, since pointwise metrics such as VRMSE may become less informative in long-horizon regimes, have the authors considered evaluating performance using alternative metrics (e.g., statistical or distributional measures commonly used for chaotic systems)? I recognize that such metrics may be application-specific and not universally applicable across all benchmark systems. However, they could provide additional insight. This is not meant as a request for additional experiments, but rather to understand whether the authors have considered or plan to explore such evaluations.
> >
> > F2/F3. I appreciate that, for foundation models, performance should be viewed as the result of a combined system including architecture, training strategy, and pretraining data, rather than any single component in isolation. In this sense, Walrus represents a strong and well-engineered piece of work.
> >
> > However, this also makes comparisons more nuanced, as the baseline models are developed under different combinations of architecture, training procedures, and data, and in some cases target somewhat different problem scopes. For example, models such as Poseidon are designed with a more specific focus on continuum dynamics.
> >
> > It would therefore strengthen the paper to more explicitly discuss these differences and clarify how they affect the interpretation of the empirical results.

---

> > > ### Author Response · Authors · 2026-04-07
> > >
> > > Thank you for the thoughtful follow-up. We have another 5000 characters, so we can go into more details with the discussion this time.
> > >
> > > **EM1** - After reviewing feedback from both yourself and others, we've decided to include the spatially averaged W1 metric suggested by another reviewer computed over normalized fields. This is over time rather than samples as these are deterministic models up to jitter. For stationary systems, time-averaged and ensemble-averaged distributions should converge by ergodicity, making this a reasonable proxy while for non-stationary systems, this still measures the similarity of the distribution over observed states. We feel this is a better fit than traditional turbulence analysis which is only applicable to a subset of systems studied here.
> > >
> > > The main reason we avoided traditional turbulence metrics is the lack of broad applicability. Many conventional turbulence analysis metrics (TKE spectra, Reynolds stresses, etc.) rely on time-averaged statistics or the Reynolds decomposition and assume statistical stationarity in time. Ie, $p(u(x), t) = p(u(x), t’)\ \forall\ t, t’ > T^*$. The stationary assumption eventually applies to several systems we explore (MHD, TRL, RB, Active Matter) but many others (RTI, acoustics, SPN, some Gray-Scott) are transient systems in the process of evolving from unstable initial conditions toward equilibrium. Reaching these equilibria (which are often just steady states) would take significantly longer burn-in windows.
> > >
> > > The other reason we initially favored pointwise comparisons is that while pointwise metrics lose value in the long run, empirically, it appears that most systems analyzed are within the limits of predictability. What we mean by this is that the error suggests we can still do better than choosing another random sample from the stationary distribution. For chaotic systems, minor perturbations lead to pointwise RMSE plateauing around $\sqrt 2 \sigma$ where $\sigma$ is the standard deviation of the field values. This occurs as trajectories become decorrelated and pointwise comparisons across frames eventually become equivalent to comparing independent samples from the stationary distribution. In Figures 12/13, it appears that most models either are not plateauing in this way or have not reached that plateau.
> > >
> > > Given these considerations, we feel the averaged W1 suggested by Reviewer yTH7 complements the existing evaluation well and plan on adding it to the appendix metrics. While it does not capture neighborhood structure, it provides a meaningful assessment of distributional fidelity over rollouts that is well-defined across all evaluation scenarios regardless of stationarity. We’ll include the definition and a note similar to the above explaining where this is most applicable compared to the pointwise metrics.
> > >
> > > **F2/F3** - This is a fair assessment. While these are all continuum dynamics models, Walrus does target a broader range of applications beyond just fluids and this focus on breadth can incur costs (particularly computationally). Experiments demonstrate this focus on breadth was productive, but we agree that making it clearer where this led to trade-offs would strengthen the paper. We can add a bit of discussion here to the main text, but mainly add an appendix section going into more exact details on what these points are  (efficiency trade-offs in Markovian design, cost of including 3D, scope of attention mechanism, or pretraining corpus).
> > >
> > > ---
> > >
> > > Again, we really appreciate the useful feedback. If we were able to address the reviewer's remaining concerns, we'd ask that you consider updating your score. Otherwise, if this format allows for further replies, we'd be happy to continue discussion.

---

### Decision · Program_Chairs · 2026-04-30

**Decision:**

Accept (spotlight)

**Comment:**

This paper introduces Walrus, a large-scale foundation model for continuum dynamics trained across diverse 2D/3D physical systems. Reviewers agreed the work is ambitious and technically strong, with particularly compelling breadth of experiments, consistent empirical gains over prior models, and meaningful systems contributions. Patch jittering and the overall training framework were viewed as useful additions likely to influence future work.

The main concerns relate to positioning and evaluation rather than correctness: reviewers requested clearer articulation of novelty versus integration, more careful discussion of baseline fairness (especially regarding scale and context length), improved clarity of key components, and additional analysis of long-horizon behavior using complementary metrics. The authors’ rebuttal addressed many of these points and committed to clarifications and additional results.

Overall, the consensus is positive. Despite moderate novelty in some components, the strong empirical results, scale, and potential impact support acceptance, provided the final version incorporates the requested clarifications and analysis.